# Microbes with higher metabolic independence are enriched in human gut microbiomes under stress

Iva Veseli[1,2]*, Yiqun T Chen[3], Matthew S Schechter[2,4], Chiara Vanni[5], Emily C Fogarty[2,4], Andrea R Watson[2,4], Bana Jabri[2], Ran Blekhman[2], Amy D Willis[6], Michael K Yu[7], Antonio Fernàndez-Guerra[8], Jessika Füssel[2,9]*, A Murat Eren[2,9,10,11,12]*

[1]Biophysical Sciences Program, The University of Chicago, Chicago, United States; [2]Department of Medicine, The University of Chicago, Chicago, United States; [3]Data Science Institute and Department of Biomedical Data Science, Stanford University, Stanford, United States; [4]Committee on Microbiology, The University of Chicago, Chicago, United States; [5]MARUM Center for Marine Environmental Sciences, University of Bremen, Bremen, Germany; [6]Department of Biostatistics, University of Washington, Seattle, United States; [7]Toyota Technological Institute at Chicago, Chicago, United States; [8]Lundbeck Foundation GeoGenetics Centre, GLOBE Institute, University of Copenhagen, Copenhagen, Denmark; [9]Institute for Chemistry and Biology of the Marine Environment, University of Oldenburg, Oldenburg, Germany; [10]Marine 'Omics Bridging Group, Max Planck Institute for Marine Microbiology, Bremen, Germany; [11]Helmholtz Institute for Functional Marine Biodiversity, Oldenburg, Germany; [12]Alfred Wegener Institute for Polar and Marine Research, Bremerhaven, Germany

*For correspondence:
iva.veseli@gmail.com (IV);
jessika.fuessel@uol.de (JF);
meren@hifmb.de (AME)

Competing interest: The authors declare that no competing interests exist.

## eLife Assessment

This study presents an **important** new bioinformatics tool for normalizing gene copy number from metagenomic assemblies and applies it to gain functional insights into the loss of microbial diversity during conditions of stress. The inclusion of extensive computational validation makes this a **compelling** study that raises intriguing new hypotheses regarding the impact of disease states on the gut microbiome. This paper will likely be of broad interest to researchers studying the role of complex microbial communities in host health and disease.

**Abstract** A wide variety of human diseases are associated with loss of microbial diversity in the human gut, inspiring a great interest in the diagnostic or therapeutic potential of the microbiota. However, the ecological forces that drive diversity reduction in disease states remain unclear, rendering it difficult to ascertain the role of the microbiota in disease emergence or severity. One hypothesis to explain this phenomenon is that microbial diversity is diminished as disease states select for microbial populations that are more fit to survive environmental stress caused by inflammation or other host factors. Here, we tested this hypothesis on a large scale, by developing a software framework to quantify the enrichment of microbial metabolisms in complex metagenomes as a function of microbial diversity. We applied this framework to over 400 gut metagenomes from individuals who are healthy or diagnosed with inflammatory bowel disease (IBD). We found that high metabolic independence (HMI) is a distinguishing characteristic of microbial communities associated with individuals diagnosed with IBD. A classifier we trained using the normalized copy numbers of 33

HMI-associated metabolic modules not only distinguished states of health vs IBD, but also tracked the recovery of the gut microbiome following antibiotic treatment, suggesting that HMI is a hallmark of microbial communities in stressed gut environments.

## Introduction

The human gut is home to a diverse assemblage of microbial cells that form complex communities (*Coyte et al., 2015*). This gut microbial ecosystem is established almost immediately after birth and plays a lifelong role in human well-being by contributing to immune system maturation and functioning (*Belkaid and Hand, 2014*; *Maynard et al., 2012*), extracting dietary nutrients (*Hijova, 2019*), providing protection against pathogens (*Khosravi and Mazmanian, 2013*), metabolizing drugs (*Zimmermann et al., 2019*), and more (*Knight et al., 2017*). There is no universal definition of a healthy gut microbiome (*Fan and Pedersen, 2021*), but associations between host disease states and changes in microbial community composition have sparked great interest in the therapeutic potential of gut microbes (*Cani, 2018*; *Sorbara and Pamer, 2022*) and led to the emergence of hypotheses that directly link disruptions of the gut microbiome to noncommunicable diseases of complex etiology (*Byndloss and Bäumler, 2018*).

Inflammatory bowel diseases (IBDs), which describe a heterogeneous group of chronic inflammatory disorders (*Shan et al., 2022*), represent an increasingly common health risk around the globe (*Kaplan, 2015*). Understanding the role of gut microbiota in IBD has been a major area of focus in human microbiome research. Studies focusing on individual microbial taxa that typically change in relative abundance in IBD patients have proposed a range of host-microbe interactions that may contribute to disease manifestation and progression (*Joossens et al., 2011*; *Schirmer et al., 2019*; *Henke et al., 2019*; *Machiels et al., 2014*). However, even within well-constrained cohorts, a large proportion of variability in the taxonomic composition of the microbiota is unexplained, and the proportion of variability explained by disease status is low (*Gevers et al., 2014*; *Schirmer et al., 2018*; *Lloyd-Price et al., 2019*; *Khan et al., 2019*). As neither individual taxa nor broad changes in microbial community composition yield effective predictors of disease (*Knox et al., 2019*; *Lee and Chang, 2021*), the role of gut microbes in the etiology of IBD – or the extent to which they are bystanders to disease – remains unclear (*Khan et al., 2019*).

The marked decrease in microbial diversity in IBD is often associated with the loss of Firmicutes populations and an increased representation of a relatively small number of taxa, such as Bacteroides, Enterococcaceae, and others (*Prindiville et al., 2000*; *Saitoh et al., 2002*; *Sartor, 2006*; *Rhodes, 2007*; *Devkota et al., 2012*; *Machiels et al., 2014*; *Vineis et al., 2016*; *Lloyd-Price et al., 2019*). Why a handful of taxa that also typically occur in healthy individuals in lower abundances (*Lee and Chang, 2021*; *Nishida et al., 2018*) tend to dominate the IBD microbiome is a fundamental but open question to gain insights into the ecological underpinnings of the gut microbial ecosystem under IBD. Going beyond taxonomic summaries, a recent metagenome-wide metabolic modeling study revealed a significant loss of cross-feeding partners as a hallmark of IBD, where microbial interactions were disrupted in IBD-associated microbial communities compared to those found in healthy individuals (*Marcelino et al., 2023*). This observation is in line with another recent work that proposed that the extent of 'metabolic independence' (characterized by the genomic presence of a set of key metabolic modules for the synthesis of essential nutrients) is a determinant of microbial survival in IBD (*Watson et al., 2023*). It is conceivable that the disrupted metabolic interactions among microbes observed in IBD (*Marcelino et al., 2023*) indicate an environment that lacks the ecosystem services provided by a complex network of microbial interactions, and selects for those organisms that harness high metabolic independence (HMI) (*Watson et al., 2023*). This interpretation offers an ecological mechanism to explain the dominance of populations with specific metabolic features in IBD and requires further investigation.

Here, we implemented a high-throughput, taxonomy-independent strategy to estimate metabolic capabilities of microbial communities directly from metagenomes and investigate whether the enrichment of populations with HMI predicts IBD in the human gut. We benchmarked our findings using representative genomes associated with the human gut and their distribution in healthy individuals as well as those who have been diagnosed with IBD. Our results suggest that high metabolic potential (indicated by a set of 33 largely biosynthetic metabolic modules) provides enough signal to

**eLife digest** The human gut hosts an array of microbes that form a complex community beginning shortly after birth. These microbes prime the immune system, help extract nutrients from the diet and offer protection against pathogens. Decades of research have shown that individuals who suffer from inflammatory bowel diseases (IBD) or other systemic disorders tend to have far less variety of gut microbes compared to healthy individuals. Yet it remains unclear to what extent the difference in microbial diversity is the cause of the disease or a consequence of it.

In 2023, a study suggested that the usual teamwork between different kinds of microbes breaks down during disease. Many microbes depend on each other to provide certain nutrients, while others can survive on their own. It could be that people with IBD lose most of the 'dependent' microbes and retain those that are more self-sufficient and thus able to survive in the stressed and deteriorating gut environment.

To test this hypothesis, Veseli et al. – who are part of the research group that performed the 2023 study – developed a computer program to quantify self-sufficient gut microbes in large numbers of stool samples collected from healthy individuals and patients with IBD. This revealed that individuals with IBD had higher numbers of self-sufficient microbes, while healthy people also harbored microbes that depended on others for the provision of essential metabolites. External disruptions to the gut homeostasis, such as antibiotics, resulted in a similar selection for independent microbes.

These findings support the idea that changes in the gut microbiome are more likely a by-product of disease, rather than its cause and offer important ecological clues for microbial therapies that aim to restore gut health. While this perspective assigns a more neutral role for gut microbial communities in non-transmissible diseases, more research is needed to see if an enrichment of self-sufficient microbes could negatively influence disease progression.

consistently distinguish gut microbiomes under stress from those that are in homeostasis, providing deeper insights into adaptive processes initiated by stress conditions that promote rare members of gut microbiota to dominance during disease.

## Results and discussion

We compiled 2893 publicly available stool metagenomes from 13 different studies, 5 of which explicitly studied the IBD gut microbiome (*Supplementary file 1a–c*). The average sequencing depth varied across individual datasets (4.2 to 60.3 million paired-end reads, with a median value of 21.4, *Supplementary file 1c*). To improve the sensitivity and accuracy of our downstream analyses that depend on metagenomic assembly, we excluded samples with less than 25 million reads, resulting in a set of 408 relatively deeply sequenced metagenomes from 10 studies (26.4 to 61.9 million paired-end reads, with a median value of 37.0, *Supplementary file 1b*, Appendix 1, Methods), which we de novo assembled individually. The final dataset included individuals who were healthy (n = 229), diagnosed with IBD (n = 101), or suffered from other gastrointestinal conditions ('non-IBD', n = 78). In accordance with previous observations of reduced microbial diversity in IBD (*Kostic et al., 2014*; *Nagalingam and Lynch, 2012*; *Knox et al., 2019*), the estimated number of populations based on the occurrence of bacterial single-copy core genes (SCGs) present in these metagenomes was higher in healthy individuals than those diagnosed with IBD (*Appendix 1—figure 1*, *Supplementary file 1*).

### Estimating normalized copy numbers of metabolic modules from metagenomic assemblies

Gaining insights into microbial metabolism requires accurate estimates of the presence/absence and completion of metabolic modules. While a myriad of tools address this task for single genomes (*Machado et al., 2018*; *Aziz et al., 2008*; *Arkin et al., 2018*; *Palù et al., 2022*; *Shaffer et al., 2020*; *Geller-McGrath et al., 2023*; *Zorrilla et al., 2021*; *Zhou et al., 2022*; *Zimmermann et al., 2021*), working with complex environmental metagenomes poses additional challenges due to the large number of organisms that are present in metagenomic assemblies. A few tools can estimate community-level metabolic potential from metagenomes without relying on the reconstruction of

individual population genomes or reference-based approaches (*Ye and Doak, 2009*; *Karp et al., 2021*; *Supplementary file 5*). These high-level summaries of module presence and redundancy in a given environment are suitable for most surveys of metabolic capacity, particularly for microbial communities of similar richness. However, since the frequency of observed metabolic modules will increase as the number of distinct microbial populations in a habitat increases, investigations of metabolic determinants of survival across environmental conditions with substantial differences in microbial richness may suffer from ambiguous observations from quantitative data. For instance, the estimated copy number of a given metabolic module may be identical between two metagenomes but its enrichment may be relatively higher in the metagenome with a lower alpha diversity, revealing its potential role in overcoming environment-specific selective pressures that influence an entire community. Working solely with raw copy numbers of metabolic modules without a normalization

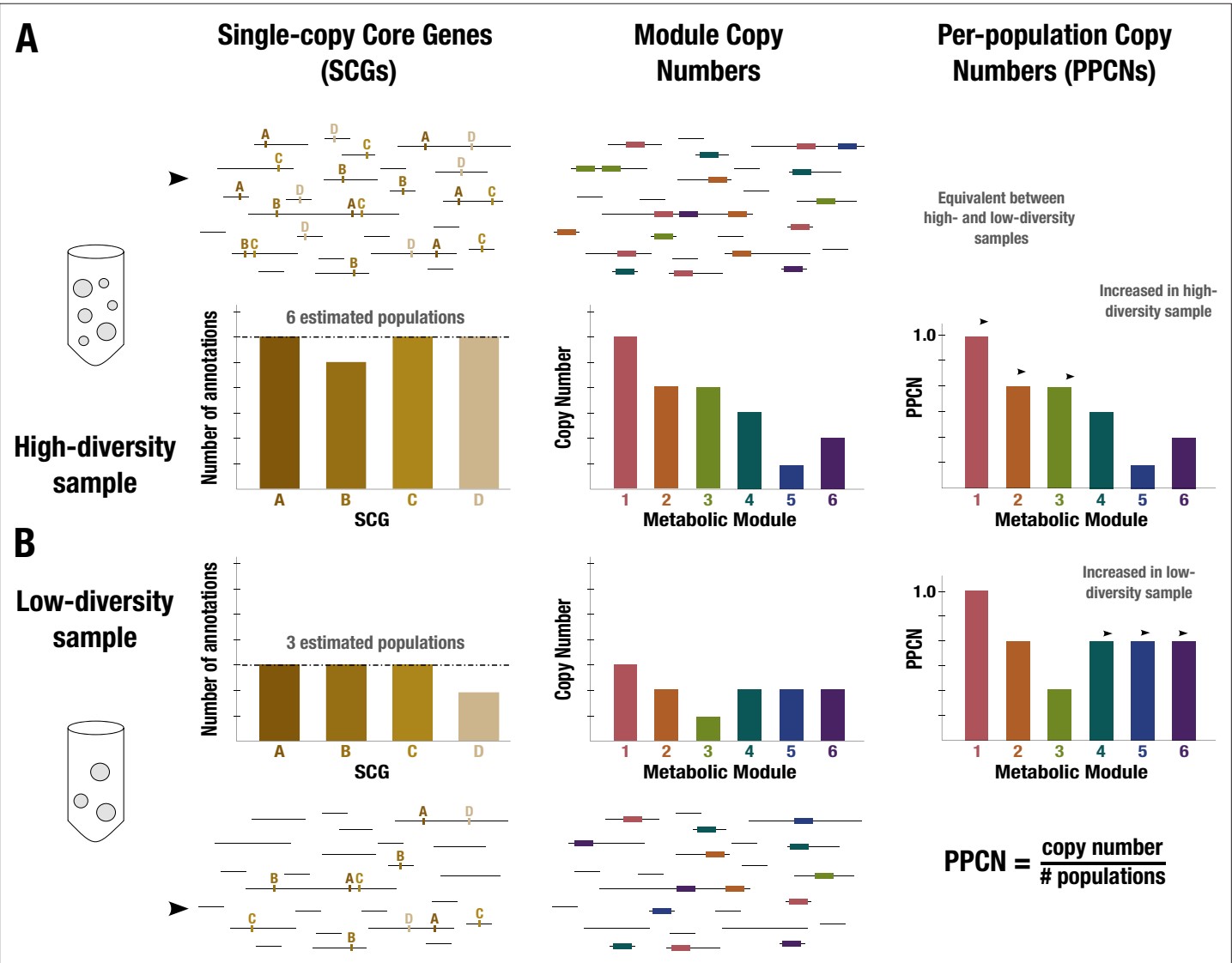

**Figure 1.** Conceptual diagram of per-population copy number (PPCN) calculation. Each step of the calculation is demonstrated in (**A**) for a sample with high diversity (six microbial populations) and in (**B**) for a sample with low diversity (three populations). Metagenome sequences are shown as black lines. The left panel shows the single-copy core genes (SCGs) annotated in the metagenome (indicated by letters), with a barplot showing the counts for different SCGs. The dashed black line indicates the mode of the counts, which is taken as the estimate of the number of populations. The middle panel shows the annotations of metabolic modules (indicated by boxes and numerically labeled), with a barplot showing the copy number of each module (for more details on how this copy number is computed, see Appendix 1 and *Appendix 1—figure 2*). The right panel shows the equation for PPCN, with the barplots indicating the PPCN values for each metabolic module in each sample and arrows differentiating between different types of modules based on the comparison of their normalized copy numbers between samples.

step that considers the microbial richness will thus shroud potentially critical insights. To quantify the differential enrichment of metabolic modules between metagenomes generated from healthy individuals and those from individuals diagnosed with IBD, we implemented a new software framework (https://anvio.org/m/anvi-estimate-metabolism) that reconstructs metabolic modules from genomes and metagenomes, and a means to calculate the per-population copy number (PPCN) of modules in metagenomes to account for potential differences in microbial richness (Methods, Appendices 1 and 2). Briefly, the PPCN estimates the proportion of microbes in a community with a particular metabolic capacity (*Figure 1*, *Appendix 1—figure 2*) by normalizing observed metabolic module copy numbers with the 'number of microbial populations in a given metagenome', which we estimate using the SCGs without relying on the reconstruction of individual genomes. Our validation of this method using simulated metagenomic data demonstrated that it is accurate in capturing metagenome-level metabolic capacity relative to genome-level metabolic capacity estimated from the same data (Appendix 2, *Supplementary file 6*).

## Key biosynthetic modules are enriched in microbial populations from IBD samples

To gain insight into potential metabolic determinants of microbial survival in the IBD gut environment, we assessed the distribution of metabolic modules within samples from each group (IBD and healthy) with and without using PPCN normalization. Without normalizing, module copy numbers were overall higher in healthy samples (*Figure 2A*) and modules exhibited weak differential occurrence between cohorts (*Figure 2B and C*, *Appendix 1—figure 3*). The application of PPCN reversed this trend, and most metabolic modules were elevated in IBD (*Appendix 1—figure 5*). This observation is influenced by two independent aspects of the healthy and IBD microbiota. The first one is the increased representation of microbial organisms with smaller genomes in healthy individuals (*Watson et al., 2023*), which increases the likelihood that the overall copy number of a given metabolic module is below the actual number of populations. In contrast, one of the hallmarks of the IBD microbiota is the generally increased representation of organisms with larger genomes (*Watson et al., 2023*). The second aspect is that the generally higher diversity of microbes in healthy individuals increases the denominator of the PPCN. This results in a greater reduction in the PPCN of metabolic modules that are not shared across all members of the diverse gut microbial populations in health.

To go beyond this general trend and identify modules that were highly conserved in the IBD group, we first selected those that passed a relatively high statistical significance threshold in our enrichment test (Wilcoxon rank-sum test, FDR-adjusted p-value < 2e-10). We then accounted for effect size by ranking these modules according to the difference between their median PPCN in IBD samples and their median PPCN in healthy samples, and keeping only those in the top 50% (which translated to an effect size threshold of > 0.12). This stringent filtering revealed a set of 33 metabolic modules that were significantly enriched in metagenomes obtained from individuals diagnosed with IBD (*Figure 2D and E*), 17 of which matched the modules that were associated with HMI previously (*Watson et al., 2023*; *Figure 2F*). This result suggests that the PPCN normalization is an important step in comparative analyses of metabolisms between samples with different levels of microbial diversity.

The majority of the metabolic modules that were significantly enriched in the microbiomes of IBD patients encoded biosynthetic capabilities (23 out of 33) that resolved to amino acid metabolism (33%), carbohydrate metabolism (21%), cofactor and vitamin biosynthesis (15%), nucleotide biosynthesis (12%), lipid biosynthesis (6%), and energy metabolism (6%) (*Supplementary file 2a*). In contrast to previous reports based on reference genomes (*Gevers et al., 2014*; *Morgan et al., 2012*), amino acid synthesis and carbohydrate metabolism were not reduced in the IBD gut microbiome in our dataset. Rather, our results were in accordance with a more recent finding that predicted amino acid secretion potential is increased in the microbiomes of individuals with IBD (*Heinken et al., 2021*).

Within our set of 33 modules that were enriched in IBD, it is notable that all the biosynthesis and central carbohydrate modules are directly or indirectly linked via shared enzymes and metabolites. Each enriched module shared on average 25.6% of its enzymes and 40.2% of metabolites with the other enriched modules, and overall 18.2% of enzymes and 20.4% of compounds across these modules were shared (*Supplementary file 2a*). Thus, modules may be enriched not just due to the importance of their immediate end products, but also because of their role in the larger metabolic network. The few standalone modules that were enriched included the efflux pump MepA and

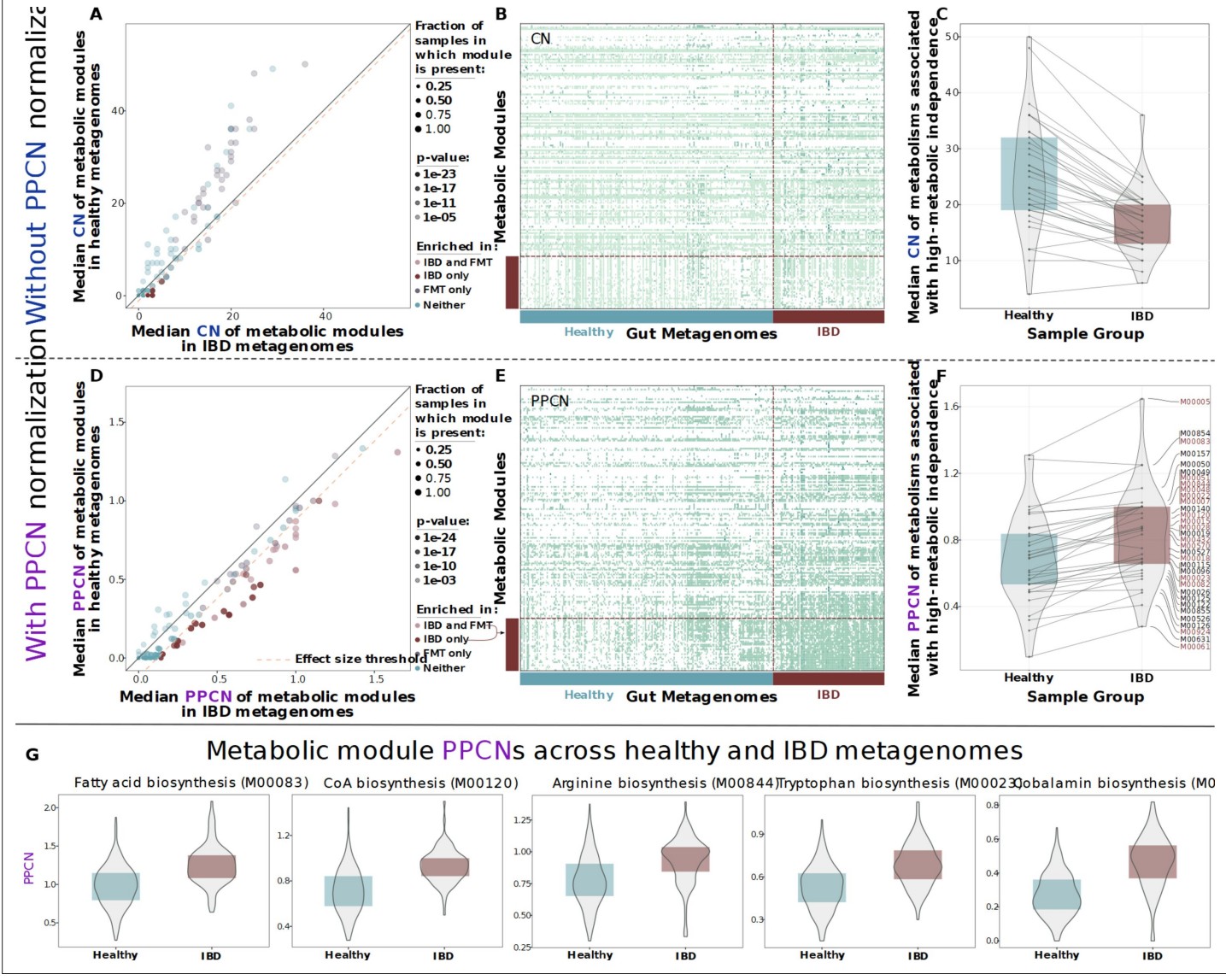

**Figure 2.** Comparison of metabolic potential across healthy and inflammatory bowel disease (IBD) cohorts. Panels **A–C** show unnormalized copy number data and the remaining panels show normalized per-population copy number (PPCN) data. (**A**) Scatterplot of module copy number in IBD samples (x-axis) and healthy samples (y-axis). Transparency of points indicates the p-value of the module in a Wilcoxon rank-sum test for enrichment (based on copy number data), and color indicates whether the module is enriched in the IBD samples (in this study), enriched in the good colonizers from the fecal microbiota transplant (FMT) study (*Watson et al., 2023*), or enriched in both. The pink dashed line indicates the effect size threshold applied to modules when determining their enrichment in IBD. (**B**) Heatmap of unnormalized copy numbers for all modules. The 33 modules that were found to be IBD-enriched based on PPCN data are highlighted by the red bar on the left. Sample group is indicated by the blue (healthy) and red (IBD) bars on the bottom. (**C**) Boxplots of median copy number for each module enriched in the FMT colonizers from *Watson et al., 2023*, in the healthy samples (blue) and the IBD samples (red). Solid lines connect the same module in each plot. (**D**) Scatterplot of module PPCN values in IBD samples (x-axis) and healthy samples (y-axis). Transparency and color of points are defined as in panel **A**, but based on PPCN data. The pink dashed line indicates the effect size threshold applied to modules when determining their enrichment in IBD. (**E**) Heatmap of PPCN values for all modules. Side bars defined as in (**B**). (**F**) Boxplots of median PPCN values for modules enriched in the FMT colonizers from *Watson et al., 2023*, in the healthy samples (blue) and the IBD samples (red). Lines defined as in (**D**). Modules that were also enriched in the IBD samples (in this study) are highlighted in red. (**G**) Boxplots of PPCN values for individual modules in the healthy samples (blue) and the IBD samples (red). All example modules were enriched in both this study and in *Watson et al., 2023*.

the beta-lactam resistance system, which are associated with drug resistance. These capacities may provide an advantage since antibiotics are a common treatment for IBDs (*Nitzan et al., 2016*), but are not necessarily related to the systematic enrichment of biosynthesis modules that likely provide resilience to general environmental stress rather than to a specific stressor such as antibiotics.

Microbiome data generated by different groups can result in systematic biases that may outweigh biological differences between otherwise similar samples (*Lozupone et al., 2013*; *Sinha et al., 2017*; *Clausen and Willis, 2022*). The potential impact of such biases constitutes an important consideration for meta-analyses such as ours that analyze publicly available metagenomes from multiple sources. To account for cohort biases, we conducted an analysis of our data on a per-cohort basis. All cohorts within a given group exhibited similar distributions of PPCN values, which indicates that the trends we observed above result from an overall between-group difference in signal rather than a cohort-specific signal (*Appendix 1—figure 6B and C*). Another source of potential bias stems from the annotation efficiency of gene function. For instance, we noticed that, independent of the annotation strategy, a smaller proportion of genes resolved to known functions in metagenomic assemblies of samples from healthy individuals compared to the samples from individuals who were diagnosed with IBD (*Appendix 1—figure 4*). This highlights the possibility that samples from healthy individuals merely appear to harbor less metabolic capabilities due to missing annotations. Indeed, we found that the normalized copy numbers of most metabolic modules were reduced in the healthy group, where 84% of KEGG modules (98 out of 118) have significantly lower median copy numbers (*Appendix 1—figure 5C*, Appendix 1). While the presence of a bias between the two cohorts is clear, the source of this bias and its implications are not. One hypothesis that could explain this phenomenon is that the increased proportion of unknown functions in environments where populations with low metabolic independence (LMI) thrive is due to our inability to identify distant homologs of even well-studied functions in poorly studied novel genomes through public databases. If true, this would indeed impair our ability to annotate genes using state-of-the-art functional databases and bias metabolic module completion estimates. Such a limitation would warrant a careful reconsideration of common workflows and studies that rely on public resources to characterize gene function in complex environments. Another hypothesis that could explain our observation is that the general absence of microbes with smaller genomes in culture had a historical impact on the characterization of novel functions that represent a relatively larger fraction of their gene repertoire. If true, this would suggest that the unknown functions are unlikely essential for well-studied metabolic capabilities. Furthermore, HMI and LMI genomes may be indistinguishable with respect to the distribution of such novel genes, but the increased number of genes in HMI genomes that resolve to well-studied metabolisms would reduce the proportion of known functions in LMI genomes, and thus in metagenomes where they thrive. While testing these hypotheses falls outside the scope of our work, we find the latter hypothesis more likely due to examples in literature that have successfully identified genes that belong to known metabolisms in some of the most obscure organisms via annotation strategies similar to those we have used in our work (*Jaffe et al., 2020*; *Farag et al., 2020*).

Taken together, these results (1) demonstrate that the PPCN normalization is an important consideration for investigations of metabolic enrichment in complex microbial communities as a function of microbial diversity, and (2) reveal that the enrichment of HMI populations in an environment offers a high-resolution marker to resolve different levels of environmental stress.

## Reference genomes with higher metabolic independence are overrepresented in the gut metagenomes of individuals with IBD

So far, our findings demonstrate an overall, metagenome-level trend of increasing HMI within gut microbial communities as a function of IBD status without considering the individual genomes that contribute to this signal. Since we can measure the extent of metabolic independence as defined in our study based on the completion of a few key metabolic modules for any given genome, we next considered a genome-based approach to further benchmark our findings by investigating whether publicly available microbial genomes that appear to have properties of HMI are more commonly found in individuals diagnosed with IBD.

To identify a set of microbial genomes that are generally associated with the human gut environment, we cast a broad net by surveying the ecology of 19,226 genomes in the Genome Taxonomy Database (GTDB) (*Parks et al., 2022*) that belonged to three major phyla: Bacteroidetes, Firmicutes, and Proteobacteria, which represent the vast majority of microbial diversity in the human gut environment (*Woting and Blaut, 2016*; *Turnbaugh et al., 2009*). As these phyla also include a large number of taxa that primarily occur outside of the human gut, we only kept for downstream analyses those that were detected in at least 2% of the participants of the Human Microbiome Project (HMP) (*Human*

*Microbiome Project Consortium, 2012*; *Appendix 1—figure 8*, Methods). Of the final set of 338 reference genomes that passed our filters, 258 (76.3%) resolved to Firmicutes, 60 (17.8%) to Bacteroidetes, and 20 (5.9%) to Proteobacteria. Most of these genomes resolved to families common to the colonic microbiota, such as Lachnospiraceae (30.0%), Ruminococcaceae/Oscillospiraceae (23.1%), and Bacteroidaceae (10.1%) (*Arumugam et al., 2011*), while 5.9% belonged to poorly studied families with temporary code names (*Supplementary file 3a*). Finally, we performed a more comprehensive read recruitment analysis on this smaller set of genomes using all deeply sequenced metagenomes from cohorts that included healthy, non-IBD, and IBD samples (*Figure 3*). This provided us with a quantitative summary of the detection patterns of GTDB genome representatives common to the human gut across our dataset.

We assumed that a given genome had HMI if its average completeness of the 33 HMI-associated metabolic modules was at least 80%, equivalent to a summed metabolic independence score of 26.4 (Methods). Given the number of ways a genome can pass or fail this threshold, this arbitrary cutoff has significant shortcomings, which was demonstrated by the fact that several species in the *Bacteroides* group were not classified as HMI despite their frequent dominance of the gut microbiome of individuals with IBD (*Saitoh et al., 2002*; *Wexler, 2007*; *Vineis et al., 2016*) (Appendix 1). That said, the genomes that were classified as HMI by this approach were consistently higher in their detection and abundance in IBD samples (*Figure 3A*). It is likely that there are multiple ways to have HMI which are not fully captured by the 33 IBD-enriched metabolic modules identified in this study. Across all genomes, the mean metabolic independence score was 24.0 (Q1: 19.9, Q3: 25.7). We identified 17.5% (59) of the reference genomes as HMI. HMI genomes were on average substantially larger (3.8 Mbp) than non-HMI genomes (2.9 Mbp) and encoded more genes (3634 vs 2683 genes, respectively), which is in accordance with the reduced metabolic potential of non-HMI populations (*Supplementary file 3a*). Our read recruitment analysis showed that HMI reference genomes were present in a significantly higher proportion of IBD samples compared to non-HMI genomes (*Figure 3B*, $p < 1e-5$, Wilcoxon rank-sum test). Similarly, the fraction of HMI populations was significantly higher within a given IBD sample compared to samples classified as 'non-IBD' and those from healthy individuals (*Figure 3C*, $p < 1e-24$, Kruskal-Wallis rank-sum test). In contrast, the detection of HMI populations and non-HMI populations was similar in healthy individuals (*Figure 3B*, $p = 0.267$, Wilcoxon rank-sum test). The intestinal environment of healthy individuals likely supports both HMI and non-HMI populations, wherein 'metabolic diversity' is maintained by metabolic interactions such as cross-feeding. Indeed, loss of cross-feeding interactions in the gut microbiome appears to be associated with a number of human diseases, including IBD (*Marcelino et al., 2023*). This interpretation is further supported by the fact that the top two HMI-associated modules are required for the synthesis of cobalamin from glutamate. Auxotrophy for cobalamin biosynthesis is common among gut bacteria that rely on cross-feeding for this essential cofactor (*Degnan et al., 2014a*; *Magnúsdóttir et al., 2015*; *Kelly et al., 2019*) (Appendix 1).

Overall, the classification of reference gut genomes as HMI and their enrichment in individuals diagnosed with IBD strongly supports the contribution of HMI to stress resilience of individual microbial populations. We note that survival in a disturbed gut environment will likely require a wide variety of additional functions that are not covered in the list of metabolic modules we consider to determine HMI status – e.g., see *Degnan et al., 2014b*; *Martens et al., 2014*; *Zong et al., 2020*; *Feng et al., 2020*; *Goodman et al., 2009*; *Powell et al., 2016*. Indeed, there may be many ways for a microbe to be metabolically independent, and our strategy likely failed to identify some HMI populations. Nonetheless, these data suggest that HMI serves as a reliable proxy for the identification of microbial populations that are particularly resilient.

## HMI-associated metabolic potential predicts general stress on gut microbes

Our analysis identified HMI as an emergent property of gut microbial communities associated with individuals diagnosed with IBD. This community-level signal translates to individual microbial populations and provides insights into the microbial ecology of stressed gut environments. HMI-associated metabolic modules were enriched at the community level, and microbial populations encoding these modules were more prevalent in individuals with IBD than in healthy individuals. Furthermore, the copy number of these modules and the proportion of HMI populations reflect the severity of environmental

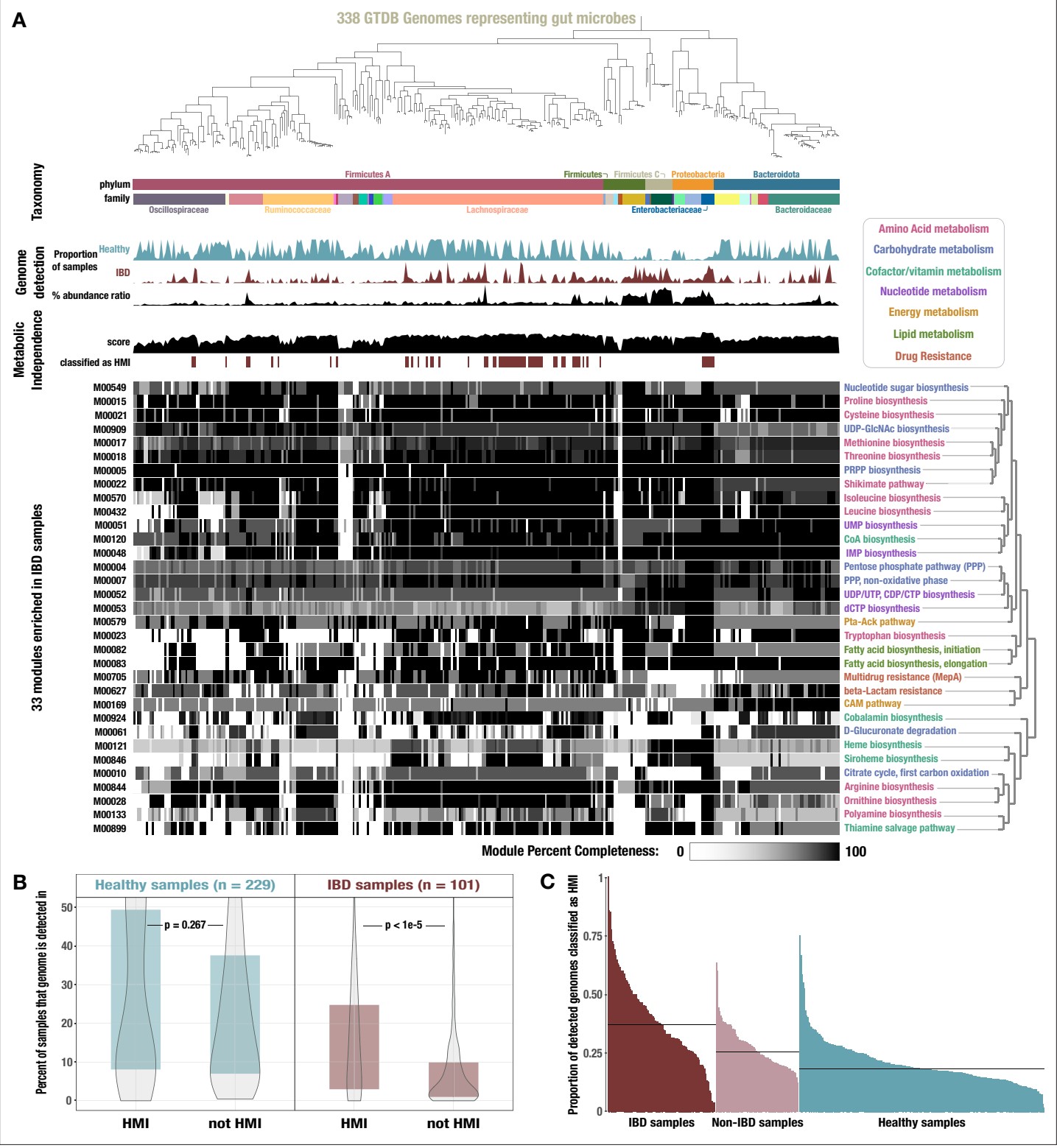

**Figure 3.** Identification of high metabolic independence (HMI) genomes and their distribution across gut samples. (**A**) The phylogeny of 338 gut-associated genomes from the Genome Taxonomy Database (GTDB) along with the following data, from top to bottom: taxonomic classification as assigned by GTDB; proportion of healthy samples with at least 50% detection of the genome sequence; proportion of inflammatory bowel disease (IBD) samples with at least 50% detection of the genome sequence; square-root normalized ratio of percent abundance in IBD samples to percent abundance in healthy samples; metabolic independence score (sum of completeness scores of 33 HMI-associated metabolic modules); whether (red) or not (white)

*Figure 3 continued on next page*

*Figure 3 continued*

the genome is classified as having HMI with a threshold score of 26.4; heatmap of completeness scores for each of the 33 HMI-associated metabolic modules (0% completeness is white and 100% completeness is black). Module name is shown on the right and colored according to its category of metabolism. (**B**) Boxplot showing the proportion of healthy (blue) or IBD (red) samples in which genomes of each class are detected ≥ 50%, with p-values from a Wilcoxon rank-sum test on the underlying data. (**C**) Barplot showing the proportion of detected genomes (with ≥ 50% genome sequence covered by at least 1 read) in each sample that are classified as HMI, for each group of samples. The black lines show the median for each group: 37.0% for IBD samples, 25.5% for non-IBD samples, and 18.4% for healthy samples.

stress and translate to host health states (*Appendix 1—figure 5B*, *Figure 3C*). The ecological implications of these observations suggest that HMI may serve as a predictor of general stress in the human gut environment.

So far, efforts to identify IBD using microbial markers have presented classifiers based on (1) taxonomy in pediatric IBD patients (*Papa et al., 2012*; *Gevers et al., 2014*), (2) community composition in combination with clinical data (*Halfvarson et al., 2017*), (3) untargeted metabolomics and/ or species-level relative abundance from metagenomes (*Franzosa et al., 2019*), and (4) k-mer-based sequence variants in metagenomes that can be linked to microbial genomes associated with IBD (*Reiter et al., 2022*). Performance varied both between and within studies according to the target classes and data types used for training and validation of each classifier (*Supplementary file 4a*). For those studies reporting accuracy, a maximum accuracy of 77% was achieved based on either metabolite profiles (for prediction of IBD subtype) (*Franzosa et al., 2019*) or k-mer-based sequence variants (for differentiating between IBD and non-IBD samples) (*Reiter et al., 2022*). Some studies reported performance as area under the receiver operating characteristic curve (AUROCC), a typical measure of classifier utility describing both sensitivity (ability to correctly identify the disease) and specificity (ability to correctly identify absence of disease). For this metric the highest value was 0.92, achieved by *Franzosa et al., 2019*, when using metabolite profiles, with or without species abundance data, for classifying IBD vs non-IBD. However, the majority of these classifiers were trained and tested on a relatively small group of individuals that all come from the same region, i.e., clinical studies confined to a specific hospital. Though some had high performance, they either relied on data that are inaccessible to most laboratories and clinics considering that untargeted metabolomics analyses are difficult to reproduce (*Koek et al., 2011*; *Lin et al., 2020*), or they required complex k-mer-based models without the resolution to differentiate gradients in host health (*Reiter et al., 2022*). These classifiers thus have limited translational potential across global clinical settings and do not provide an ecological framework to explain the observed shifts in community composition and activity. For practical use as a diagnostic tool, a microbiome-based classifier for IBD should rely on an ecologically meaningful, easy to measure, and high-level signal that is robust to host variables like lifestyle, geographical location, and ethnicity. HMI could potentially fill this gap as a metric related to the ecological filtering that defines microbial community changes in the IBD gut microbiome.

We trained a logistic regression classifier to explore the applicability of HMI as a noninvasive diagnostic tool for IBD. The classifier's predictors were the PPCNs of IBD-enriched metabolic modules in a given metagenome. Across the 330 deeply sequenced IBD and healthy samples included in this analysis, the classifier had high sensitivity and specificity (*Figure 4*). It correctly identified (on average) 76.8% of samples from individuals diagnosed with IBD and 89.5% of samples representing healthy individuals, for an overall accuracy of 85.6% and an average AUROCC of 0.832 (*Supplementary file 4c*). Our model outperforms (*Gevers et al., 2014*; *Halfvarson et al., 2017*; *Reiter et al., 2022*) or has comparable performance to *Franzosa et al., 2019*; *Papa et al., 2012* the previous attempts to classify IBD from fecal samples in more restrictively defined cohorts. It also has the advantage of being a simple model, utilizing a relatively low number of features compared to the other classifiers. Determining whether such a model has broader utility as a diagnostic tool requires further research and validation; however, these results demonstrate the potential of HMI as an accessible diagnostic marker of IBD. Due to the lack of time-series studies that include individuals in the pre-diagnosis phase of IBD development, we cannot test the applicability of HMI to predict IBD onset (*Lloyd-Price et al., 2019*).

Yet, the gradient of metabolic independence reflected by per-population module copy number and the relative increase in the number of HMI populations detected in non-IBD samples (*Appendix 1—figure 5B*, *Figure 3C*) suggests that the degree of HMI in the gut microbiome may be indicative of general gut stress, such as the stress induced by antibiotic use. Antibiotics can cause long-lasting

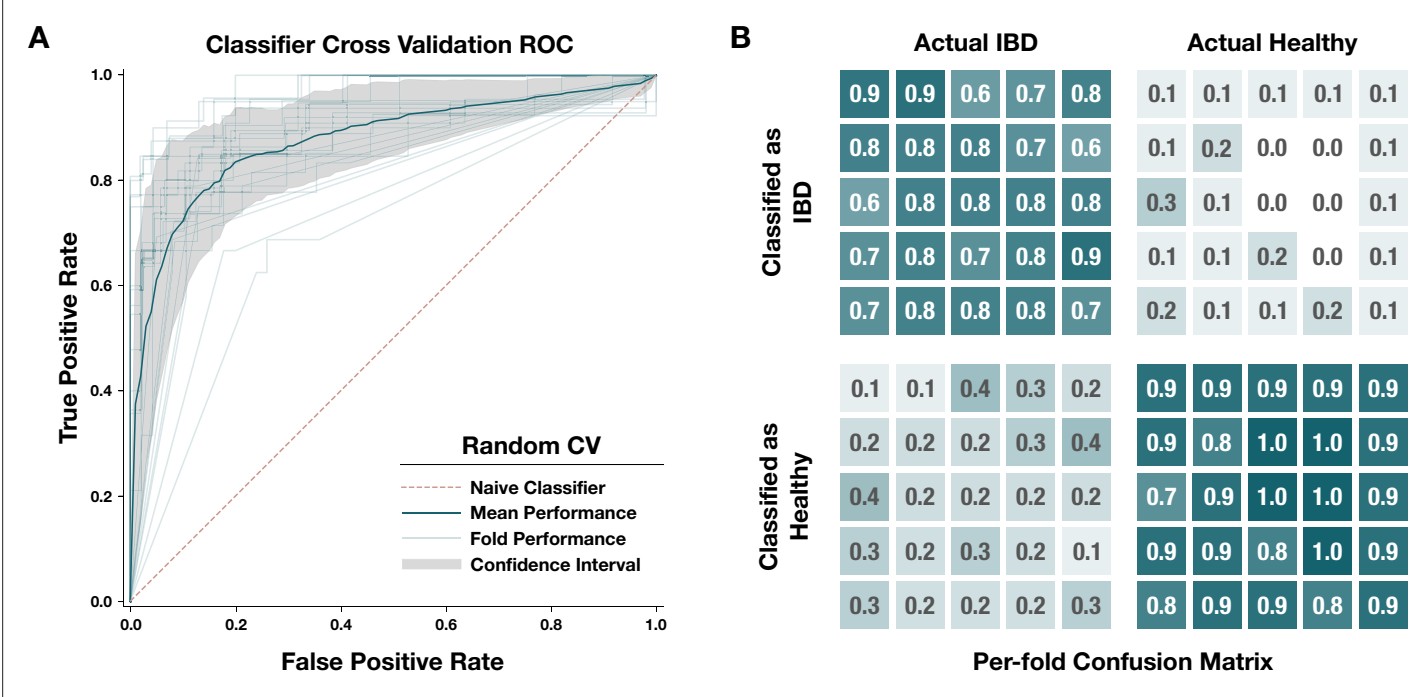

**Figure 4.** Performance of our metagenome classifier trained on per-population copy numbers (PPCNs) of inflammatory bowel disease (IBD)-enriched modules. (**A**) Receiver operating characteristic (ROC) curves for 25-fold cross-validation. Each fold used a random subset of 80% of the data for training and the other 20% for testing. In each fold, we calculated a set of IBD-enriched modules from the training dataset and used the PPCN of these modules to train a logistic regression model whose performance was evaluated using the test dataset. Light gray lines show the ROC curve for each fold, the dark blue line shows the mean ROC curve, the gray area delineates the confidence interval for the mean ROC, and the pink dashed line indicates the benchmark performance of a naive (random guess) classifier. (**B**) Confusion matrix for each fold of the random cross-validation. Categories of classification, from top left to bottom right, are: true positives (correctly classified IBD samples), false positives (incorrectly classified healthy samples), false negatives (incorrectly classified IBD samples), and true negatives (correctly classified healthy samples). Each fold is represented by a box within each category. Opacity of the box indicates the proportion of samples in that category, and the actual proportion is written within the box with one significant digit. Underlying data for this matrix can be accessed in *Supplementary file 4d*.

perturbations of the gut microbiome – including reduced diversity, emergence of opportunistic pathogens, increased microbial load, and development of highly resistant strains – with potential implications for host health (*Ramirez et al., 2020*). We applied our metabolism classifier to a metagenomic dataset that reflects the changes in the microbiome of healthy people before, during, and up to 6 months following a 4-day antibiotic treatment (*Palleja et al., 2018*). The resulting pattern of sample classification corresponds to the posttreatment decline and subsequent recovery of species richness documented in the study by *Palleja et al., 2018*.

All pretreatment samples were classified as 'healthy' followed by a decline in the proportion of 'healthy' samples to a minimum 8 days posttreatment, and a gradual increase until 180 days posttreatment, when over 90% of samples were classified as 'healthy' (*Figure 5*, *Supplementary file 4b*). In other words, the increase in the HMI metric serves as an indicator of stress in the gut microbiome, regardless of whether that stress arises from the IBD condition or the application of antibiotics. These observations support the role of HMI as an ecological driver of microbial resilience during gut stress caused by a variety of environmental perturbations and demonstrate its diagnostic power in reflecting gut microbiome state.

## Conclusions

Overall, our observations that stem from the analysis of hundreds of reference genomes, deeply sequenced gut metagenomes, and multiple categories of human disease states suggest that environmental stress in the human gut – whether it is associated with inflammation, cancer, or antibiotic use – promotes the survival and relative expansion of microbial populations with HMI. These results establish HMI as a high-level metric to classify gradients of human health states through

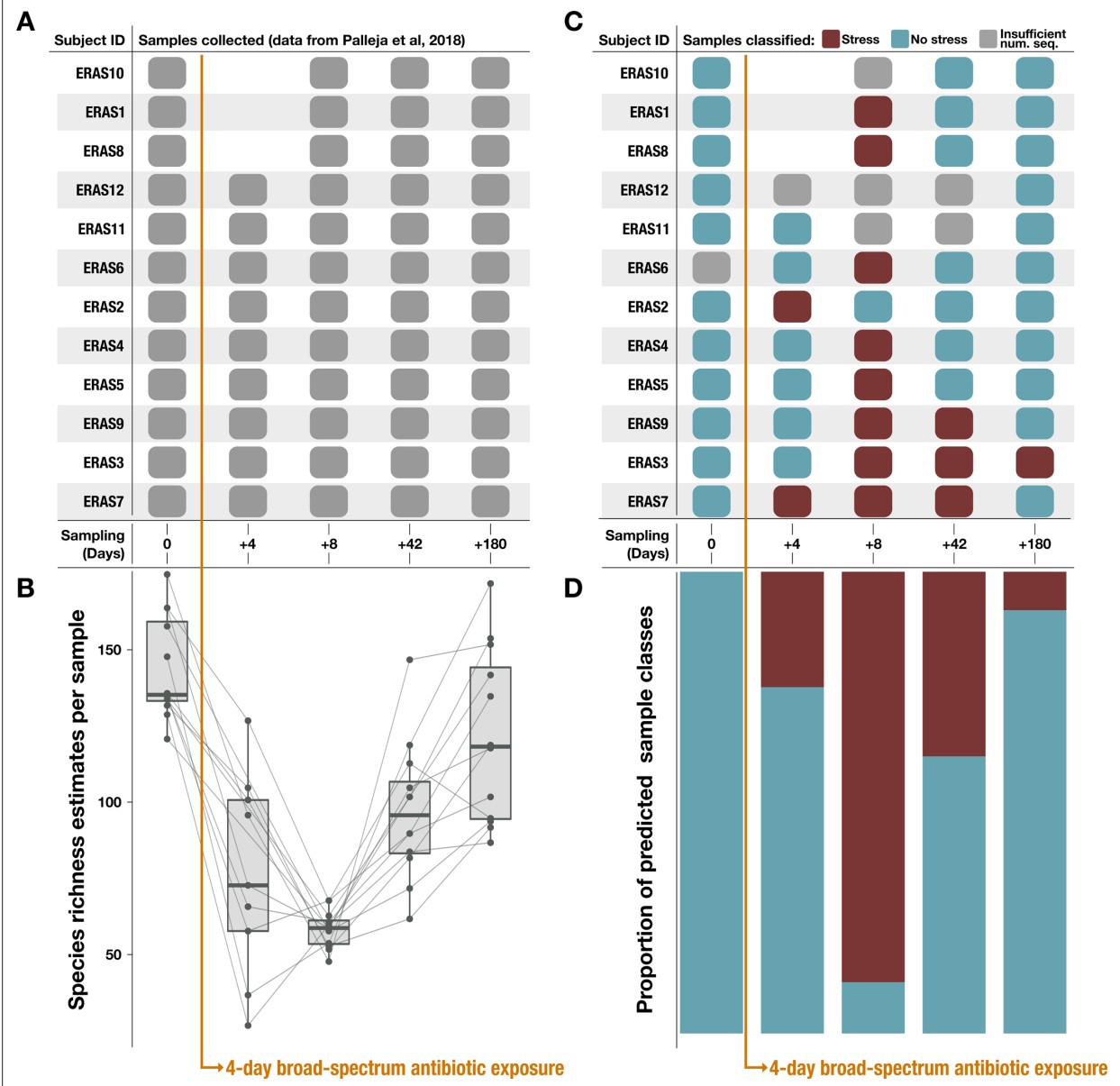

**Figure 5.** Classification results on an antibiotic time-series dataset from *Palleja et al., 2018*. Note that antibiotic treatment was taken on days 1–4. (**A**) Samples collected per subject during the time series. (**B**) Species richness data (figure created using data from *Palleja et al., 2018*). (**C**) Classification of each sample by the metabolism classifier profiled in *Figure 4*. Samples with insufficient sequencing depth were not classified. (**D**) Proportion of classes assigned to samples per day in the time series. Samples classified as 'healthy' by the model were considered to have 'no stress' (blue), while samples classified as inflammatory bowel disease ('IBD') were considered to be under 'stress' (red).

the gut microbiota that is robust to ethnic, geographical, or lifestyle factors. Taken together with recent evidence that models altered ecological relationships within gut microbiomes under stress due to disrupted metabolic cross-feeding (*Heinken et al., 2021*; *Marcelino et al., 2023*), our data support the hypothesis that the reduction in microbial diversity, or more generally 'dysbiosis', is an emergent property of microbial communities responding to disease pathogenesis or other external factors such as antibiotic use that disrupt the gut microbial ecosystem. This paradigm depicts microbes as bystanders by default, rather than perpetrators or drivers of noncommunicable human diseases, and provides an ecological framework to explain the frequently observed reduction in microbial diversity associated with IBD and other noncommunicable human diseases and disorders.

## Methods

A bioinformatics workflow that further details all analyses described below and gives access to reproducible data products is available at the URL https://merenlab.org/data/ibd-gut-metabolism/.

### A new framework for metabolism estimation

We developed a new program 'anvi-estimate-metabolism' (https://anvio.org/m/anvi-estimate-metabolism), which uses gene annotations to estimate 'completeness' and 'copy number' of metabolic modules that are defined in terms of enzyme accession numbers. By default, this tool works on metabolic modules from the KEGG MODULE database (*Kanehisa et al., 2012*; *Kanehisa et al., 2023*) which are defined by KEGG KOfams (*Aramaki et al., 2020*), but user-defined modules based on a variety of functional annotation sources are also accepted as input. Completeness estimates describe the percentage of steps (typically, enzymatic reactions) in a given metabolic module that are encoded in a genome or a metagenome. Likewise, copy number summarizes the number of distinct sets of enzyme annotations that collectively encode the complete module. This program offers two strategies for estimating metabolic potential: a 'stepwise' strategy with equivalent treatment for alternative enzymes – i.e., enzymes that can catalyze the same reaction in a given metabolic module – and a 'pathwise' strategy that accounts for all possible variations of the module. Appendix 1 file includes more information on these two strategies and the completeness/copy number calculations. For the analysis of metagenomes, we used stepwise copy number of KEGG modules. Briefly, the calculation of stepwise copy number is done as follows: the copy number of each step in a module (typically, one chemical reaction or conversion) is individually evaluated by translating the step definition into an arithmetic expression that summarizes the number of annotations for each required enzyme. In cases where multiple enzymes or an enzyme complex are needed to catalyze the reaction, we take the minimum number of annotations across these components. In cases where there are alternative enzymes that can each catalyze the reaction individually, we sum the number of annotations for each alternative. Once the copy number of each step is computed, we then calculate the copy number of the entire module by taking the minimum copy number across all the individual steps. The use of minimums results in a conservative estimate of module copy number such that only copies of the module with all enzymes present are counted. For the analysis of genomes, we calculated the stepwise completeness of KEGG modules. This calculation is similar to the one described above for copy number, except that the step definition is translated into a Boolean expression that, once evaluated, indicates the presence or absence of each step in the module. Then, the completeness of the modules is computed as the proportion of present steps in the module.

### Metagenomic datasets and sample groups

We acquired publicly available gut metagenomes from 13 different studies (*Le Chatelier et al., 2013*; *Feng et al., 2015*; *Franzosa et al., 2019*; *Lloyd-Price et al., 2019*; *Qin et al., 2012*; *Quince et al., 2015*; *Rampelli et al., 2015*; *Raymond et al., 2016*; *Schirmer et al., 2018*; *Vineis et al., 2016*; *University of Sydney, 2022*; *Wen et al., 2017*; *Xie et al., 2016*). The studies were chosen based on the following criteria: (1) they included shotgun metagenomes of fecal matter (primarily stool, but some ileal pouch luminal aspirate samples (*Vineis et al., 2016*) are also included); (2) they sampled from people living in industrialized countries in the case where a study (*Rampelli et al., 2015*) included samples from hunter-gatherer populations, only the samples from industrialized areas were included in our analysis; (3) they included samples from people with IBD and/or they included samples from people without gastrointestinal (GI) disease or inflammation; and (4) clear metadata differentiating between case and control samples was available. A full description of the studies and samples can be found in *Supplementary file 1a–c*. We grouped samples according to the health status of the sample donor. Briefly, the 'IBD' group of samples includes those from people diagnosed with Crohn's disease, ulcerative colitis (UC), or pouchitis. The 'non-IBD' group contains non-IBD controls, which includes both healthy people presenting for routine cancer screenings and people with benign or nonspecific symptoms that are not clinically diagnosed with IBD. Colorectal cancer patients from *Feng et al., 2015*, were also put into the 'non-IBD' group on the basis that tumors in the GI tract may arise from local inflammation (*Kraus and Arber, 2009*) and represent a source of gut stress without an accompanying diagnosis of IBD. Finally, the 'HEALTHY' group contains samples from people without GI-related diseases or inflammation. Note that only control or pretreatment samples were taken from the studies

covering type 2 diabetes (*Qin et al., 2012*), ankylosing spondylitis (*Wen et al., 2017*), antibiotic treatment (*Raymond et al., 2016*), and dietary intervention *University of Sydney, 2022*; these controls were all assigned to the 'HEALTHY' group. At least one study (*Le Chatelier et al., 2013*) included samples from obese people, and these were also included in the 'HEALTHY' group.

## Processing of metagenomes

We made single assemblies of most gut metagenomes using the anvi'o metagenomics workflow implemented in the program 'anvi-run-workflow' (*Shaiber et al., 2020*). This workflow uses Snakemake (*Köster and Rahmann, 2012*), and a tutorial is available at the URL https://merenlab.org/anvio-workflows/. Briefly, the workflow includes quality filtering using 'iu-filter-quality-minoche' (*Eren et al., 2013*); assembly with IDBA-UD (*Peng et al., 2012*) (using a minimum contig length of 1000); gene calling with Prodigal v2.6.3 (*Hyatt et al., 2010*); tRNA identification with tRNAscan-SE v2.0.7 (*Chan and Lowe, 2019*); as well as annotations of ribosomal RNAs (*Seemann, 2018*), single-copy core genes (SCGs) for Bacteria, Archaea, and Protista, KEGG KOfams (*Aramaki et al., 2020*), NCBI Clusters of Orthologous Group (COGs) (*Galperin et al., 2021*), and Pfam protein families (release 33.1, *Mistry et al., 2021*). The aforementioned annotations relied on HMMER v3.3 (*Eddy, 2011*) as well as Diamond v0.9.14.115 (*Buchfink et al., 2015*). As part of this workflow, all single assemblies were converted into anvi'o contigs databases. Samples from *Vineis et al., 2016*, were processed differently because they contained merged reads rather than individual paired-end reads: no further quality filtering was run on these samples, we assembled them individually using MEGAHIT (*Li et al., 2015*), and we used the anvi'o contigs workflow to perform all subsequent steps described for the metagenomics workflow above. Note that we used a version of KEGG downloaded in December 2020 (for reproducibility, the hash of the KEGG snapshot available via 'anvi-setup-kegg-kofams' is 45b7cc2e4fdc). Additionally, the annotation program 'anvi-run-kegg-kofams' includes a heuristic for annotating hits with bitscores that are just below the KEGG-defined threshold (*Kananen et al., 2025*), which is further described at https://anvio.org/m/anvi-run-kegg-kofams/.

## Genomic dataset

We downloaded all reference genomes for 'species' cluster representatives from the GTDB, release 95.0 (*Parks et al., 2018*; *Parks et al., 2020*), and processed them with the same anvi'o gene annotation workflow described above.

## Estimation of the number of microbial populations per metagenome

We used SCG sets belonging to each domain of microbial life (Bacteria, Archaea, Protista) to estimate the number of populations from each domain present in a given metagenomic sample. For each domain, we calculated the number of populations by taking the mode of the number of copies of each SCG in the set. We then summed the number of populations from each domain to get a total number of microbial populations within each sample. We accomplished this using SCG annotations provided by 'anvi-run-hmms' (which was run during metagenome processing) and a custom script relying on the anvi'o class 'NumGenomesEstimator' (see reproducible workflow).

## Removal of samples with low sequencing depth

We observed that, at lower sequencing depths, our estimates for the number of populations in a metagenomic sample were moderately correlated with sequencing depth (*Appendix 1—figure 1*, R > 0.5). These estimates rely on having accurate counts of SCGs, so we hypothesized that lower-depth samples were systematically missing SCGs, especially from populations with lower abundance. Since accurate population number estimates are critical for proper normalization of module copy numbers, keeping these lower-depth samples would have introduced a bias into our metabolism analyses. To address this, we removed samples with low sequencing depth from downstream analyses using a sequencing depth threshold of 25 million reads, such that the remaining samples exhibited a weaker correlation (R < 0.5) between sequencing depth and number of estimated populations. We kept samples for which both the R1 file and the R2 file contained at least 25 million reads (and for the *Vineis et al., 2016*, dataset, we kept samples containing at least 25 million merged reads). This produced our final sample set of 408 metagenomes.

## Estimation of normalized module copy numbers in metagenomes

We ran 'anvi-estimate-metabolism', in genome mode and with the '--add-copy-number' flag, on each individual metagenome assembly to compute stepwise copy numbers for KEGG modules from the combined gene annotations of all populations present in the sample. We then divided these copy numbers by the number of estimated populations within each sample to obtain a PPCN for each module.

## Selection of IBD-enriched modules

We used a one-sided Mann-Whitney-Wilcoxon test with an FDR-adjusted p-value threshold of p ≤ 2e-10 on the per-sample PPCN values for each module individually to identify the modules that were most significantly enriched in the IBD sample group compared to the healthy group. We calculated the median PPCN of each metabolic module in the IBD samples, and again in the healthy samples. After filtering for p-values ≤ 2e-10, we also applied a minimum effect size threshold based on the median PPCN in each group ($M_{IBD}$ - $M_{Healthy}$ ≥ 0.12) – this threshold was calculated by taking the mean effect size over all modules that passed the p-value threshold. The set originally contained 34 modules that passed both thresholds, but we removed one redundant module (M00006) which represents the first half of another module in the set (M00004).

## Test for enrichment of biosynthesis modules

We used a one-sided Fisher's exact test (also known as hypergeometric test, see e.g., *Boyle et al., 2004*) for testing the independence between the metabolic modules identified to be IBD-enriched (i.e. using the methods described in 'Selection of IBD-enriched modules') and functionality (i.e. modules annotated to be involved in biosynthesis).

## Module comparisons

Because the 33 IBD-enriched modules were selected using PPCNs of healthy and IBD samples, statistical tests comparing PPCN distributions for these modules need to be interpreted with care, because the hypotheses were selected and tested on the same dataset (*Fithian et al., 2014*). Therefore, to assess the statistical validity of the identified IBD-enriched modules, we performed the following repeated sample-split analysis: we first randomly split the IBD and healthy samples into the equal-sized training and validation sets. We select IBD-enriched modules in the training set using the Mann-Whitney-Wilcoxon test, and then compute the p-values on the validation set. We repeat this sample split analysis 1000 times with an FDR-adjusted p-value threshold of 1e-10 on the first split; most identified modules (89.4%; 95% CI: [87.5%, 91.3%]) on the training sets remain significant at a slightly less stringent threshold (1e-8) on the validation sets. This indicates that the approach we used to identify IBD-enriched modules yields stable and statistically significant results on this dataset.

## Metagenome classification

We trained logistic regression models to classify samples as 'IBD' or 'healthy' using PPCNs of IBD-enriched modules as features. We ran a 25-fold cross-validation pipeline on the set of 330 healthy and IBD metagenomes in our analysis, using an 80% train – 20% test random split of the data in each fold. The pipeline included selection of IBD-enriched modules within the training samples using the same strategy as described above, followed by training and testing of a logistic regression model as implemented in the 'sklearn' Python package. We set the 'penalty' parameter of the model to 'None' and the 'max_iter' parameter to 20,000 iterations, and we used the same random state in each fold to ensure changes in performance only come from differences in the training data rather than differences in model initialization. To summarize the overall performance of the classifier, we took the mean (over all folds) of each performance metric.

We trained a final classifier using the 33 IBD-enriched modules selected earlier from the entire set of 330 healthy and IBD metagenomes. We then applied this classifier to the metagenomic samples from *Palleja et al., 2018*, which we processed in the same way as the other samples in our analysis (including removal of samples with low sequencing depth and calculation of PPCNs of KEGG modules for use as input features to the classifier model).

## Identification of gut microbial genomes from the GTDB

We took 19,226 representative genomes from the GTDB species clusters belonging to the phyla Firmicutes, Bacteroidetes, and Proteobacteria, which are most common in the human gut microbiome

(*Woting and Blaut, 2016*). To evaluate which of these genomes might represent gut microbes in a computationally tractable manner, we ran the anvi'o 'EcoPhylo' workflow (https://anvio.org/m/ecophylo) to contextualize these populations within 150 healthy gut metagenomes from the HMP (*Human Microbiome Project Consortium, 2012*). Briefly, the EcoPhylo workflow (1) recovers sequences of a gene family of interest from each genome and metagenomic sample in the analysis, (2) clusters resulting sequences and picks representative sequences using mmseqs2 (*Steinegger and Söding, 2017*), and (3) uses the representative sequences to rapidly summarize the distribution of each population cluster across the metagenomic samples through metagenomic read recruitment analyses. Here, we used the Ribosomal Protein S6 as our gene of interest, since it was the most frequently assembled SCG in our set of GTDB genomes. We clustered the Ribosomal Protein S6 sequences from GTDB genomes at 94% nucleotide identity.

To identify genomes that were likely to represent gut microbes, we selected genomes whose ribosomal protein S6 belonged to a gene cluster where at least 50% of the representative sequence was covered (i.e. detection ≥ 0.5×) in more than 10% of samples (i.e. n > 15). There are 100 distinct individuals represented in the 150 HMP gut metagenomes – 56 of which were sampled just once and 46 of which were sampled at 2 or 3 time points – so this threshold is equivalent to detecting the genome in 5–15% of individuals. From this selection we obtained a set of 836 genomes; however, these were not exclusively gut microbes, as some non-gut populations have similar ribosomal protein S6 sequences to gut microbes and can therefore pass this selection step. To eliminate these, we mapped our set of 330 healthy and IBD metagenomes to the 836 genomes using the anvi'o metagenomics workflow and extracted genomes whose entire sequence was at least 50% covered (i.e. detection ≥ 0.5×) in over 2% (n > 6) of these samples. Our final set of 338 genomes was used in downstream analysis.

## Genome phylogeny

To create the phylogeny, we identified the following ribosomal proteins that were annotated in at least 90% (n = 304) of the genomes: Ribosomal_S6, Ribosomal_S16, Ribosomal_L19, Ribosomal_L27, Ribosomal_S15, Ribosomal_S20p, Ribosomal_L13, Ribosomal_L21p, Ribosomal_L20, and Ribosomal_L9_C. We used 'anvi-get-sequences-for-hmm-hits' to extract the amino acid sequences for these genes, align the sequences using MUSCLE v3.8.1551 (*Edgar, 2004*), and concatenate the alignments. We used trimAl v1.4.rev15 (*Capella-Gutiérrez et al., 2009*) to remove any positions containing more than 50% of gap characters from the final alignment. Finally, we built the tree with IQtree v2.2.0.3 (*Minh et al., 2020*), using the WAG model and running 1000 bootstraps.

## Determination of HMI status for genomes

We estimated metabolic potential for each genome with 'anvi-estimate-metabolism' (in genome mode) to get stepwise completeness scores for each KEGG module, and then we used the script 'anvi-script-estimate-metabolic-independence' to give each genome a metabolic independence score based on completeness of the 33 IBD-enriched modules. Briefly, the latter script calculates the score by summing the completeness scores of each module of interest. Genomes were classified as having HMI if their score was greater than or equal to 26.4. We calculated this threshold by requiring these 33 modules to be, on average, at least 80% complete in a given genome.

## Genome distribution across sample groups

We mapped the gut metagenomes from the healthy, non-IBD, and IBD groups to each genome using the anvi'o metagenomics workflow in reference mode. We used 'anvi-summarize' to obtain a matrix of genome detection across all samples. We summarized this data as follows: for each genome, we computed the proportion of samples in each group in which at least 50% of the genome sequence was covered by at least 1 read (≥ 50% detection). For each sample, we calculated the proportion of detected genomes that were classified as HMI. We also computed the percent abundance of each genome in each sample by dividing the number of reads mapping to that genome by the total number of reads in the sample.

## Visualizations

We used ggplot2 (*Wickham, 2016*) to generate most of the initial data visualizations. The phylogeny and heatmap in *Figure 3* were generated by the anvi'o interactive interface and the ROC curves in *Figure 4* were generated using the pyplot package of matplotlib (*Hunter, 2007*). These visualizations were refined for publication using Inkscape, an open-source graphical editing software that is available at https://inkscape.org/.

## Supplementary table files

Supplementary Table files and our Appendix files can be accessed at https://doi.org/10.6084/m9.figshare.22679080.

## Acknowledgements

We thank Christopher Quince for advice on statistical significance testing. IV acknowledges support from the National Science Foundation Graduate Research Fellowship under Grant No. 1746045; ADW acknowledges support from the National Institutes of General Medical Sciences under R35 GM133420. YTC acknowledges support from the Stanford Data Science Postdoctoral Fellowship. RB acknowledges support from the National Institutes of Health under R35 GM128716. ECF acknowledges support from the University of Chicago International Student Fellowship. Additional support for ECF and AME came from an NIH NIDDK grant (RC2 DK122394) to AME.

# Additional information

### Funding

| Funder | Grant reference number | Author |
|---|---|---|
| National Science Foundation Graduate Research Fellowship Program | 1746045 | Iva Veseli |
| National Institutes of General Medical Sciences | R35 GM133420 | Amy D Willis |
| Stanford Data Science Postdoctoral Fellowship | | Yiqun T Chen |
| National Institutes of Health | R35 GM128716 | Ran Blekhman |
| University of Chicago International Student Fellowship | | Emily C Fogarty |
| National Institutes of Health | RC2 DK122394 | A Murat Eren |

The funders had no role in study design, data collection and interpretation, or the decision to submit the work for publication. Open access funding provided by Max Planck Society.

### Author contributions

Iva Veseli, Conceptualization, Data curation, Software, Formal analysis, Validation, Visualization, Methodology, Writing – original draft, Writing – review and editing; Yiqun T Chen, Formal analysis, Visualization, Writing – review and editing, Supported statistical analyses; Matthew S Schechter, Software, Writing – review and editing; Chiara Vanni, Antonio Fernàndez-Guerra, Resources, Writing – review and editing; Emily C Fogarty, Andrea R Watson, Resources, Data curation, Writing – review and editing; Bana Jabri, Ran Blekhman, Amy D Willis, Michael K Yu, Investigation, Writing – review and editing; Jessika Füssel, Conceptualization, Formal analysis, Supervision, Writing – original draft, Writing – review and editing; A Murat Eren, Conceptualization, Software, Formal analysis, Visualization, Writing – original draft, Project administration, Writing – review and editing

### Author ORCIDs

Iva Veseli (ID) https://orcid.org/0000-0003-2390-5286
Yiqun T Chen (ID) http://orcid.org/0000-0002-4100-1507
Matthew S Schechter (ID) https://orcid.org/0000-0002-8435-3203
Chiara Vanni (ID) http://orcid.org/0000-0002-1124-1147
Emily C Fogarty (ID) http://orcid.org/0000-0002-8957-9922
Andrea R Watson (ID) http://orcid.org/0000-0003-0128-6795
Bana Jabri (ID) http://orcid.org/0000-0001-7427-4424
Ran Blekhman (ID) http://orcid.org/0000-0003-3218-613X
Amy D Willis (ID) https://orcid.org/0000-0002-2802-4317
Michael K Yu (ID) http://orcid.org/0000-0002-9560-2017
Antonio Fernàndez-Guerra (ID) https://orcid.org/0000-0002-8679-490X
Jessika Füssel (ID) http://orcid.org/0000-0002-4210-2318
A Murat Eren (ID) https://orcid.org/0000-0001-9013-4827

Reviewer #1 (Public review): https://doi.org/10.7554/eLife.89862.3.sa1
Reviewer #2 (Public review): https://doi.org/10.7554/eLife.89862.3.sa2
Reviewer #3 (Public review): https://doi.org/10.7554/eLife.89862.3.sa3
Author response https://doi.org/10.7554/eLife.89862.3.sa4

---

# Additional files

### Supplementary files

Supplementary file 1. Samples and cohorts used in this study. (a) Description of studies/cohorts providing publicly available gut metagenomes from healthy people, non-inflammatory bowel disease (IBD) controls, and people with IBD. For each study, we note the sample groups it contributes metagenomes to; whether or not those samples were sufficiently deeply sequenced to be included in the main analyses; the country of origin of the samples; the sample type (fecal metagenome or ileal pouch luminal aspirate); the number of samples it contributes to each group before and after applying the sequencing depth threshold; and cohort details/exclusions as described within the study. (b) Description of 408 samples included in the primary analyses of this manuscript (i.e. those with sufficient sequencing depth of $\geq$ 25 million reads), including their associated diagnosis (ulcerative colitis (UC), Crohn's disease (CD), non-IBD, healthy, colorectal cancer with adenoma (CRC_ADENOMA), or colorectal cancer with carcinoma (CRC_CARCINOMA)); study of origin; sample group; sequencing depth; and number of microbial populations estimated to be represented within the metagenome. (c) Description of all samples initially considered and their SRA accession numbers. (d) The number of gene calls and the number/proportion of annotations per gene call for KOfams, Clusters of Orthologous Groups (COGs), and Pfams in each sample. (e) The number of genes with at least one functional annotation and the number of tRNAs in each sample from the subset of deeply sequenced samples. (f) Description of the 57 antibiotic time-series gut metagenomes from *Palleja et al., 2018* used for classifier testing, including SRA accession number; sampling day in the time series; sequencing depth; and estimated numbers of microbial populations represented in the sample.

Supplementary file 2. Metabolism data in metagenomes. (a) Description of the 33 KEGG modules enriched in inflammatory bowel disease (IBD) samples, including: module name, KEGG categorization, and definition; their median per-population copy numbers (PPCNs) in the healthy sample group and IBD sample group; the p-value, FDR-adjusted p-value, and W statistic from the per-module Wilcoxon rank-sum test used to determine enrichment in IBD; the difference between its median PPCN in IBD samples and median PPCN in healthy samples ('effect size'); the fraction of samples in which the module occurs with nonzero copy number; whether the module is also enriched in the high metabolic independence (HMI) populations analyzed in *Watson et al., 2023*; the number of total enzymes in the module; the number of total compounds in the modules; and the numbers and proportions of shared enzymes or compounds between this module and the other IBD-enriched modules. (b) Description of all 179 KEGG modules with nonzero copy number in at least one metagenome. Most of the columns match the corresponding column in sheet (b) with the exception of the 'enrichment status' column, which indicates whether the module was found to be enriched in the IBD samples in this study ('IBD_ENRICHED'), in the high-metabolic independence genomes in *Watson et al., 2023* ('HMI_ENRICHED'), in both ('HMI_AND_IBD'), or

in neither ('OTHER'). (c) Matrix of stepwise copy number of each module in each deeply sequenced gut metagenome. (d) Per-population copy number of each module in each deeply sequenced gut metagenome in the IBD, non-IBD, and healthy sample groups. (e) Per-population copy number of each module in each antibiotic time-series sample from *Palleja et al., 2018*.

Supplementary file 3. Genome Taxonomy Database (GTDB) genome data. (a) List of 338 GTDB representative genomes identified as gut microbes, their taxonomy, metabolic independence score, classification as high metabolic independence ('HMI') or not ('non-HMI'), genome length in base pairs, and number of gene calls. (b) Matrix of stepwise completeness of each module in each genome. (c) Matrix of genome detection in each deeply sequenced gut metagenome in the inflammatory bowel disease (IBD), non-IBD, and healthy sample groups. (d) Percent abundance of each genome in each deeply sequenced gut metagenome. (e) Per-genome proportion of samples from each sample group that the genome is detected in using a threshold of 50% (i.e. at least half of the genome sequence is covered by at least one sequencing read in a given sample). (f) Per-sample proportion of detected genomes that are classified as HMI. (g) Average completion of each IBD-enriched module within the HMI genome group and the non-HMI genome group, as well as the difference between these values. (h) Genome-level results when using different HMI score thresholds for determining HMI status. Each threshold is shown both as the average percent completeness required for the 33 IBD-enriched modules and as the HMI score above which a genome is considered HMI. Results for each threshold include the number of genomes assigned as HMI, the percent of genomes assigned as HMI (out of 338), the number and percent of *Bacteroides* genomes assigned as HMI, the mean genome size and mean number of gene calls for both HMI and non-HMI genomes, the p-values and W statistics of the Wilcoxon rank-sum tests comparing detection of HMI vs non-HMI genomes (1) in IBD samples and (2) in healthy samples, and the p-value of the Kruskal-Wallis rank-sum test comparing the fraction of genomes classified as HMI in IBD samples vs healthy/non-IBD samples.

Supplementary file 4. Metagenome classifier information. (a) Details and performance of previously published classifiers for inflammatory bowel disease (IBD) and IBD subtypes. For each classifier, we summarize the cohort details as described by the study; the size of training datasets and validation datasets (if any); the type(s) of samples, data, and extracted features used for classification; the target classes (i.e. what the samples were being classified as); the classifier type and training/validation strategy; and the performance metrics as reported by the study. (b) Classification of each (*Palleja et al., 2018*) metagenome by our logistic regression model trained for distinguishing IBD vs healthy samples on the basis of PPCN data for IBD-enriched modules. This table describes whether the sample was classified as healthy ('HEALTHY') or stressed ('IBD', which we consider to be equivalent to an identification of gut stress), and also whether the sample had low sequencing depth (<25 million reads) or not. (c) Summary of the performance of our metagenome classifier across different training/validation strategies using the IBD and healthy metagenome samples. It also includes the details of our final classifier trained on all 330 samples, though performance data is not available for this model since there were no IBD/healthy samples left for validation – however, see manuscript for its performance on the (*Palleja et al., 2018*) antibiotic time-series dataset. The subsequent sheets include per-fold data and performance information for each train-test strategy: (d) random split cross-validation (25-fold) on PPCN data; (e) leave-two-studies-out cross-validation (24-fold); and (f) (10-fold) cross-validation leaving out samples from the two dominating studies in our dataset (*Le Chatelier et al., 2013*; *Vineis et al., 2016*).

Supplementary file 5. Details of available software for metabolism estimation. For each tool (including the one published in this study), we summarize: the software category (based upon the tool's architecture and mode of use); its metabolism reconstruction strategy (whether it is a pathway prediction tool or a modeling tool or both); the data source(s) it uses for enzyme and metabolic pathway information; how it calculates pathway completeness or generates models (depending on reconstruction strategy); what input and output types it accepts/generates; any additional capabilities as advertised by the tool's publication; whether or not the tool is open-source; the program type; and what language(s) it is developed in (if known). The reference publication and code repository or webpage for each tool is also included.

Supplementary file 6. Data from validation of the per-population copy number (PPCN) approach with simulated metagenomic data. (a) Distribution (mean and standard deviation) of PPCN and PPCN error (computed relative to either average genomic completeness or average genomic copy number) across each validation test case, as well as the proportion of correct, off-by-one, and off-by-two community size estimates in each test case. (b) Spearman's correlation test results between sample parameters (genome size, community size, and diversity level) and important

values computed in our approach (PPCN, PPCN accuracy metrics, and accuracy of community size estimates). Negative values are shown in red and nonsignificant p-values are highlighted in blue. (c) Normalized and rank-ordered relative abundance data from the top 20 most abundant microbial populations in healthy gut metagenomes that we used to recreate a 'typical' relative abundance curve for our simulated metagenomes, based upon data from *Beghini et al., 2021*. Each initial column provides the data from a single sample, and the final four columns describe: the average relative abundance at each rank order; the averages when scaled such that the minimum abundance is 1; the corresponding (integer) coverage values for each scaled average relative abundance value; and the coverage values when increased by 20× for sufficient sequencing depth for assembly.

MDAR checklist

## Data availability

Accession numbers for publicly available data are listed in our Supplementary Tables that are also accessible at https://doi.org/10.6084/m9.figshare.22679080. Anvi'o contigs databases of our assemblies for the 408 deeply-sequenced metagenomes, as well as assemblies of the Palleja et al. 2018 metagenomes are available at can be accessed at https://doi.org/10.5281/zenodo.7897987. Finally the URL https://doi.org/10.5281/zenodo.7883421 gives access to anvi'o contigs databases for the 338 Genome Taxonomy Database (GTDB) reference genomes that represent populations that are prevalent in human gut metagenomes.

The following datasets were generated:

| Author(s) | Year | Dataset title | Dataset URL | Database and Identifier |
|---|---|---|---|---|
| Veseli I, Eren AM | 2024 | Supplementary Tables for Veseli et al. 2023 | https://doi.org/10.6084/m9.figshare.22679080 | Figshare, 10.6084/m9.figshare.22679080 |
| Veseli I | 2023 | Palleja et al. 2018 Metagenome Assemblies for Veseli et al. 2023 | https://doi.org/10.5281/zenodo.7897987 | Zenodo, 10.5281/zenodo.7897987 |
| Veseli I | 2023 | GTDB Genome Contigs DBs for Veseli et al. 2023 | https://doi.org/10.5281/zenodo.7883421 | Zenodo, 10.5281/zenodo.7883421 |

The following previously published datasets were used:

| Author(s) | Year | Dataset title | Dataset URL | Database and Identifier |
|---|---|---|---|---|
| Qin J, Li Y, Cai Z, Li S, Zhu J, Zhang F, Liang S | 2012 | A metagenome-wide association study of gut microbiota in type 2 diabetes | https://www.ncbi.nlm.nih.gov/sra?term=SRA050230 | NCBI Sequence Read Archive, SRA050230 |
| Le Chatelier E, Nielsen T, Qin J, Prifti E, Hildebrand F, Falony G, Almeida M | 2013 | Richness of human gut microbiome correlates with metabolic markers | https://www.ebi.ac.uk/ena/browser/view/PRJEB4336 | EBI European Nucleotide Archive, PRJEB4336 |
| Feng Q, Liang S, Jia H, Stadlmayr A, Tang L, Lan Z, Zhang D | 2015 | Gut microbiome development along the colorectal adenoma-carcinoma sequence | https://www.ebi.ac.uk/ena/browser/view/PRJEB7774 | EBI European Nucleotide Archive, PRJEB7774 |
| Franzosa EA, Sirota-Madi A, Avila-Pacheco J, Fornelos N, Haiser HJ, Reinker S, Vatanen T | 2019 | Gut microbiome structure and metabolic activity in inflammatory bowel disease | https://www.ncbi.nlm.nih.gov/bioproject/PRJNA400072 | NCBI BioProject, PRJNA400072 |
| Lloyd-Price J, Arze C, Ananthakrishnan AN, Schirmer M, Pacheco JA, Poon TW, Andrews E | 2019 | Longitudinal Multi'omics of the Human Microbiome in Inflammatory Bowel Disease | https://www.ncbi.nlm.nih.gov/bioproject/PRJNA398089 | NCBI BioProject, PRJNA398089 |

*Continued on next page*

*Continued*

| Author(s) | Year | Dataset title | Dataset URL | Database and Identifier |
|---|---|---|---|---|
| Qin J, Li Y, Cai Z, Li S, Zhu J, Zhang F, Liang S | 2012 | A metagenome-wide association study of gut microbiota in type 2 diabetes | https://www.ncbi.nlm.nih.gov/sra?term=SRA045646 | NCBI Sequence Read Archive, SRA045646 |
| Quince C, Ijaz UZ, Loman N, Eren AM, Saulnier D, Russell J, Haig SJ | 2015 | Exclusive enteral nutrition modulates the faecal metagenome in paediatric Crohn's disease not by enriching the abundance of presumably 'beneficial' commensals but by suppressing 'dysbiotic' bacteria | https://www.ncbi.nlm.nih.gov/bioproject/PRJEB7576 | NCBI BioProject, PRJEB7576 |
| Raymond F, Ouameur AA, Déraspe M, Iqbal N, Gingras H, Dridi B, Leprohon P | 2016 | The initial state of the human gut microbiome determines its reshaping by antibiotics | https://www.ebi.ac.uk/ena/browser/view/PRJEB8094 | EBI European Nucleotide Archive, PRJEB8094 |
| Schirmer M, Franzosa EA, Lloyd-Price J, McIver LJ, Schwager R, Poon TW, Ananthakrishnan AN | 2018 | Dynamics of metatranscription in the inflammatory bowel disease gut microbiome | https://www.ncbi.nlm.nih.gov/bioproject/PRJNA389280/ | NCBI BioProject, PRJNA389280 |
| University of Sydney | 2016 | metagenome fecal microbiota, Ilumina seq reads of 12 individuals at 2 timepoints | https://www.ncbi.nlm.nih.gov/bioproject/PRJEB6092/ | NCBI BioProject, PRJEB6092 |
| Vineis JH, Ringus DL, Morrison HG, Delmont TO, Dalal S, Raffals LH, Antonopoulos DA | 2016 | Patient-Specific Bacteroides Genome Variants in Pouchitis | https://www.ncbi.nlm.nih.gov/bioproject/PRJNA46881 | NCBI BioProject, PRJNA46881 |
| Wen C, Zheng Z, Shao T, Liu L, Xie Z, Chatelier EL, He Z | 2017 | Quantitative metagenomics reveals unique gut microbiome biomarkers in ankylosing spondylitis | https://www.ncbi.nlm.nih.gov/sra/?term=SRP100575 | NCBI Sequence Read Archive, SRP100575 |
| Qin N, Yang F, Li A, Prifti E, Chen Y, Chatelier EL, Yao J, Wu L, Zhou J, Ni S, Liu L, Pons N, Batto JM, Kennedy SP, Leonard P | 2014 | Alterations of the human gut microbiome in liver cirrhosis | https://www.ebi.ac.uk/ena/browser/view/PRJEB6337 | EBI European Nucleotide Archive, PRJEB6337 |
| Xie H, Guo R, Feng Q, Lan Z, Qin B, Ward KJ, Zhong H | 2016 | Shotgun Metagenomics of 250 Adult Twins Reveals Genetic and Environmental Impacts on the Gut Microbiome | https://www.ebi.ac.uk/ena/browser/view/PRJEB9584 | EBI European Nucleotide Archive, PRJEB9584 |
| Palleja A, Mikkelsen KH, Forslund SK, Kashani A, Allin KH, Nielsen T, Hansen TH | 2018 | Recovery of gut microbiota of healthy adults following antibiotic exposure | https://www.ebi.ac.uk/ena/browser/view/PRJEB20800 | EBI European Nucleotide Archive, PRJEB20800 |
| Rampelli S, Schnorr SL, Consolandi C, Turroni S, Severgnini M, peano C, Brigidi P, Crittenden AN, Henry AG, Candela M | 2015 | Metagenome Sequencing of the Hadza Hunter-Gatherer Gut Microbiota | https://www.ncbi.nlm.nih.gov/bioproject/PRJNA278393 | NCBI BioProject, PRJNA278393 |

*Continued on next page*

*Continued*

| Author(s) | Year | Dataset title | Dataset URL | Database and Identifier |
|---|---|---|---|---|
| Parks DH, Chuvochina M, Rinke C, Mussig AJ, Chaumeil PA, Hugenholtz P | 2022 | GTDB: an ongoing census of bacterial and archaeal diversity through a phylogenetically consistent, rank normalized and complete genome-based taxonomy | https://gtdb.ecogenomic.org/ | GTDB, release95.0 |

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

## Appendix 1

### Low sequencing depth results in poor characterization of community richness in assembled metagenomes

Within our dataset, we observed a correlation between the estimated number of distinct populations in assembled sequences and sequencing depth, i.e., the number of short reads generated from a given sample. In shallow sequencing of metagenomes, short reads may not cover the entirety of population genomes, thus decreasing the rate of recovery of SCGs in the assembly and resulting in an underestimation of the number of populations present. Indeed, the linear relationship between sequencing depth and the number of observed microbial populations we observe in lower depths of sequencing (*Appendix 1—figure 1*) starts to plateau once the sequencing depth exceeds approximately 25 million reads, suggesting that our strategy to estimate the number of distinct microbial populations within these samples serves as a good approximation of the true number of genomes only at relatively higher depths of sequencing. Since an incomplete recovery of population genomes in metagenomic samples also interferes with a meaningful quantification of metabolic potential in a given sample, we set a minimum sequencing depth threshold of 25 million sequencing reads (*Appendix 1—figure 1*). A set of 408 samples (101 IBD, 229 healthy, and 78 non-IBD) from 10 different studies passed our quality threshold to be utilized for further analysis.

Low sequencing depth disproportionately affects IBD metagenomes, thereby reducing our ability to effectively study this disease model in comparison with healthy controls. It also disproportionately affects some studies over others, which could allow cohort or study-specific effects to influence the differential signal between the groups. However, we concluded that the benefits of stringent thresholding outweigh the potential complications arising from imbalanced cohort sizes in our sample subset.

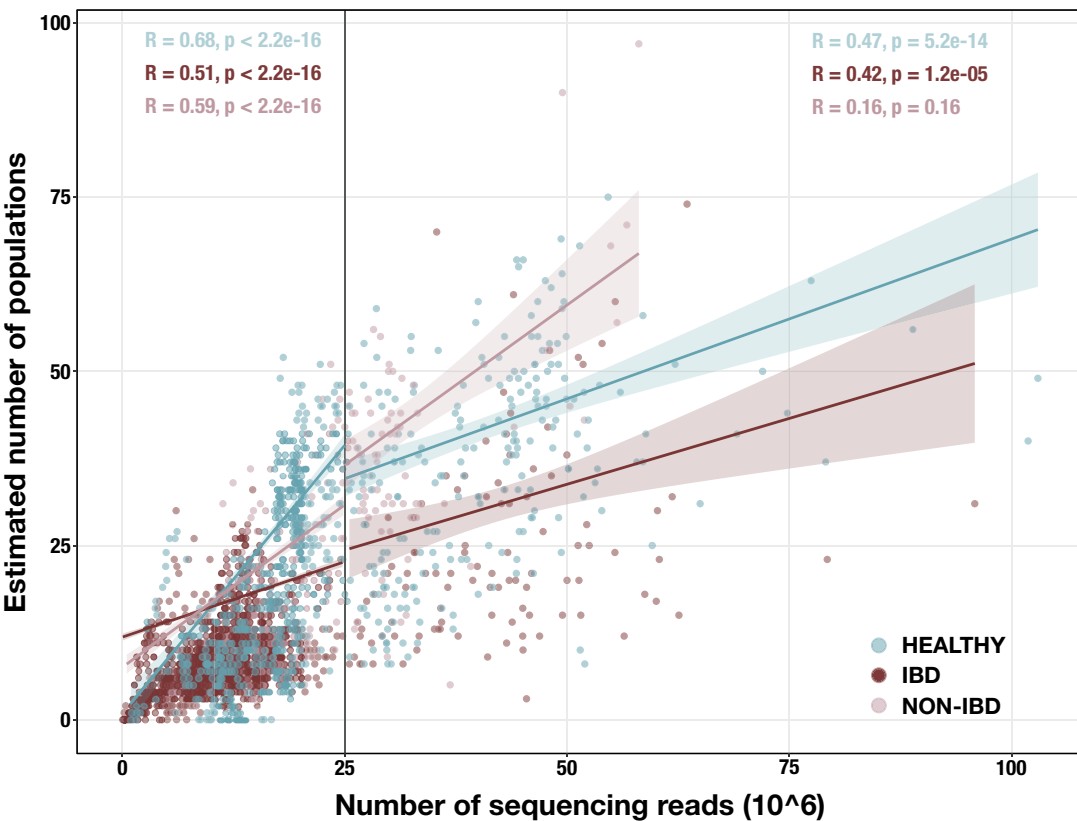

**Appendix 1—figure 1.** Scatterplot of sequencing depth vs estimated number of microbial populations in each of 2893 stool metagenomes. Sequencing depth is represented by the number of R1 reads, except for (*Vineis et al., 2016*) samples, in which case it is the number of merged paired-end reads. The vertical line indicates our sequencing depth threshold of 25 million reads. Per-group Spearman's correlation coefficients and p-values are shown for the subset of samples with depth < 25 million reads (top left) and for the subset with depth ≥ 25 million

reads (top right). Regression lines are shown for each group in each subset, with standard error indicated by the colored background.

## Technical details of metabolism estimation in anvi'o

This section describes technical details of the program 'anvi-estimate-metabolism', which is the main program in the metabolism reconstruction framework in anvi'o (*Appendix 1—figure 2A*). Documentation for this program, including an extended and more up-to-date version of these technical details, can be found at https://anvio.org/m/anvi-estimate-metabolism.

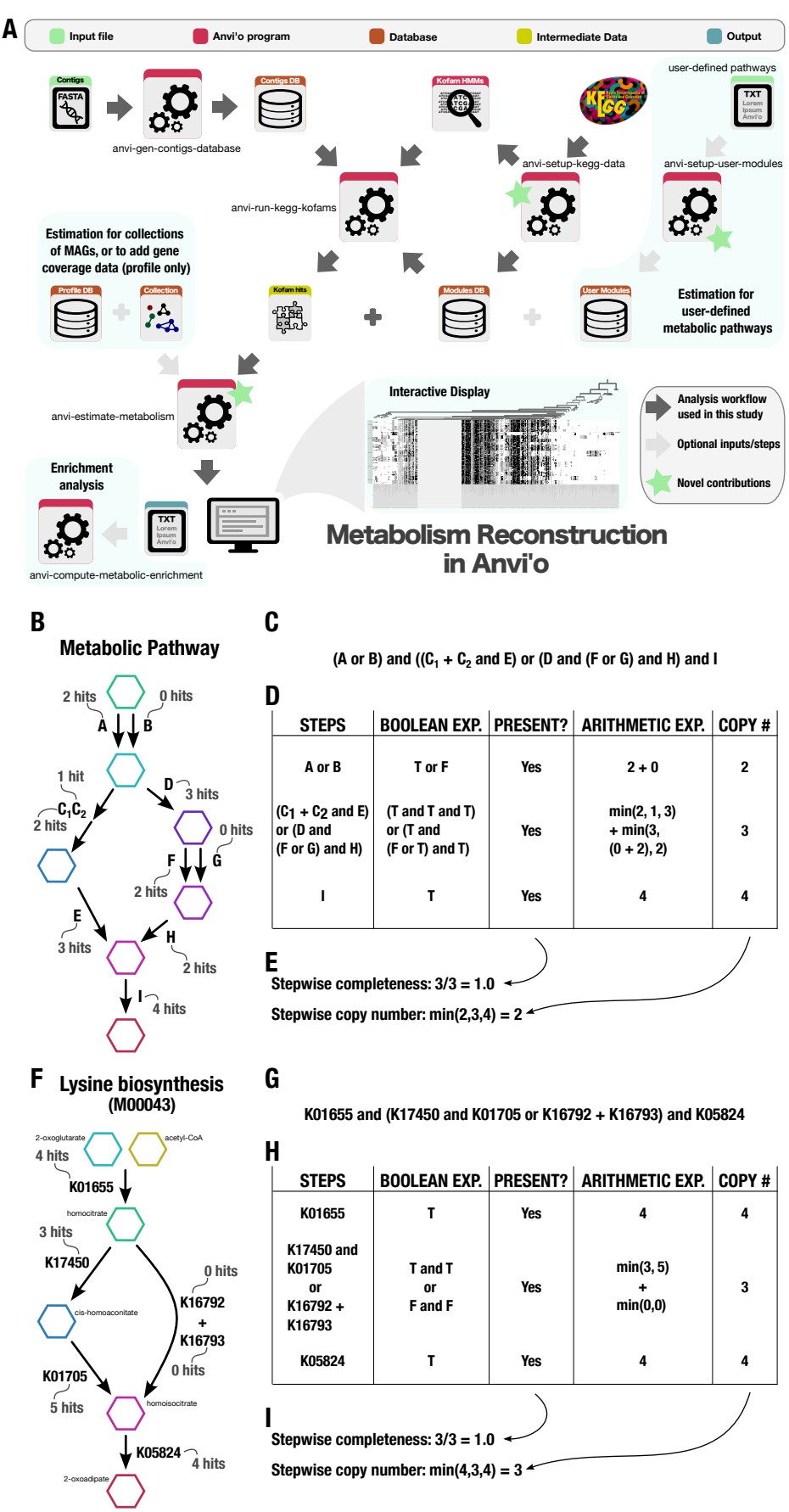

**Appendix 1—figure 2.** Technical details of the metabolism reconstruction software framework in anvi'o. (**A**) Workflow of metabolism reconstruction programs and their inputs/outputs. Dark arrows indicate the primary analysis path utilized in this study. Blue background indicates optional features in the framework. A demonstration of completeness score and copy number calculations for metabolic pathways performed by the program 'anvi-estimate-metabolism' is shown using example enzyme annotation data in panels **B–E** (for a theoretical pathway) and **F–I** (for a real pathway). (**B**) Theoretical metabolic pathway, where hexagons represent metabolites, arrows represent chemical reactions, letters represent enzymes (subscripts indicate enzyme components), and the example number of gene annotation hits for each enzyme is written in gray. (**C**) The definition of the theoretical pathway from panel B, written in terms of the required enzymes. (**D**) Table showing the major steps in the pathway and example calculations for step presence and copy number. Step presence is calculated by evaluating a Boolean expression created from the step definition in which enzymes with > 0 hits are replaced with True (**T**) and the others with False (**F**). Step copy number is calculated by evaluating the corresponding arithmetic expression in which the enzymes are replaced with their annotation counts. (**E**) Final calculations of completeness score (fraction of present steps) and copy number for the theoretical metabolic pathway. (**F–I**) Same as panels **B–E**, but for KEGG module M00043. A high-resolution version of this figure is available at https://doi.org/10.6084/m9.figshare.22851173.

## Summary of program usage

The program 'anvi-estimate-metabolism' predicts the metabolic capabilities of organisms based on their genetic content. It relies upon enzyme annotations and metabolism information from KEGG, specifically using metabolic modules from the KEGG MODULE (*Kanehisa et al., 2023*) database, which are defined in terms of KEGG Orthologs (KOs) that can be annotated via the KOfam database of hidden Markov model (HMM) profiles (*Aramaki et al., 2020*). It can also work with user-defined metabolic pathways, as described in the documentation page https://anvio.org/m/user-modules-data. In our analysis, we used the program 'anvi-run-kegg-kofams' to annotate the KOs used for downstream metabolism estimation; this software implements a heuristic to obtain more accurate and comprehensive annotation results than other contemporary tools for KOfam annotation, as described in *Kananen et al., 2025*.

The program 'anvi-estimate-metabolism' determines which enzymes are annotated in an input sample and uses these functions to compute the completeness and copy number of each metabolic module within the sample. Input samples can be individual genomes, binned or unbinned metagenomes, or ad hoc lists of enzyme accessions. The output of 'anvi-estimate-metabolism' is one or more tabular text files detailing the completeness and copy number scores per module as well as (customizable) information such as pathway metadata; shared/unique enzymes; gene coverage data; and pathway substrates, intermediates, and products. A detailed output description and examples can be found at https://anvio.org/m/kegg-metabolism/.

## Module definitions and interpretation strategies

Metabolic pathways are defined by the enzymes responsible for each reaction in the pathway, using the convention established by the KEGG MODULE database. In these definitions, commas separate alternative enzymes that can catalyze the same reaction, spaces separate subsequent reactions, plus signs indicate essential components of enzyme complexes, minus signs indicate nonessential components of complexes, and parentheses indicate the order of operations. These definitions can also be written in terms of the logical relationships between reactions, such that spaces and plus signs are converted into 'AND' relationships and commas are converted into 'OR' relationships (*Appendix 1—figure 2B–C and F–G*).

'anvi-estimate-metabolism' has two strategies for interpreting module definition strings that treat alternative enzymes and pathway branches differently. One is the 'pathwise' strategy, which considers all possible combinations of enzymes. In this method, each alternative set of enzymes that could be used together to catalyze every reaction in the metabolic pathway is called a 'path' through the module. The program computes completeness and copy number metrics for each path separately, and then identifies the most complete path(s) as the most biologically relevant representative of the module as a whole. Alternatively, with the 'stepwise' strategy the module definition is parsed into high-level 'steps' that each encompasses a set of alternative enzymes for a particular reaction or branch point. The presence and copy numbers of each step are respectively combined into a completeness score and copy number for the entire module.

## Calculation of stepwise completeness and copy number

The analyses in this paper rely on the 'stepwise' metrics of module completeness and copy number, which are calculated as demonstrated in *Appendix 1—figure 2D–E and H–I*. We divide each module into steps by splitting the definition string on the outermost 'AND' relationships (spaces not within parentheses). To determine whether each step is present, we convert the step definition into a Boolean expression in which 'True' represents annotated enzymes and 'False' represents enzymes without annotations. If the Boolean expression evaluates to 'True', then the step is considered present. The module completeness score is the number of present steps divided by the total number of steps. To determine the step copy number, we convert the step definition into an arithmetic expression wherein 'AND' relationships become minimum operations and 'OR' relationships become addition operations. We take the minimum of all per-step copy numbers obtained by evaluating these arithmetic operations to get the overall module copy number.

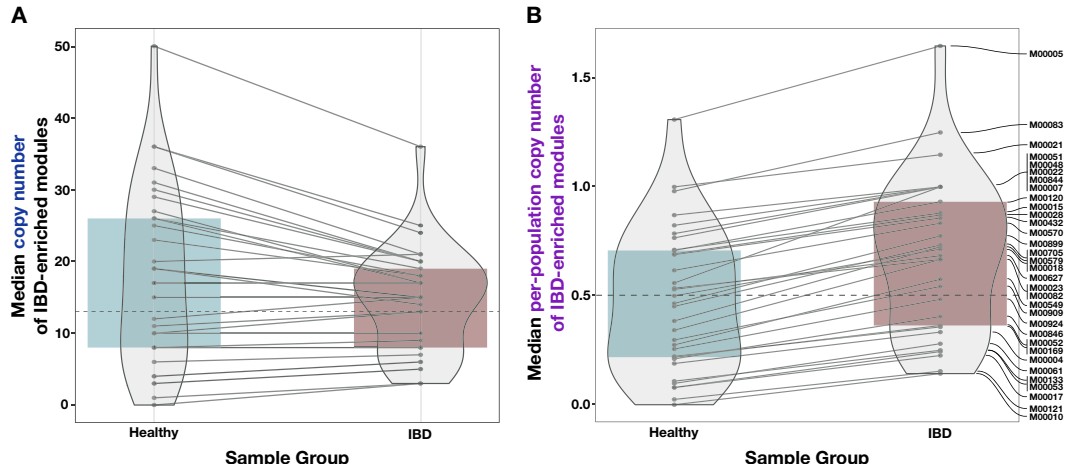

**Appendix 1—figure 3.** Comparison of unnormalized copy number data and normalized per-population copy number (PPCN) data for the inflammatory bowel disease (IBD)-enriched modules. (**A**) Boxplot of median copy numbers for each module in the healthy samples (blue) and IBD samples (red). (**B**) Boxplots of median PPCN for each module in the healthy samples (blue) and IBD samples (red). Lines connect data points for the same module in each plot. The gray dashed line in each plot indicates the overall median value.

## Differential annotation efficiency between IBD and healthy samples

We observed that the proportion of predicted genes with functional annotations was markedly less in healthy metagenomes than in IBD samples, for both sequence homology-based annotation methods (NCBI COGs) and annotation with probabilistic models (KEGG KOfams and Pfams) (*Appendix 1—figure 4*, *Supplementary file 1d*). One possible interpretation of this that aligns with our metabolic competency hypothesis is that the populations with LMI that thrive in the healthy gut environment are relatively less well characterized than the HMI populations that are more likely to survive in the stressful conditions of IBD, resulting in an annotation bias against healthy samples. This interpretation is congruent with our observation that most uncharacterized gut microbial genomes from the GTDB, which have temporary code names in place of taxonomic assignments, were identified as non-HMI (*Figure 3A*).

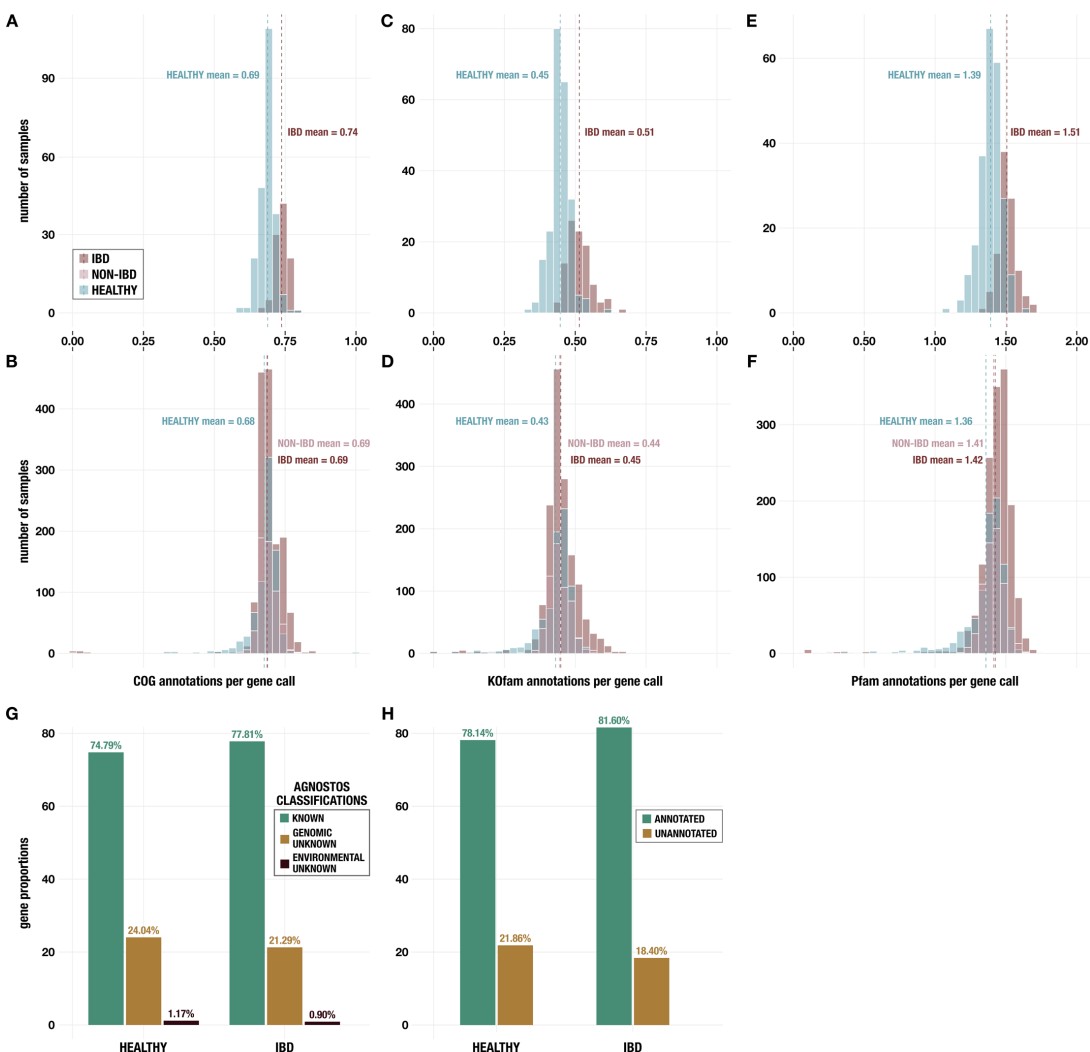

**Appendix 1—figure 4.** Histograms of annotations per gene call from (**A, B**) NCBI Clusters of Orthologous Groups (COGs); (**C, D**) KEGG KOfams; and (**E, F**) Pfams. Panels A, C, and E show data for metagenomes in the subset of 330 deeply sequenced samples from healthy people and people with inflammatory bowel disease (IBD), and panels B, D, and F show data for all 2893 samples including those from non-IBD controls. (**G**) Proportion of genes with each classification from AGNOSTOS (*Vanni et al., 2022*) in the subset of 330 deeply sequenced samples. (**H**) Proportion of genes with at least one annotation from KEGG KOfams (*Aramaki et al., 2020*), NCBI COGs (*Galperin et al., 2015*), or Pfams (*Mistry et al., 2021*) (green) and proportion without any annotation (brown) in the subset of 330 deeply sequenced samples.

The reduced metabolic capacity of LMI microbes and their resulting reliance on robust community interactions (i.e. cross-feeding) may increase their resistance to cultivation that typically aims to isolate individual populations rather than communities. Microbes auxotrophic for key metabolites rely on metabolic interactions with their surrounding community, and current cultivation practices may not sufficiently account for the lack of such interactions. As a result, the available genomes of such populations would be limited to sequences from metagenomic surveys, which are often incomplete and/or composite and are therefore typically not included in efforts to generate models and nonredundant sequence databases for gene annotation (*Aramaki et al., 2020*; *Galperin et al., 2015*; *Sonnhammer et al., 1997*). Thus, the reduced proportion of annotated genes in healthy metagenomes may reflect missing annotations due to lack of sufficiently homologous sequences in state-of-the-art databases. The true reduction in metabolic potential in the healthy sample group may not be as extensive as we have observed in this study.

The discrepancy in annotation efficiency between the healthy and IBD groups disappeared when analyzing all 2893 samples (*Appendix 1—figure 4*). This suggests that the observed annotation bias

does not strongly affect microbial populations that are readily assembled via shallow sequencing – likely, these are populations of high relative abundance in both healthy and IBD samples. Populations of lower abundance, which are less likely to be assembled from shallow metagenomes due to lack of sufficient coverage, are probably also less well characterized as a result. For this to contribute to fewer annotations per gene in healthy samples would necessitate that healthy samples contain relatively more low-abundance populations than IBD samples. Indeed, this is the case: healthy samples contain an average of 86 detected genomes from our set of GTDB gut microbes, and those genomes have a low average percent abundance of 0.61% across these samples. Non-IBD samples are similar, having an average of 77 detected genomes per sample with an average percent abundance of 0.79%. IBD samples, meanwhile, contain 30 detected genomes on average, with a higher average percent abundance of 2.24%. Therefore, the lack of characterization of low-abundance populations may be another factor that contributes to the relative reduction in gene annotations in the healthy samples.

To understand the potential origins of the reduced annotation rate in healthy metagenomes, we ran AGNOSTOS (*Vanni et al., 2022*) to classify known and unknown genes within the healthy and IBD sample groups. AGNOSTOS clusters genes to contextualize them within an extensive reference dataset and then categorizes each gene as 'known' (has homology to genes annotated with Pfam domains of known function), 'genomic unknown' (has homology to genes in genomic reference databases that do not have known functional domains), or 'environmental unknown' (has homology to genes from metagenomes or MAGs that do not have known functional domains). The resulting classifications confirm that healthy metagenomes contain fewer 'known' genes than metagenomes in the IBD sample group – the proportion of 'known' genes classified by AGNOSTOS is about 3.0% less in the healthy metagenomes than in the IBD sample group, which is similar to the ~3.5% decrease in the proportion of 'unannotated' genes observed by simply counting the number of genes with at least one functional annotation (*Appendix 1—figure 4G and H*, *Supplementary file 1e*). Furthermore, the majority of the unannotated genes in either sample group were categorized by AGNOSTOS as 'genomic unknown' (*Appendix 1—figure 4G*), suggesting that the unannotated sequences are genes without biochemically characterized functions currently associated with them and are thus legitimately lacking a functional annotation in our analysis, rather than representing distant homologs of known protein families that we failed to annotate. Based upon the classifications, a systematic technical bias is unlikely driving the annotation discrepancy between the sample groups.

Our observations into the annotation efficiency of microbial genes in different contexts, and our speculations regarding the sources of such bias and its implications warrant further investigations.

## 'Non-IBD' samples are intermediate to IBD and healthy samples

While so far we divided samples into two groups, our dataset also includes individuals who do not suffer from IBD, yet are not healthy either. A recent study using flux balance analysis to model metabolite secretion potential in the dysbiotic, non-dysbiotic, and control gut communities of Crohn's disease patients has shown that several predicted microbial metabolic activities align with gradients of host health (*Heinken et al., 2021*). To test whether the HMI signal captures gradients in host health, we included the 'non-IBD' group of patients that suffer from GI conditions other than IBD in our analysis. The set of 78 samples classified as 'non-IBD' indeed represent an intermediate group between healthy individuals and those diagnosed with IBD (*Appendix 1—figure 5B*). While the HMI signal was reduced in 'non-IBD' patients, 75% of the pathways enriched in IBD patients were also enriched in the 'non-IBD' group compared to healthy individuals. Similarly, when sorting each individual cohort along a health gradient based on cohort descriptions in their respective studies (see 'Characterizing cohort-specific metabolic capacity across the gradient of health and disease'), the relative proportion of metabolic pathways indicative of HMI increased as a function of increasing disease severity (*Appendix 1—figure 6A*). These findings suggest that the HMI signal is sufficiently sensitive to resolve gradients in host health and could serve as a diagnostic tool to monitor changing stress levels in a single individual over time.

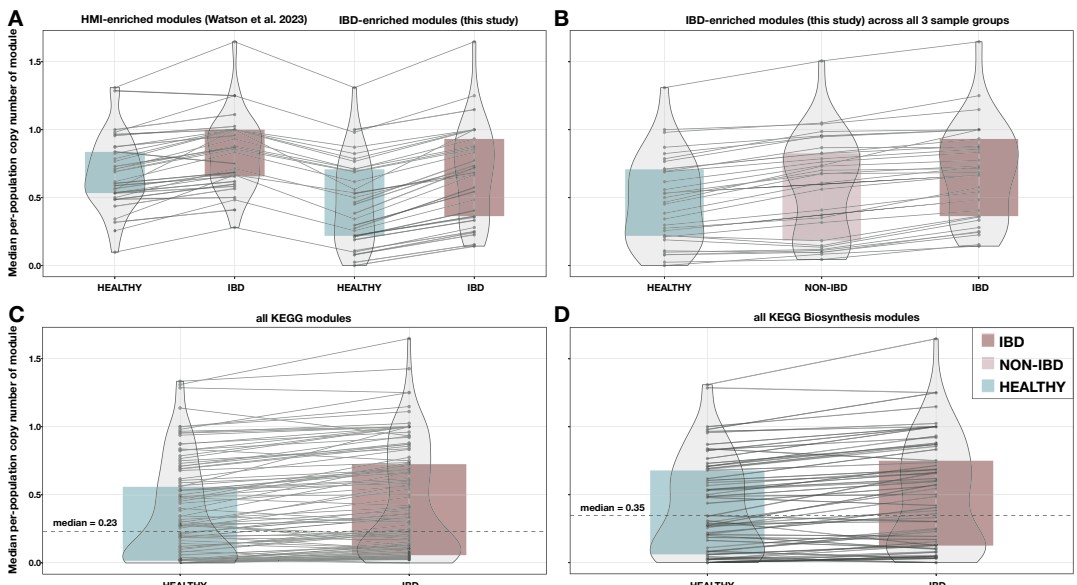

**Appendix 1—figure 5.** Additional boxplots of median per-population copy number for various subsets of metabolic pathways and metagenome samples. (**A**) 33 modules enriched in high metabolic independence (HMI) populations from *Watson et al., 2023*, compared to the 33 inflammatory bowel disease (IBD)-enriched modules from this study, with medians computed in the set of deeply sequenced healthy (n = 229) and IBD (n = 101) samples. (**B**) The 33 IBD-enriched modules from this study, with medians computed in the set of deeply sequenced healthy (n = 229), non-IBD (n = 78), and IBD (n = 101) samples. (**C**) All KEGG modules (n = 117) with nonzero copy number in at least one sample, with medians computed in the set of deeply sequenced healthy (n = 229) and IBD (n = 101) samples. (**D**) All biosynthesis modules (n = 88) from the KEGG MODULE database, with medians computed in the set of deeply sequenced healthy (n = 229) and IBD (n = 101) samples. Where applicable, dashed lines indicate the overall median for all modules, and solid lines connect the points for the same module in each sample group. The IBD sample group is highlighted in red, the NON-IBD group in pink, and the HEALTHY group in blue.

## Module enrichment without consideration of effect size leads to nonspecific results

Our analyses indicate that the majority of KEGG modules had higher PPCN in IBD metagenomes (*Figure 2E*, *Supplementary file 2b*). Indeed, when we examine all modules with nonzero median PPCN in at least one group of samples (n = 117), their median normalized copy number is systematically higher in the IBD group than in the healthy group (*Appendix 1—figure 5C*; 98 out of 117 modules have a higher normalized copy number in the IBD group than in the healthy group at 5% FDR-adjusted significance level using a one-sided Wilcoxon test). This result is likely a natural outcome of the differential distribution of HMI and non-HMI genomes in the two sample groups, as seen in our analysis of reference genomes (*Figure 3B and C*), where the overrepresentation of HMI populations with larger genomes that encode many more complete pathways in IBD samples leads to higher PPCNs computed at the metagenome level. The consistent elevation in PPCN of metabolic modules in IBD could also be attributed, at least in part, to the aforementioned functional annotation biases that seem to disproportionally affect the characterization of healthy metagenomes. The lower annotation efficiency in healthy metagenomes could result in partial copies of pathways, which are ignored by our stringent copy number calculation that only counts complete copies. Therefore, to narrow down our results and identify which pathways are *particularly* important for microbial resilience in the IBD gut environment, we considered only those pathways with the largest difference in normalized copy number ('effect size') between the two groups to identify metabolic modules that are truly elevated in IBD metagenomes (see Methods).

With similar considerations we also investigated whether biosynthetic capacity in general was enriched in IBD samples. For this, we expanded our analysis to also consider biosynthesis pathways that did not meet the enrichment criteria we have used for inclusion in the final set of 33 IBD-enriched modules. As expected, we found that the majority of all biosynthesis pathways in the KEGG

MODULE database (n = 88) have significantly higher normalized copy numbers in IBD samples (*Appendix 1—figure 5D*; at a 5% FDR-adjusted significance level, 62 out of 88 (70%) biosynthesis pathways have a higher normalized copy number using a one-sided Wilcoxon test). This analysis also showed a similar increase for non-biosynthetic pathways: 63 out of 91 (69%) non-biosynthetic pathways showed significant increase in IBD samples (two-sample test for equality of proportion: 0.88). Overall, these data indicate that without the consideration of effect size, both biosynthetic and non-biosynthetic capacity appear to be increased in the IBD gut microbiome. In contrast, maintenance of a higher metabolic capacity for the biosynthesis of essential nutrients emerges as an important factor for microbial resilience in IBD through a strict enrichment criteria in addition to statistical significance scores calculated for differential abundance.

## A review of HMI-associated modules in the context of gut microbiome literature

The 33 pathways that are enriched in microbial communities associated with individuals with IBD likely provide competencies that are critical for survival in the stressed gut environment. In this section, we offer a review of IBD-enriched modules with existing gut microbiome scientific literature.

### Amino acid pathways

Of the eight proteinogenic amino acids that can be synthesized with IBD-enriched pathways, *leucine*, *tryptophan*, *threonine*, *isoleucine*, and *methionine* are essential amino acids for humans (*Lopez and Mohiuddin, 2023*), while *cysteine* and *arginine* are semi-essential (*Rehman et al., 2020*; *Tong and Barbul, 2004*). Furthermore, leucine, tryptophan, isoleucine, and cysteine can reduce oxidative stress for intestinal epithelial cells (*Katayama and Mine, 2007*), which may be beneficial given the increased GI oxygen levels associated with IBD (*Rigottier-Gois, 2013*). Several of these amino acids have been analyzed for their potential therapeutic effects in IBD (*Liu et al., 2017*). Regardless, it is unknown if the depleted gut microbiome in IBD would produce these amino acids in sufficient quantity to promote health benefits to the host, especially considering that the microbes themselves require these molecules for protein production and as energy sources – for instance, in proteolytic fermentation (*Wu et al., 2021*; *Lin et al., 2017*).

The *Shikimate pathway*, which converts phosphoenolpyruvate and erythrose 4-phosphate (E4P) to chorismate, is a prerequisite for *tryptophan biosynthesis*. This pathway is only present in microorganisms and plants, and it also produces intermediates for other metabolic pathways such as quinate degradation and antibiotic synthesis (*Herrmann and Weaver, 1999*). A recent analysis of paired fecal metagenomes and metatranscriptomes from the HMP using reference genome-based functional inference demonstrated that the Shikimate pathway is typically incomplete in gut microbes and only transcriptionally active in a few, suggesting that most gut microbes are auxotrophic for aromatic amino acids and therefore rely on dietary sources and potentially cross-feeding to obtain these molecules or their precursors (*Mesnage and Antoniou, 2020*). This offers a potential explanation for the enrichment of the Shikimate and tryptophan biosynthesis pathways in the IBD gut microbiome, where a depleted community may restrict the availability of cross-fed metabolites. Indeed, a tryptophan-deficient diet alters the composition of the gut microbiota in aged mice (*Yusufu et al., 2021*), providing auxiliary evidence that the loss of this amino acid impacts microbial survival. Furthermore, the serum levels of tryptophan are reduced in individuals with IBD due to high host metabolism rates (*Nikolaus et al., 2017*), which may exacerbate the lack of bioavailable tryptophan for gut microbes. On the host side, tryptophan and its derivatives influence a number of physiological processes, though it is unclear how much the microbial production of tryptophan contributes to these effects (*Agus et al., 2018*). Nevertheless, the lack of tryptophan appears to worsen intestinal inflammation while supplementation can attenuate it (*Kim et al., 2010*; *Hashimoto et al., 2012*).

*Cysteine biosynthesis* was previously found to be enriched in the IBD gut microbiome based on reference genome analysis of 16S ribosomal RNA gene amplicons (*Morgan et al., 2012*), in which the authors propose that cysteine metabolism could be important to microbial management of oxidative stress via the production of glutathione, which is protective against reactive oxygen species (*Sherrill and Fahey, 1998*; *Tepe et al., 2006*), from cysteine and glutamate. Cysteine can also be converted into hydrogen sulfide ($H_2S$) by host colonocytes and some intestinal microbes. Though $H_2S$ produced by colonocytes can help support their energy production, excess microbially

derived H$_2$S in the lumen is a risk factor for gut mucosal inflammation and H$_2$S may play a role in colorectal carcinogenesis (*Blachier et al., 2019*). Interestingly, cysteine biosynthesis is also enriched in the gut microbiomes of postmenopausal women, where it is thought to contribute to elevated homocysteine levels and therefore to increased risk of cardiovascular disease (*Zhao et al., 2019*).

*Leucine* and *isoleucine*, as branched-chain amino acids (BCAAs), are important nutrients and signaling molecules in humans (*Gojda and Cahova, 2021*). Gut microbial synthesis of these compounds does contribute to human BCAA pools, as evidenced by experiments with heavy isotope labeling and correlations between serum and fecal BCAA levels (*Metges et al., 1999*; *Dhakan et al., 2019*). The extent of this exchange has not been characterized in individuals with IBD. However, a study of individuals receiving anti-integrin therapy for Crohn's disease demonstrated that pathways for biosynthesis of L-isoleucine and arginine were enriched at baseline in the gut microbiomes of responders to the therapy (*Ananthakrishnan et al., 2017*). In a longitudinal study of mice, biosynthesis pathways for leucine and proline were more abundant in animals modeling IBD (*Sharpton et al., 2017*).

*Threonine* and *proline* are both important components of intestinal mucins (*Johansson and Hansson, 2016*; *Faure et al., 2005*) and thus contribute to mucosal barrier integrity, which is typically impaired in IBD (*Johansson et al., 2010*). For instance, threonine, proline, and cysteine supplementation has been shown to reduce symptoms and restore lactobacilli and bifidobacteria counts in rats with DSS-induced inflammation (*Sprong et al., 2010*; *Faure et al., 2006*). The latter observation suggests the importance of an external source of these three amino acids to the fitness of the lactobacilli and bifidobacterial populations and thereby supports the idea that they are community metabolites.

We also found *methionine* biosynthesis to be enriched in the IBD gut microbiome. In individuals with quiescent IBD, reduced serum levels of methionine, proline, and tryptophan are correlated with changes in the gut microbiome that are associated with increased symptoms of fatigue (*Borren et al., 2021*), demonstrating a putative link between methionine bioavailability, microbial abundances, and host wellbeing. Indeed, L-methionine supplementation in piglets results in improved mucosal integrity and villus architecture (*Chen et al., 2014*), and the activated form of methionine, *S*-adenosylmethionine, can reverse colon lesions and cytoskeletal damage in intestinal cells in DSS-treated mice (*Oz et al., 2005*). Yet, reducing methionine in high-fat diets given to mice was shown to improve intestinal barrier function, reduce inflammation, and increase the abundance of short-chain fatty acid (SCFA)-producing microbes (*Yang et al., 2019*), so the net impact of methionine on host health and microbial fitness remains unclear.

*Arginine* has been well studied in the context of IBD. It has been shown to reduce cytokine production, promote intestinal healing, and improve intestinal barrier function in DSS-treated mice, perhaps by enhancing production of nitric oxide (NO) (*Coburn et al., 2012*; *Gobert et al., 2004*; *Singh et al., 2019*). NO is a free radical that has been implicated in regulating mucosal barrier integrity, GI motility, and protection against oxidative stress, though overproduction of this compound can have detrimental effects (*Kolios et al., 2004*; *Walker et al., 2018*). Biosynthesis of *ornithine*, which is both a precursor and a derivative of arginine, was enriched in the IBD gut microbiome as well in agreement with another study that reported an increase in ornithine biosynthesis in the gut microbiome of individuals with active UC (*Hellmann et al., 2023*). Finally, *polyamines* – which are derived from arginine and were also represented in the enriched pathways – promote intestinal barrier function *Liu et al., 2009*; for instance, by regulating the growth of intestinal epithelial cells (*McCormack and Johnson, 1991*).

## Carbohydrate pathways

Three KEGG modules describing the *pentose phosphate pathway* (PPP) were enriched in IBD samples – the entire pentose phosphate cycle (M00004), the oxidative phase (M00006), and the non-oxidative phase (M00007). We removed the oxidative phase (M00006) from our set of IBD-enriched modules because it was an exact copy of the initial steps in M00004; however, we kept the non-oxidative phase (M00007) in our set because it is defined using slightly different enzymes than the non-oxidative portion of M00004. M00007 is defined in four steps and utilizes a ribulose-phosphate 3-epimerase and a ribose 5-phosphate isomerase in the last two steps, while the non-oxidative phase in M00004 is defined in three steps and utilizes a glucose-6-phosphate isomerase in the last step. The PPP is a ubiquitous pathway in most bacteria and eukaryotes, as it plays a central

role in cellular metabolism. It produces the important cellular intermediates ribose 5-phosphate and E4P, which are used for synthesis of nucleotides and aromatic amino acids, respectively (*Soderberg, 2005*). In fact, E4P is one of the inputs to the Shikimate pathway, another IBD-enriched module discussed above. The PPP also produces NADPH, a reducing equivalent important for reductive reactions and prevention of oxidative stress (*Kruger and von Schaewen, 2003*; *Christodoulou et al., 2018*). Beyond its link to other enriched amino acid biosynthesis pathways, it is unusual that such a central pathway would have an increased copy number in the IBD gut microbiome rather than being equally distributed across all samples. Some gut microbes are known to lack the transaldolase gene in this pathway and may instead encode an alternative pathway for pentose degradation called the sedoheptulose 1,7-bisphosphate pathway (SBPP) (*Garschagen et al., 2021*); it is therefore possible that the enrichment of the more common PPP in IBD is related to an increased ratio of microbial populations that use the PPP rather than the SBPP in the less-diverse microbiome of IBD patients, though this requires further investigation to verify.

The first carbon oxidation of the *citric acid cycle* (TCA cycle), which is a three-step conversion from oxaloacetate to 2-oxoglutarate (alpha-ketoglutarate), is enriched in the IBD samples. Similar to the PPP, the citric acid cycle is a central metabolic pathway, especially with regard to generation of energy and key metabolites for other pathways (*Akram, 2014*). It is unclear why only this particular portion of the cycle would be enriched, though this could perhaps be attributed to the role of alpha-ketoglutarate in the production of glutamate, the precursor to proline, ornithine, and arginine (three amino acids with enriched biosynthesis pathways in the IBD sample group, as discussed above). It has been said that 2-oxoglutarate is the most fundamental compound of this cycle, serving as the link between carbon and nitrogen metabolism and also as a critical element in the recovery of amine groups for amino acid and protein production (*Pierzynowski and Pierzynowska, 2022*; *Huergo and Dixon, 2015*). Thus, the enrichment of 2-oxoglutarate production capacity in the IBD gut environment could be related to the enrichment of amino acid biosynthesis pathways.

Two nucleotide sugar biosynthesis pathways are enriched in the IBD gut microbiome. One of these is *synthesis of UDP-glucose*, which is an important molecule implicated in a variety of key cellular metabolisms. It is an intermediate in polysaccharide biosynthesis and pyrimidine metabolism, a precursor of lipopolysaccharides in the outer cell membrane of Gram-negative bacteria, and an extracellular signaling molecule (*Ralevic, 2015*). Additionally, as an agonist for P2Y-14 receptors, it could play a role in modulating host GI functions like muscular contraction (*Bassil et al., 2009*), and in modulating host inflammatory responses by activating this receptor specifically in T-lymphocytes (*Scrivens and Dickenson, 2005*) and in immature monocyte-derived dendritic cells (*Skelton et al., 2003*). The other enriched nucleotide sugar pathway is *UDP-GlcNAc biosynthesis*. Flux through this pathway is linked to a multitude of other central metabolisms, including amino acid and fatty acid metabolism (*Hardivillé and Hart, 2014*). Furthermore, UDP-GlcNAc is an important substrate in protein glycosylation pathways (*Hardivillé and Hart, 2014*; *Ryczko et al., 2016*), and a precursor to critical cell wall components in bacteria (*Liu and Breukink, 2016*; *Mikkola, 2020*; *van Dam et al., 2009*). In the gut, this molecule has been implicated in regulation of nutrient uptake by the host (*Ryczko et al., 2016*).

*D-Glucuronate (glucuronic acid) degradation* into pyruvate and D-glyceraldehyde 3-phosphate is also enriched in the IBD gut microbiome. Some gut microbes are capable of growth on host-derived uronic acids (*Lopez-Siles et al., 2012*), so this pathway may serve as a source of energy to microbes living in the IBD gut environment. In mice, there is evidence that derivatives of glycosaminoglycan degradation such as D-glucuronate can worsen colitis (*Lee et al., 2009*).

Finally, the *phosphoribosyl diphosphate (PRPP) biosynthesis pathway* is important because PRPP is used in the formation of glycosidic bonds as well as in the biosynthesis of a number of cofactors, amino acids, and nucleotides (*Hove-Jensen et al., 2017*). It is discussed further below in the context of nucleotide metabolism.

## Cofactor and vitamin pathways

Biosynthesis or salvage pathways for the following five cofactors and vitamins are enriched in IBD: *heme*, *siroheme*, *thiamine (vitamin B1)*, *cobalamin (vitamin B12)*, and *coenzyme A (CoA)*. Heme is required for aerobic respiration (*Gruss et al., 2012*) and the increase in this pathway may be related to elevated oxygen levels in the gut as a result of inflammation, which promotes the growth of aerotolerant microbes (*Shah, 2016*; *Cevallos et al., 2019*). Dietary heme has also been associated with gut dysbiosis, aggravated colitis, and increased cytotoxicity in the colon (*Constante et al.,*

*2017*; *Ijssennagger et al., 2015*); and genes related to heme and siroheme biosynthesis have also been found with high abundance in infants with neonatal necrotizing enterocolitis (*Claud et al., 2013*).

Both *thiamine* and *cobalamin* are important cofactors that are commonly shared between gut microbes (*Magnúsdóttir et al., 2015*), suggesting that microbes incapable of synthesizing these cofactors are unlikely to thrive in the low-diversity microbial communities of the IBD gut environment. Neither of these vitamins is produced by host cells but they are typically acquired from dietary sources (cobalamin, in particular, is absorbed in the small intestine) (*Seetharam and Alpers, 1982*; *Degnan et al., 2014b*; *Hossain et al., 2022*), so the enrichment of these pathways is unlikely to have a large impact on host health.

*Coenzyme A* can be produced from pantothenate (vitamin B5) by most gut microbes (*Magnúsdóttir et al., 2015*) and its biosynthesis has been described as 'essential' considering that CoA is required for a large number of enzymatic reactions (*Spry et al., 2008*; *Leonardi et al., 2005*). It is therefore interesting that this pathway appears to be enriched in the IBD gut microbiome, which implies a relative deficiency of CoA biosynthesis in the healthy gut microbiome. It is possible that the module is spuriously enriched, despite its low p-value of 5.7e-21, given the short length of this pathway – it has three major steps when the KEGG module definition is interpreted in a 'stepwise' fashion by anvi-estimate-metabolism, though there are in fact five chemical conversions (*Supplementary file 2a*). An alternative possibility is that the KO HMMs for the required enzymes do not sufficiently represent the diversity of these proteins across the gut microbiota, which could cause this pathway to be undercounted due to lack of proper annotations.

## Nucleotide pathways

The IBD-enriched modules include pathways for synthesis of the first complete *purine*, *inosine monophosphate*, as well as a series of *pyrimidine biosynthesis pathways* encoding the conversion from uridine monophosphate to ribonucleotides (UDP/UTP, CDP/CTP) and finally to the cytosine deoxyribonucleotide (dCTP). The *PRPP biosynthesis pathway* is also included in this list; though it is classified as central carbohydrate metabolism in KEGG due to its role in glycosidic bond formation, this molecule is an important precursor for nucleotide biosynthesis (both purines and pyrimidines) and synthesis of the amino acids tryptophan and histidine (*Hove-Jensen et al., 2017*). Though many microbes are capable of producing their own nucleotides, some – especially lactic acid bacteria – are not and rely on uptake of exogenous nucleosides and bases, which are converted to nucleotides via salvage pathways (*Nygaard, 2014*; *Kilstrup et al., 2005*). Notably, these salvage pathways are not enriched in the IBD gut microbiome, suggesting that self-sufficiency in nucleotide biosynthesis (especially in the early stages in this process) is selected for in these communities. This also implies the importance of pyrimidine and purine cross-feeding in the healthy gut environment, which is supported by evidence that some gut microbes (e.g. *Bacteroides vulgatus*) actively secrete nucleosides in the colon (*Wong et al., 2023*; *Teng et al., 2023*).

## Lipid pathways

Two lipid biosynthesis pathways – *initiation and elongation of fatty acids* – are enriched in IBD. Fatty acids are essential components of cell membranes and also serve as signaling molecules (*Brown et al., 2023*); thus, the ability to synthesize them is an important fitness determinant. For example, gut *Bacteroides* species that are deficient in sphingolipid production capabilities are much less resilient to oxidative stress than wild-type species (*An et al., 2011*). Since oxidative stress is a hallmark of IBD, it is possible that this environment selects for microbes capable of fatty acid biosynthesis.

## Energy pathways

The *Pta-Ack pathway* is important for microbial energy production and adaptation to different growth conditions via the 'acetate switch', which enables either production or consumption of acetate depending on available nutrients (*Wolfe, 2005*). SCFAs such as acetate serve as important energy sources to intestinal epithelial cells. They also play a role in regulating gut barrier function and host immune responses (*Martin-Gallausiaux et al., 2021*; *Zhang et al., 2022*), and impaired absorption and oxidation of SCFAs can contribute to the development of IBD (*Zhang et al., 2022*). Acetate promotes host intestinal IgA production and thereby has a protective effect against gut inflammation (*Wu et al., 2017*), but acetate levels are reduced in children with IBD (*Treem et al.,*

*1994*). Further study is required to determine the flux direction of the Pta-Ack pathway and whether it contributes to the reduction of acetate in the IBD gut environment.

*CAM metabolism* is categorized as a carbon fixation pathway in the KEGG MODULE database, yet is a short (two-step) pathway utilizing enzymes required in other common metabolisms. Its first step is catalyzed by PEP carboxylase, an enzyme that is involved in gluconeogenesis, serine biosynthesis, and carbon skeleton conversions in the citric acid cycle (*Yang et al., 2009*). Its second step is catalyzed by malate dehydrogenases, a ubiquitous class of enzymes that convert 2-hydroxy acids to 2-keto acids and are involved in gluconeogenesis, the TCA cycle, glyoxylate bypass, and amino acid synthesis (*Minárik et al., 2002*; *Musrati et al., 1998*). The increase in this pathway in IBD gut microbiomes could be attributed in part to the increase in aerobic respiration due to elevated oxygen levels (*Shah, 2016*; *Cevallos et al., 2019*) and in part to the increase in amino acid biosynthesis capacity as evidenced by the multiple amino acid pathways that are also enriched.

## Drug resistance pathways

The use of antibiotics to treat IBD and its complications is known to increase antibiotic resistance in the gut microbiome (*Nitzan et al., 2016*; *Ledder, 2019*) and several studies have noted that individuals exposed to antibiotics are more likely to develop IBD (*Kronman et al., 2012*; *Ungaro et al., 2014*; *Ledder, 2019*; *Shaw et al., 2011*). This potentially explains the enrichment of two drug resistance pathways in the IBD microbiome: *efflux pump MepA* (conferring multidrug resistance) and the *bla system* (conferring beta-lactam resistance), as higher rates of antibiotic exposure in this sample group naturally leads to selection for resistance phenotypes (*Levy, 2000*; *Alekshun and Levy, 2007*). Beta-lactamases in particular have been found with higher frequency in people with IBD (*Vich Vila et al., 2018*; *Leung et al., 2012*; *Vaisman et al., 2013*). Increased microbial drug resistance can heighten the risk of a severe infection such as *Clostridium difficile* infection (CDI) (*Llor and Bjerrum, 2014*). CDI already occurs with higher frequency in individuals with IBD (*Jodorkovsky et al., 2010*), though the higher incidence of CDI is not necessarily linked to chronic antibiotic use in these individuals (at least in one retrospective study of Crohn's disease) (*Roy and Lichtiger, 2016*). Regardless, antibiotic resistance is a global health problem that affects everyone, not just those with IBD.

## Most enriched pathways in HMI reference genomes

The three HMI-associated pathways with the largest difference in average completion (>40%) between HMI and non-HMI reference genomes were *siroheme biosynthesis*, *cobalamin biosynthesis*, and *tryptophan biosynthesis* (*Supplementary file 3g*). Siroheme and cobalamin biosynthesis represent complex pathways that require 6–8 and 11–13 enzymatic steps, respectively, and both compounds belong to the tetrapyrroles that are involved in various essential biological functions (*Bryant et al., 2020*). Siroheme is a cofactor required for nitrite and sulfite reduction and its biosynthetic pathway provides the precursors required for cobalamin biosynthesis. Genes belonging to biosynthetic pathways of siroheme and cobalamin had higher average relative abundance in infants diagnosed with neonatal necrotizing enterocolitis (*Claud et al., 2013*), an inflammatory bowel condition affecting premature newborns. The siroheme biosynthesis pathway is upregulated in some human pathogens in response to high NO levels likely in relation to the NO detoxification function of nitrite reductase (*Porrini et al., 2021*). Increased NO levels are commonly associated with active inflammation in IBD (*Soufli et al., 2016*).

While siroheme is central to sulfite and nitrite reduction in prokaryotes, *cobalamin* (vitamin B12) is essential not only for the majority of gut microbes (~80%) (*Kelly et al., 2019*; *Hossain et al., 2022*; *Degnan et al., 2014a*) but also for the human host, and functions as a coenzyme in key metabolic pathways in humans and bacteria. However, only relatively few gut microbes (~20–40%) encode the metabolic pathway for its synthesis (*Degnan et al., 2014b*; *Magnúsdóttir et al., 2015*; *Kelly et al., 2019*) and humans largely rely on cobalamin supplied via their diet. B12 deficiency in humans leads to reduced villi length (*Berg et al., 1972*) and may affect intestinal barrier functioning (*Bressenot et al., 2013*). However, microbially produced cobalamin alone is insufficient to sustain the host's requirements (*Magnúsdóttir et al., 2015*). The high average completion of this complex pathway in reference genomes classified as HMI (86%) in contrast to non-HMI reference genomes (40%) demonstrates the importance of metabolic independence for the survival of microorganisms in stressed gut environments, whereas in a healthy gut environment cross-feeding of B-vitamins

likely supports those microbes that do not have metabolic means to synthesize them (*Magnúsdóttir et al., 2015*).

*Tryptophan* is an essential amino acid that serves as a precursor for a variety of microbial (*Alkhalaf and Ryan, 2015*) and human metabolites that play a potential role in IBD pathogenesis (*Agus et al., 2018*). Tryptophan metabolites mediate a variety of host microbe interactions in the human gut (*Agus et al., 2018*), contribute to gut barrier integrity, and exert anti-inflammatory functions (*Bansal et al., 2010*; *Roager and Licht, 2018*). While fecal tryptophan concentrations can be elevated in IBD patients (*Jansson et al., 2009*), tryptophan host metabolism via the Kynurenine pathway also appears to be elevated in disease, resulting in decreased serum levels of the amino acid (*Nikolaus et al., 2017*). At the same time, a tryptophan-deficient diet in mice is linked to intestinal inflammation and alterations of the microbial community composition (*Hashimoto et al., 2012*; *Yusufu et al., 2021*).

Overall, our data contains no evidence or indication for any direct links between the increased representation of microbial metabolic modules in IBD and the role of the products these metabolic activities yield in human disease states.

## Characterizing cohort-specific metabolic capacity across the gradient of health and disease

We sought to evaluate the cohort-specific trends in metabolic capacity by computing the median PPCN of the 33 IBD-enriched modules within each sample from each study. Considering the heterogeneity within each sample group, we ordered the studies from most healthy to least healthy, using the cohort description from each publication to approximate relative healthiness based on the number and types of exclusions listed for healthy or non-IBD controls, or on the diagnostic criteria for people with IBD (*Supplementary file 1a*).

The amount of detail provided as well as the GI conditions considered varied between studies. To overcome this challenge, we placed more emphasis on exclusionary conditions to sort samples based on host health status. For instance, *Le Chatelier et al., 2013* and *Raymond et al., 2016* used the most stringent criteria in the selection of healthy individuals. Both studies excluded patients with GI-related conditions like disease, surgery, and medication; medications affecting the immune system; or antibiotics. Additionally, *Le Chatelier et al., 2013* also excluded individuals diagnosed with type 2 diabetes while *Raymond et al., 2016* did not, and we assigned a higher 'health score' to samples classified as healthy by *Le Chatelier et al., 2013*. Similarly, *Schirmer et al., 2018* applied more stringent exclusion criteria for samples classified as 'non-IBD' controls than *Franzosa et al., 2019*, and was therefore considered a healthier cohort within that group.

To arrange samples classified as 'IBD' along a gradient of host health status, we considered similar cohorts to be more unhealthy if their diagnosis was supported by several lines of evidence. For example, *Schirmer et al., 2018* diagnosed IBD based on a screening colonoscopy and included existing patients with consistent diagnosis over the past 5+ years, *Lloyd-Price et al., 2019* required a combination of endoscopic and histopathologic evidence for diagnosis, and *Franzosa et al., 2019* only considered patients diagnosed via endoscopic, histopathologic, and radiographic approaches. Regardless, these cohorts are likely extremely similar in healthiness. Patients with the lowest health score were described by *Vineis et al., 2016* – with a cohort composed of total proctocolectomy patients with ileal pouches, some of which developed pouchitis.

Ordering the per-sample median PPCN values along this gradient of cohort health indicates that the HMI metric for gut microbial metabolic capacity increases as host health decreases (*Appendix 1— figure 6A*). Therefore, HMI adequately captures the variability in gut environment conditions that challenge microbial survival.

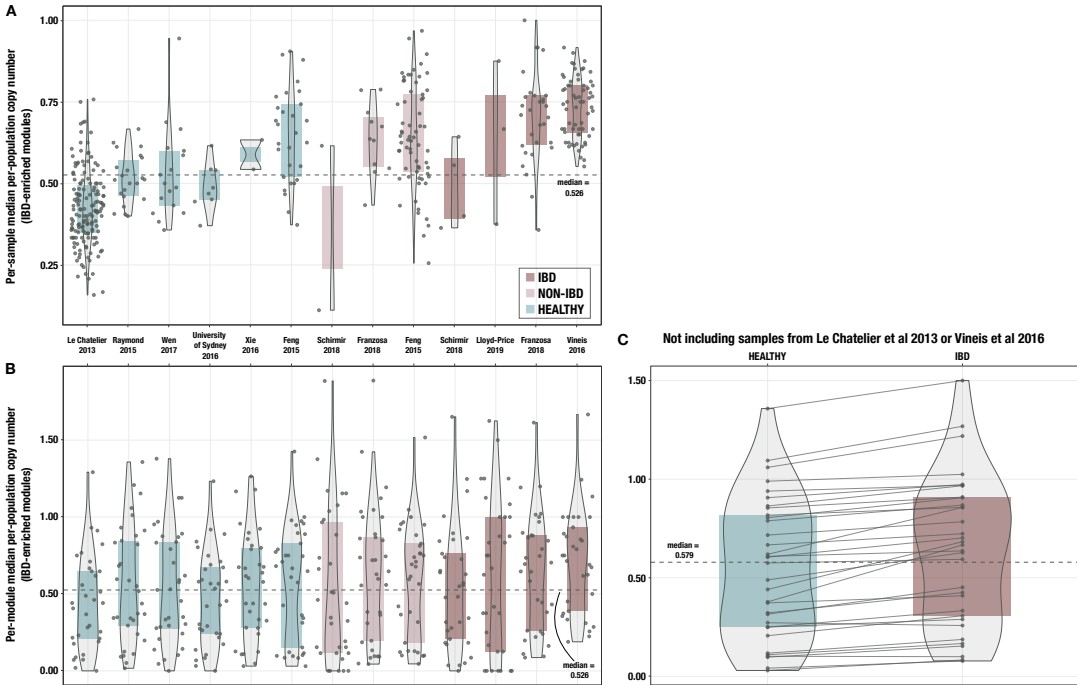

**Appendix 1—figure 6.** Boxplots of median per-population copy number of 33 inflammatory bowel disease (IBD)-enriched modules for samples from each individual cohort. (**A**) with medians computed within each sample (i.e. one point per sample) and (**B**) with medians computed for each IBD-enriched module (i.e. one point per module). The x-axis indicates study of origin. (**C**) Boxplots of median per-population copy number of 33 IBD-enriched modules for the 115 samples in the deeply sequenced set that are not from *Le Chatelier et al., 2013*, or *Vineis et al., 2016*. The dashed line indicates the overall median for all 33 modules, and solid lines connect the points for the same module in each sample group.

## Considerations of batch effect

One concern in comparing samples from multiple studies is that differential sample processing strategies could contribute to the signal between groups; in other words, batch effects could partially explain the observed trends between different cohorts. However, by including samples from a variety of studies in each group (healthy, non-IBD, and IBD) for our meta-analysis, we can mitigate the impact of batch effects on our observations. The similar distribution of the median normalized copy number for each of the 33 IBD-enriched metabolic modules (summarized across all samples within a given study), across all studies within a given sample group (*Appendix 1—figure 6B*), confirms that the sample group explains more of the trend than the study of origin.

Two studies dominate our sample set: *Le Chatelier et al., 2013* contributes 151 (52.8%) of the healthy samples, and *Vineis et al., 2016* contributes 64 (63.4%) of the IBD samples (*Supplementary file 1b*). To exclude that a cohort effect between these studies influences our observations, we repeated the IBD enrichment analysis on (1) *Le Chatelier et al., 2013* and *Vineis et al., 2016* only; as well as on (2) the remaining samples. While the results obtained from the two larger studies tend to have smaller p-values, the top IBD-enriched modules are broadly similar (Kendall correlation of Wilcoxon test p-values computed on two subsets: 0.59; see *Appendix 1—figure 7*), demonstrating that we are capturing generic signals across studies in our sample set.

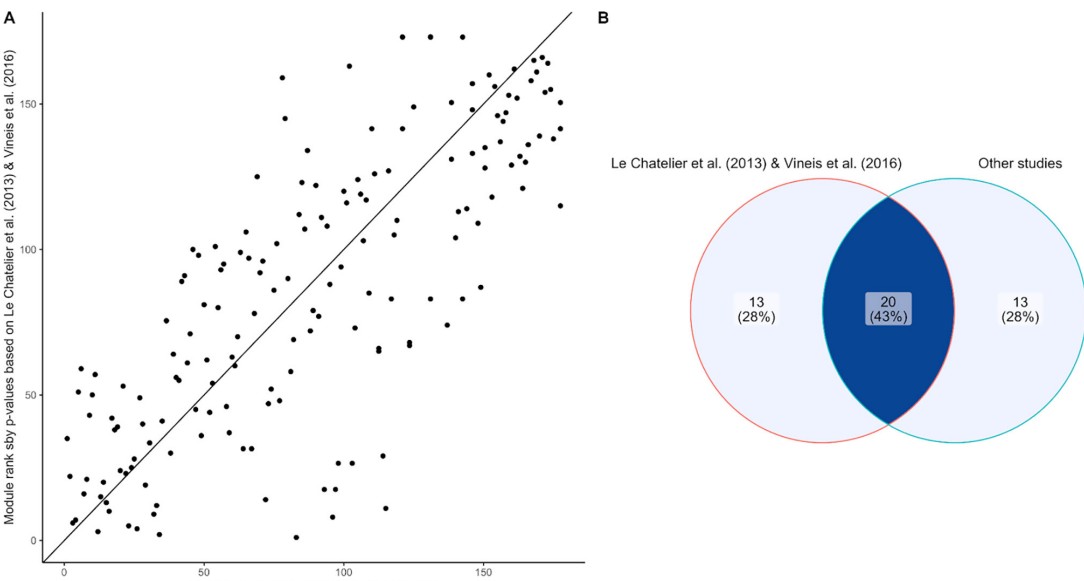

**Appendix 1—figure 7.** Assessing batch effect of the inflammatory bowel disease (IBD) enrichment study. (**A**) Scatter plot comparing the module ranks of Wilcoxon-Mann-Whitney p-values comparing IBD and healthy subjects on *Le Chatelier et al., 2013* and *Vineis et al., 2016* (y-axis) and the rest of our dataset (x-axis). (**B**) Venn diagram displaying the overlap of IBD-enriched modules identified by the 33 smallest p-values in *Le Chatelier et al., 2013* and *Vineis et al., 2016* (left) and the rest of our dataset (right). There is good agreement (20 out of 33) between the two sets of modules, indicating generalizability of the signals across studies used in our sample set.

## Testing the generalizability of the metagenome classifier

To check whether performance of our logistic regression classifier was similar across the different studies in our sample set, we tested the model's performance using a leave-two-studies-out cross-validation strategy, whereby we trained the classifier on all samples except for those from one IBD study and one healthy study, and then tested it using samples from the two studies that were left out, for a total of 24 folds. Performance was quite variable across the different folds, as expected considering the large range of sample sizes from each study and the variability in health status of each cohort. The best overall performance occurred when testing on healthy samples from *Le Chatelier et al., 2013*, with average accuracy of 89.9% across 3 folds. The worst performance occurred when testing on healthy samples from *Feng et al., 2015*, with average accuracy of 43.1% across 4 folds. In the fold leaving out healthy samples from *Le Chatelier et al., 2013* and IBD samples from *Vineis et al., 2016*, no IBD-enriched modules had p-values below our FDR-adjusted significance threshold of 2e-10 and therefore no classifier was trained. As these two studies contributed the largest number of samples to our deeply sequenced subset (*Le Chatelier et al., 2013*: n = 151 out of 330 or 45.8%, all of which were healthy samples; *Vineis et al., 2016*: n = 64 out of 330 or 19.4%, all of which were IBD samples), we considered that cohort-specific or study-specific effects could be driving the differential signal between healthy and IBD samples. To test this, we removed the samples from *Le Chatelier et al., 2013* and *Vineis et al., 2016*, and ran 10-fold cross-validation using an 80–20 train-test split of the remaining 115 samples (37 IBD, 78 healthy), using the 33 IBD-enriched modules (computed from the full sample set) as features. We found that the model performed better than a naive classifier, with an average fold accuracy of 66.5%, average true healthy rate of 69.4%, and an average true IBD rate of 61%. Therefore, while a portion of the signal in our initial analysis is indeed attributable to the differences between samples from *Le Chatelier et al., 2013* and *Vineis et al., 2016*, the classifier still captures an IBD-specific signal across the other studies using this set of IBD-enriched pathways.

Furthermore, we note that the two dominating studies represent individuals at the extremes of the health gradient across our sample set, as described previously. The *Le Chatelier et al., 2013* cohort, with its numerous exclusionary conditions, contains the healthiest individuals, while the *Vineis et al., 2016* cohort of proctocolectomy and pouchitis patients contains the unhealthiest. It is therefore unsurprising that there is a large contrast in the metabolic potential of the gut microbiome in these individuals, considering the biological differences in their respective gut environments. This

is also supported by the aforementioned ability of HMI to resolve the variability in host health, as demonstrated in *Appendix 1—figures 5B and 6A*.

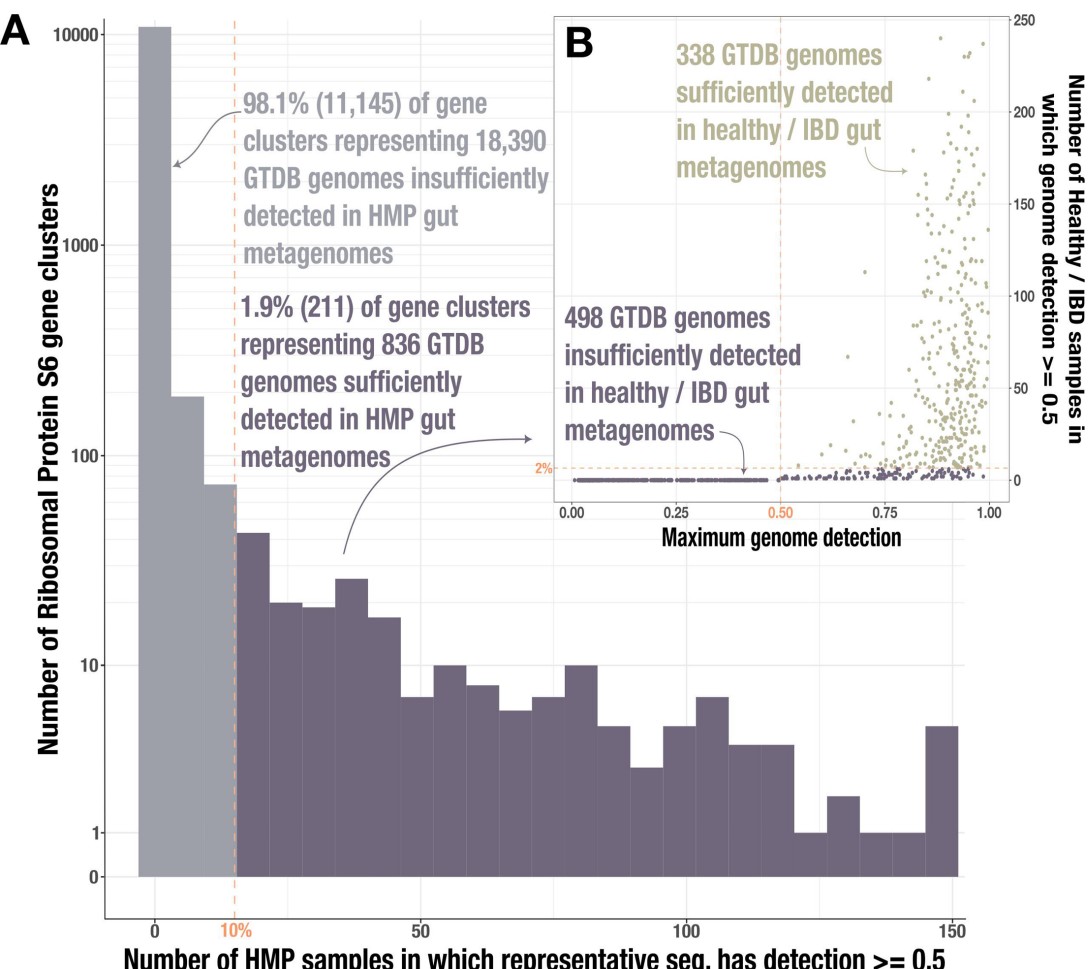

**Appendix 1—figure 8.** Identification of gut-associated genomes. (**A**) Histogram of Ribosomal Protein S6 gene clusters (94% ANI) for which at least 50% of the representative gene sequence is covered by at least 1 read (≥50% 'detection') in fecal metagenomes from the Human Microbiome Project (HMP) (*Human Microbiome Project Consortium, 2012*). The dashed line indicates our threshold for reaching at least 50% detection in at least 10% of the HMP samples; gray bars indicate the 11,145 gene clusters that do not meet this threshold while purple bars indicate the 836 clusters that do. (**B**) Data for the 836 genomes whose Ribosomal Protein S6 sequences belonged to one of the passing (purple) gene clusters. The y-axis indicates the number of healthy/inflammatory bowel disease (IBD) gut metagenomes from our set of 330 in which the full genome sequence has at least 50% detection, and the x-axis indicates the genome's maximum detection across all 330 samples. The dashed line indicates our threshold for reaching at least 50% genome detection in at least 2% of samples; the 338 genomes that pass this threshold are tan and those that do not are purple.

## Examining the impact of different HMI score thresholds on genome-level results

Determining the HMI status of a given genome required us to set a threshold for the HMI score above which a genome would be considered to have HMI. We tested several different thresholds by varying the average percent completeness of the 33 IBD-enriched metabolic modules that we expected from the 'HMI' genomes from ≥ 75% (corresponding to an HMI score of ≥ 24.75) to ≥ 85% (corresponding to an HMI score of ≥ 28.05). For each threshold, we computed the same statistics and ran the same statistical tests as those reported in our main manuscript to assess the impact of these thresholds on the results (*Supplementary file 3h*). At the highest threshold we tested (HMI score ≥ 28.05), a small proportion of the reference genomes (7%, or n = 24) were classified as HMI, so we did not test higher thresholds.

We found that the results from comparing HMI genomes to non-HMI genomes are similar regardless of which HMI score threshold is used to classify genomes into either group. No matter which HMI score threshold was used, the mean genome size and mean number of genes were higher for HMI genomes than for non-HMI genomes. On average, the HMI genomes were about 1 Mb larger and had 1032 more gene calls than non-HMI genomes. We ran two Wilcoxon rank-sum statistical tests to assess the following null hypotheses: (1) HMI genomes do not have higher detection in IBD samples than non-HMI genomes, and (2) HMI genomes do not have higher detection in healthy samples than non-HMI genomes. For both tests, the p-values decreased (grew more significant) as the HMI score threshold decreased due to the inclusion of more genomes in the HMI bin. The first test for higher detection of HMI genomes than non-HMI genomes in IBD samples yielded p-values less than α=0.05 at all HMI score thresholds. The second test for higher detection of HMI genomes than non-HMI genomes in healthy samples yielded p-values less than α=0.05 for the three lowest HMI score thresholds (HMI score ⩾ 24.75, ⩾25.08, or ⩾ 25.41). However, irrespective of significance threshold and HMI score threshold, there was always far stronger evidence to reject the first null hypothesis than the second, given that the p-value for the first test in IBD samples was 1–5 orders of magnitude lower (more significant) than the p-value for the second test in healthy samples.

IBD samples harbored a significantly higher fraction of genomes classified as HMI than healthy or non-IBD samples, regardless of HMI score threshold (p < 1e-15, Kruskal-Wallis rank-sum test). The p-values for this test increased (grew less significant) as the HMI score threshold decreased. This suggests that, at higher thresholds, relatively more genomes drop out of the HMI fraction in healthy/non-IBD samples than in IBD samples, thereby leading to larger differences and more significant p-values. Consequently, the HMI scores of genomes detected in IBD samples must be higher than the HMI scores of genomes detected in the other sample groups – indeed, the average HMI score of genomes detected within at least one IBD sample is 24.75, while the average score of genomes detected within at least one healthy sample is 22.78. Within a given sample, the mean HMI score of genomes detected within that sample is higher for the IBD group than in the healthy group: the average per-sample mean HMI score is 25.14 across IBD samples compared to the average of 23.00 across healthy samples.

We also assessed how many *Bacteroides* genomes were classified as HMI with each HMI score threshold. Given the prevalence of *Bacteroides* populations in individuals with IBD (*Vineis et al., 2016*), there is an a priori expectation that several genomes of microbes in this genus would be classified as HMI. However, because the maximum HMI score of *Bacteroides* genomes is 25.73 (average score: 24.17), only the three lowest HMI score thresholds enabled any Bacteroides to be classified as HMI, and even then the proportion of Bacteroides genomes classified as HMI was quite low (≤30%; *Supplementary file 3h*). In addition, several members of other groups of microbes that are not empirically associated with IBD – such as Lachnospiraceae and Ruminococcaceae species (*Vacca et al., 2020*; *Martín et al., 2023*) – were classified as HMI at these lower thresholds. Thus, to avoid introducing false positive genomes into the HMI group, we elected to set the HMI score threshold to at least 26.4, or 80% average completeness of the 33 IBD-enriched modules, despite the fact that none of the expected *Bacteroides* are classified as HMI at this level of stringency. There are likely multiple ways for a genome to be metabolically independent that are not captured by the simple classification strategy (and the specific IBD-enriched metabolic modules) utilized in this study.

## Appendix 2

## Comparison of anvi-estimate-metabolism to existing tools for metabolism reconstruction

There are two main strategies for estimation of metabolic potential from sequencing data. The first is metabolic modeling, in which genome-scale metabolic models (GSMMs) are built to computationally represent the network of available metabolic reactions for a particular organism (*Fang et al., 2020*; *Gu et al., 2019*). This strategy enables mathematical modeling of metabolic fluxes, typically with the linear programming technique known as flux-balance analysis (FBA) (*Orth et al., 2010*), which contextualizes the metabolic network within a set of constraints and thereby enables simulation of particular physiological conditions (*Sen and Orešič, 2019*). The second strategy is pathway prediction, which estimates the presence/absence and/or completeness of metabolic pathways to produce a summary of the metabolic capacity encoded in the input sequences. This technique has received less attention than metabolic modeling, but its results are more readily interpretable than models, and it is critical for understanding microbial functional roles without the need for a parameterized, in silico environment (*Zhou et al., 2022*). Both methods can be integrated with auxiliary information such as gene expression data or growth kinetics for validation of predicted metabolisms (*Gu et al., 2019*).

A variety of software tools exist for both types of metabolism reconstruction. Two early examples with basic approaches are the web-based server platforms KAAS (*Moriya et al., 2007*) and RAST (*Aziz et al., 2008*). KAAS simply highlights annotated enzymes within pathway maps from the KEGG database (*Kanehisa et al., 2006*), without producing any quantitative estimates. RAST similarly produces a limited summary of metabolism by categorizing enzymes into metabolic 'subsystems', but is also able to produce a metabolic model using the SEED infrastructure (*DeJongh et al., 2007*). There are a plethora of more contemporary modeling tools that generate GSMMs, including ModelSEED (*Henry et al., 2010*), RAVEN (*Agren et al., 2013*), merlin (*Dias et al., 2015*), CarveMe (*Machado et al., 2018*), and AuReMe (*Aite et al., 2018*). For comprehensive reviews about these tools, we refer the reader to several previous publications (*Faria et al., 2018*; *Mendoza et al., 2019*; *Gu et al., 2019*).

Software for pathway prediction include MinPath, DRAM, METABOLIC, and metaPathPredict. MinPath (*Ye and Doak, 2009*) uses integer programming to determine the minimum set of pathways that explain an input set of annotations. DRAM (*Shaffer et al., 2020*) and METABOLIC (*Zhou et al., 2022*) both integrate annotation of genes from various enzyme databases with estimation of pathway completeness; DRAM is specialized for working with metagenome-assembled genomes (MAGs) while METABOLIC focuses on biogeochemical cycles. The goal of metaPathPredict (*Geller-McGrath et al., 2023*) is to produce better estimations for incomplete genomes (especially MAGs reconstructed from environmental samples) using machine learning models trained on reference databases.

Though most of these tools specialize in one method of metabolism reconstruction, some software – such as Pathway Tools, KBase, gapseq, and KEMET – have the capacity for both reconstruction strategies. Pathway Tools (*Karp et al., 2016*) is a primarily web-based platform for numerous functional analyses based upon a custom 'omics data format called a Pathway/Genome Database, which can be used for both FBA and querying available metabolic capacity. KBase (*Arkin et al., 2018*) is an online workspace for hosting scientific analyses on 'omics datasets, and it contains apps for running existing metabolism software (such as DRAM, ModelSeed, and Rast) on uploaded data. Both gapseq (*Zimmermann et al., 2021*) and KEMET (*Palù et al., 2022*) were designed to produce more accurate metabolic models by incorporating a gap-filling process into their model generation workflows, and their pathway prediction capabilities are a side effect of this strategy. Gapseq achieves this via a novel linear programming algorithm and by utilizing a highly curated reaction database, while KEMET uses pathway prediction results for updating the metabolic models that it creates by internally running CarveMe (*Machado et al., 2018*).

Within the landscape of these current tools, 'anvi-estimate-metabolism' represents a software for pathway prediction, providing quantitative predictions of metabolic capacity by computing completeness scores for a predefined set of metabolic pathways. Anvi-estimate-metabolism distinguishes itself from the existing pathway prediction tools in several important ways. First, it is the only current tool that calculates a pathway redundancy metric, to the best of our knowledge.

It achieves this primarily by computing pathway copy numbers, which is an essential strategy for community-level analysis of metabolic capabilities. This program also generates an alternative metric for pathway redundancy by providing gene-level coverage values for enzymes within each pathway through its integration with the wider anvi'o codebase and data structures, assuming that read recruitment results are available. Second, 'anvi-estimate-metabolism' offers two distinct interpretation strategies for metabolic pathway definitions – a 'pathwise' strategy which considers all possible enzyme combinations ('paths') that would yield a complete pathway, and a 'stepwise' strategy that equally weighs alternative enzymes for the same reaction step. In other words, the specific enzymes used for a given metabolic conversion matter for the 'pathwise' metrics, but not for the 'stepwise' metrics. Each strategy is suitable for a different type of analysis – for instance, 'pathwise' metrics can be advantageous for studies of individual genomes while 'stepwise' metrics are appropriate for metagenomic analysis. Thus, the combinations of pathway interpretation strategies and metric type allow for the application of this tool to a variety of different research questions and input data types. In contrast, most of the other pathway prediction tools explicitly target genomic data.

Finally, 'anvi-estimate-metabolism' is one of the only tools that supports user-defined metabolic pathways rather than exclusively relying on reference pathways (i.e. from KEGG). The program enables users to create their own pathway files, using enzyme annotations from any functional annotation source (including from various standard databases such as NCBI COGs, Pfam, and CAZymes as well as from custom annotations imported by the user into their anvi'o databases). The only other software with a similar feature is DRAM, which offers estimation from 'custom distillate' files. However, DRAM's custom pathways entirely rely on enzymes from the KO database.

## Validation of PPCN approach on simulated metagenomic data

Our novel approach for normalizing metabolic pathway copy numbers by the estimated number of populations within a community to get PPCNs required validation. We used simulated metagenomes to test the robustness of our approach to the following common parameters of microbial communities that could potentially influence our comparison between healthy and IBD gut metagenomes: genome size, community size (e.g. number of distinct microbial populations within a metagenome), and diversity level (e.g. number of distinct phyla). We generated these synthetic metagenomes by randomly combining bacterial and archaeal representative genomes from different species clusters in the GTDB v95 (*Parks et al., 2022*) according to which parameter we wanted to test (see Supplementary Methods). These representative genomes included both isolate genomes and MAGs that are not necessarily complete or well studied, but represent a wide diversity of microbial taxa. Most of the samples we created were synthetic assemblies, generated by concatenating the contig sequences from each selected genome's FASTA into one file. To validate the full process starting from assembly, we also generated a test case starting from synthetic short reads (see Supplementary Methods).

We applied our PPCN approach to the synthetic metagenome assemblies to mimic our analysis of pathway completeness in gut metagenomes. This approach included gene annotation with KEGG KOfams and microbial SCGs, estimation of KEGG module copy numbers with 'anvi-estimate-metabolism', estimation of the number of populations in the community based on SCGs, and calculation of the normalized PPCN values from the resulting data. We then analyzed the accuracy of the PPCN values and their correlation with sample parameters.

### Two metrics for PPCN accuracy relative to genomic values

Validating the accuracy of the PPCN calculation required comparison of computed PPCN values to the true PPCN within a given metagenome. Obtaining this 'true' value for each synthetic community is difficult without expert knowledge of each microbe's metabolic capacities and extensive manual calculation. Therefore, we approximated the 'true' PPCN in a high-throughput manner by averaging the genomic pathway metrics within a given sample. We used 'anvi-estimate-metabolism' to compute the stepwise completeness or copy number for each metabolic pathway within each genome in the synthetic community, then averaged these values. Though our ability to predict metabolic capacity from individual genomes is in itself limited by genome (in)completeness and missing annotations, these values can serve as a reference point for how well we summarize community-level metabolism given our current genome-level knowledge.

We used both average genomic completeness and average genomic copy number values because each metric has advantages and limitations when approximating the true PPCN value. Average

genomic copy number could be considered the most direct analog to metagenomic PPCN, yet the copy number calculation in 'anvi-estimate-metabolism' very conservatively does not take into account partial copies of a pathway. Even when a pathway is highly complete in a given genome, a copy is not counted unless 100% of the steps are present; hence, the average genomic copy number value can underestimate the true PPCN. Genomic completeness scores can capture these partial versions of a pathway, making them a better approximation when the synthetic community harbors multiple incomplete genomes. However, completeness scores cannot resolve multiple copies of the same pathway encoded in an individual genome and can underestimate the true PPCN value, especially for short or simple pathways. Given these limitations, we assessed the accuracy of our computed PPCN values using both metrics individually as reference points. We computed 'PPCN error' by subtracting either average genomic completeness or average genomic copy number from the metagenomic PPCN value.

## The PPCN calculation is generally accurate relative to genomic values but can slightly overestimate community-level metabolic capacity

Across all our simulated test cases, we observed that metagenomic PPCN was typically very close to the average genomic metrics. Relative to average completeness scores, the distribution of PPCN error was centered close to 0 and somewhat right-skewed, with a mean error ranging from –0.13 to –0.10 and a standard deviation ranging from 0.18 to 0.22. Relative to average copy number, PPCN error was centered closer to 0 and left-skewed, with a mean error ranging from 0.04 to 0.06 and a standard deviation ranging from 0.10 to 0.11 (*Supplementary file 6a*; *Appendix 2—figure 1*). The error range was limited in both cases, but was much smaller for error computed relative to average genomic copy number. Thus, while overall quite accurate, metagenomic PPCN has a slight tendency to underestimate average genomic completeness and overestimate average genomic copy number. For the subset of IBD-enriched pathways, the PPCN error distributions showed similar trends but were centered at a slightly higher point, with a mean of –0.06 to –0.02 relative to average completeness and a mean of 0.14–0.17 relative to average copy number.

Examining the outliers in the underlying data revealed explanations for these trends. A primary reason for the overestimation of average genomic copy number was a phenomenon we term the 'pathway complementarity effect'. When multiple members of a synthetic community contained partial yet complementary portions of a given metabolic pathway, these enzymes combined at the metagenomic level to produce additional 'complete' copies of the pathway. This effect was not observed when PPCN is compared to average genomic completeness scores because the latter metric takes partial copies of the pathway into account, thus reducing the observed error. Although cross-feeding is known to occur within microbial communities (*Culp and Goodman, 2023*; *Pacheco et al., 2019*) and in some cases pathway complementarity at the metagenome level could capture a legitimate biological signal, for the most part this is a technical artifact yielding a degree of error in the PPCN calculation, and is a natural outcome of the decision to consider the metagenome as one large pot of genes for the purposes of pathway prediction. That said, the magnitude of this error is typically small.

Metagenomic PPCN tends to slightly underestimate average genomic completeness due to the conservative nature of the copy number calculation. In cases when multiple members of a synthetic community have incomplete pathways (with nonzero completeness scores) and their respective portions of a given pathway are noncomplementary, the average completeness scores will always be higher than the PPCN value, which does not count partial copies.

A few specific pathways had systematically overestimated PPCNs relative to one of the genomic metrics. For example, PPCN values for the beta-oxidation pathway (M00086) had the highest average error (0.946) relative to genomic completeness scores. This is an extremely short pathway, with only one reaction that can be catalyzed by one of two alternative enzymes. Thus, it represents an extreme case in which the copy number of the pathway is directly equivalent to the number of annotations for these two enzymes, which can lead to an extremely high copy number at the metagenome level. Within a given genome, the completeness of this pathway can never increase beyond 100% regardless of the number of annotations, and this limitation translates into a maximum average completeness score of 100% at the metagenome level. The systematic overestimation of PPCN for M00086 in this case is therefore due to the nature of the pathway itself and the limitation of average completeness score as a reference point. Relative to average genomic copy number, a

module for glycolysis (M00001) had the highest average PPCN error (0.426) across all test cases. This is an extremely common metabolic pathway expected to occur in the majority of microbial genomes, which likely increases the pathway complementarity effect.

## Accuracy of estimating the number of microbial populations within a metagenome

We also explored the accuracy of our method for estimating the number of populations from SCGs. In general, this method has a slight tendency to underestimate the true number of populations, and errors are more likely when the actual community size is larger (*Appendix 2—figure 3*). Regardless, the estimates were within 2 of the correct value over 90% of the time in all test cases for which we combined genomic contigs to create a synthetic metagenomic assembly. In the more realistic test case, when we generated synthetic short reads and assembled those reads de novo, the estimation accuracy dropped and was only within 2 of the correct value 67% of the time (*Supplementary file 6a*). Accuracy increased with greater sequencing depth (*Appendix 2—figure 2*), similar to what we observed in the gut metagenomes used in our main analysis (*Appendix 1—figure 1*). This suggests that most errors in estimation were due to missing SCGs from incomplete genomes.

Since the estimated number of populations is the denominator in the PPCN calculation, underestimating these values can contribute to overestimation of the PPCN. This effect was not as strong as the pathway complementarity effect in the samples that we manually checked, which all belonged to the ideal test cases.

## The impact of genome size in an 'ideal' scenario

To explore how genome size influences the PPCN approach, we 'binned' genomes according to genome length to obtain the following size categories: small genomes (<2 Mb), medium genomes (2 Mb up to 5 Mb), and large genomes (5 Mb up to 20 Mb). Genomes larger than 20 Mb in size were excluded. We then generated 189 random communities each containing 20 genomes from two size categories (S vs M, S vs L, and M vs L), such that each sample included a different proportion of genomes from each size category on a gradient from 0 to 1 (see Supplementary Methods). We independently analyzed each group of samples with the same size category pair with a Spearman's correlation test to identify the relationship between genome size and PPCN, PPCN accuracy, and the estimated number of populations in the metagenome.

Across all pathways in the KEGG MODULE database, proportion of small genomes in a given sample had a weak negative correlation (–0.09 < R<–0.02) with PPCN values (*Appendix 2—figure 4*, *Supplementary file 6b*). The correlation became moderate (–0.48 ≤ R≤–0.21) for the subset of 33 IBD-enriched modules identified in our comparison of healthy and IBD gut metagenomes (*Appendix 2—figure 5*, *Supplementary file 6b*). All correlations were significant with a p-value threshold of p < 0.05, and the correlations were strongest for the S vs L group (*Supplementary file 6b*). Thus, communities with more large genomes tend to have higher PPCN values, which makes sense considering that larger microbial genomes encode more genes and therefore more metabolic pathways. The stronger correlation for the subset of IBD-enriched modules mirrors our observation that IBD gut metagenomes harbor microbes with larger genomes with increased metabolic capacity (see 'Reference genomes with higher metabolic independence are overrepresented in the gut metagenomes of individuals with IBD' in the main manuscript), and suggests that this subset of pathways is particularly likely to be found in large genomes regardless of environmental or taxonomic context.

The accuracy of the PPCN calculation had a weak correlation (–0.02 < R < 0.07) with proportion of small genomes, regardless of which genomic metric was used to approximate the error (*Appendix 2—figure 4*, *Supplementary file 6b*). The correlations became weakly negative for the subset of IBD-enriched pathways (–0.13 < R < 0.04), indicating that PPCN values are slightly more accurate for these pathways in communities of larger genomes, although several of the latter correlations were nonsignificant (*Appendix 2—figure 5*, *Supplementary file 6b*).

Proportion of small genomes had significant, moderate to strong negative correlations with the accuracy of community size estimates (–0.71 ≤ R≤–0.36; p < 1e-02), indicating that these estimates are more accurate when genomes in the community are larger (*Appendix 2—figure 6*, *Supplementary file 6b*). This might reflect a general tendency of larger genomes to contain more complete sets of SCGs.

## The impact of genome size in a 'realistic' scenario

The prior test, which is based on the combination of pre-assembled genomic contigs into a synthetic metagenomic 'assembly', validates how our approach works for an 'ideal' scenario in which all community members can be assembled. However, a full metagenomic analysis workflow starts from highly fragmented sequencing reads, which must be assembled into contigs before gene annotation and other downstream analyses can be run. The assembly process can result in data loss if some microbial populations in the community cannot be fully assembled, which can impact downstream results. To simulate this entire process, we took the same 189 synthetic communities generated for the genome size test case and generated simulated short reads from each genome to create synthetic metagenome sequencing samples (see Supplementary Methods). We randomly assigned each genome in the community a relative abundance value from a normalized relative abundance curve of the top 20 most abundant populations in a healthy human gut metagenome (*Appendix 2—figure 7*, *Supplementary file 6*, Supplementary Methods). We then converted the relative abundance values into coverage values ranging from 20× to 420×, and generated synthetic sequencing reads from each genome to produce its corresponding coverage value. Thus, each of the synthetic samples had the same coverage distribution across its 20 community members. After assembling the synthetic samples, we ran the PPCN workflow and performed the same validation described above.

The validation results for this more realistic scenario showed largely the same trends as the ideal genome size case (*Appendix 2—figure 8*, *Supplementary file 6a and b*) with two exceptions: (1) the accuracy of the population size estimates was lower, as mentioned previously. (2) The correlation between genome size proportion and population size estimation accuracy was weakly positive and nonsignificant for the M vs L group of genomes (*Appendix 2—figure 9*, *Supplementary file 6a and b*). Given the similarity between the results for the 'ideal' case and the 'realistic' case, we decided to only test the less computationally intensive 'ideal' case for the remaining parameters.

## The impact of community size

To analyze the impact of community size on our approach, we generated 120 synthetic metagenome assemblies each containing 5, 10, 15, or 20 randomly selected genomes from the same size category (S, M, or L; see Supplementary Methods). We independently analyzed each size category as described above.

As expected, we observed a significant positive correlation between metagenomic copy number (the numerator of PPCN) and community size in each group, likely driven by the increase in the copy number of core metabolic pathways in larger communities (*Appendix 2—figure 10*). Interestingly, this correlation was much stronger for the subset of IBD-enriched pathways ($0.49 \leq R \leq 0.67$) than for all modules ($0.12 \leq R \leq 0.13$).

However, the correlation was much weaker and often nonsignificant for the normalized PPCN data in both groups of modules (all modules: $0.01 < R < 0.04$, enriched modules: $0.04 < R < 0.09$, *Supplementary file 6b*; *Appendix 2—figure 11*), which demonstrates the suitability of our normalization method to remove the effect of community size in comparisons of metagenome-level metabolic capacity.

The correlations between community size and PPCN accuracy were weakly yet significantly positive ($0.07 < R \leq 0.16$), were slightly stronger for the subset of enriched modules ($0.11 \leq R \leq 0.25$), and indicate that PPCN values are slightly more accurate for smaller communities (*Supplementary file 6b*, *Appendix 2—figure 11*).

Finally, the accuracy of estimating community size was negatively correlated with community size for communities of small and medium-sized genomes ($R \leq -0.61$, $p < 1e-04$), meaning our method more accurately predicts the size of smaller communities. However, this trend disappeared for communities of large genomes, in which community sizes were predicted with 100% accuracy (*Appendix 2—figure 12*). Similar to the observation in the genome size test case where the presence of more large genomes increased the accuracy of community size estimates, this trend might result from a tendency of larger genomes to contain more complete sets of SCGs.

## The impact of diversity

Our final test case assessed how diversity level influences the PPCN approach. We generated 100 synthetic metagenome assemblies, each containing 20 randomly selected genomes from a different number of phyla (see Supplementary Methods). The number of genomes representing a given phylum

was approximately equal in a given sample; however, genome size was not necessarily consistent. Due to the lack of genome size groups in this test case, we analyzed all 100 samples collectively.

The number of phyla represented in a given community was weakly correlated with both PPCN and PPCN accuracy (0.02 < R < 0.04, *Supplementary file 6*, *Appendix 2—figure 13*), with only slightly stronger correlations for the subset of IBD-enriched modules (0.04 < R < 0.07, *Supplementary file 6b*, *Appendix 2—figure 13*). The accuracy of community size estimates were not significantly correlated with diversity level (*Appendix 2—figure 14*).

## Overall impact on the comparison between healthy and IBD gut metagenomes

In summary, our validation strategy revealed good accuracy at estimating metagenome-level metabolic capacity relative to our genome-level knowledge in the simulated data. While it often underestimated average genomic completeness by ignoring partial copies of metabolic pathways and often overestimated average genomic copy number due to the effect of pathway complementarity between different community members, the magnitude of error was overall limited in range and the error distributions were centered at or near 0. Furthermore, we observed these broad error trends in all cases we tested, and therefore we expect that they would also apply to both sample groups in our comparative analysis. Thus, we next considered how the PPCN approach might have influenced our analyses that considered metagenomes from healthy individuals and from those who have IBD – two groups that differed from one another with respect to some of the variables considered in our tests.

Most of the correlations between PPCN or PPCN accuracy and sample parameters were weak, yet significant (*Appendix 2—table 1*). They showed that community size and diversity level have limited influence on the PPCN calculation, while genome size does not influence its accuracy. The only exception was the moderate correlation between PPCN and genome size, particularly for the subset of IBD-enriched pathways. It was a negative correlation with the proportion of small genomes in a metagenome, indicating that PPCN values for these pathways are larger when there are more large genomes in the community and suggesting that these pathways tend to occur frequently in larger genomes. This is in line with our observation that IBD communities contain more large genomes and therefore confirms our interpretation that the populations surviving in the IBD gut microbiome are those with the genomic space to encode more metabolic capacities.

**Appendix 2—table 1.** Summary of correlation relationships between per-population copy number (PPCN), PPCN accuracy, and sample parameters from Supplementary file 6b.
We labeled each pair according to the strength of correlation indicated with the R value: NO, very weak with |R| ≤ 0.19; SOME, weak with 0.19 < |R| ≤ 0.39; YES, moderate to strong with |R| > 0.39. A '+' sign in front of the R value indicates that all R values were positive, a '-' sign indicates that all were negative, and the absolute value sign indicates that they had mixed signs. An asterisk (*) indicates that some of the correlations were nonsignificant (p > 0.05).

| Correlation? | PPCN (all) | PPCN error (all) | PPCN (enriched) | PPCN error (enriched) |
|---|---|---|---|---|
| Genome size (proportion of smaller genomes) | NO (-R < 0.09) | NO (+R ≤ 0.11) | YES (-R ≥ 0.48) | NO* (\|R\| < 0.13) |
| Community size (# of populations) | NO* (+R < 0.04) | SOME (+R ≤ 0.16) | NO* (+R < 0.085) | SOME (+R ≤ 0.25) |
| Diversity (# of phyla) | NO (+R = 0.033) | NO (+R < 0.04) | NO (+R = 0.069) | NO (+R < 0.065) |

If we consider even the weak correlations, two of those relationships indicate that our approach would be more accurate for IBD metagenomes than for healthy metagenomes. For instance, PPCN accuracy was slightly higher for smaller communities (as in IBD samples), with a weakly positive correlation between PPCN error and community size. It was also slightly more accurate for less diverse communities (as in IBD samples), with a weakly positive correlation between PPCN error and number of phyla. The only opposing trend was the weakly positive correlation between PPCN error and proportion of smaller genomes, which favors higher accuracy in communities with smaller genomes (as in healthy samples). Given that our analysis focuses on the pathways enriched in IBD samples, an overall higher accuracy in IBD samples would increase the confidence in our enrichment results.

We also examined the accuracy of our method to predict the number of populations within a metagenome based on the distribution and frequency of SCGs (i.e. the denominator in the calculation of PPCN). Our benchmarks show that the estimates are overall accurate, where most errors reflect a negligible amount of underestimations of the actual number of populations. Errors occurred more frequently for the realistic synthetic assemblies generated from simulated short read data than for the ideal synthetic assemblies generated from the combination of genomic contigs. The correlations between estimation accuracy and sample parameters indicated that the population estimates are more accurate for smaller communities and communities with more large genomes, as in IBD samples (*Appendix 2—table 2*). Thus, this method is more likely to underestimate the community size in healthy samples, and these errors could lead to overestimation of PPCN in healthy samples relative to IBD samples. Thus, the enrichment of a given pathway in the IBD samples would have to overcome its relative overestimation in the healthy sample group, making it more likely that we identified pathways that were truly enriched in the IBD communities.

**Appendix 2—table 2.** Summary of correlation relationships between the accuracy of community size estimates and sample parameters from Supplementary file 6b.

We labeled each pair according to the strength of correlation indicated with the R value: NO, very weak with $|R| \leq 0.19$; SOME, weak with $0.19 < |R| \leq 0.39$; YES, moderate to strong with $|R| > 0.39$. A '+' sign in front of the R value indicates that all R values were positive, a '-' sign indicates that all were negative, and the absolute value sign indicates that they had mixed signs. An asterisk (*) indicates that some of the correlations were nonsignificant ($p > 0.05$).

| Correlation? | Error in community size estimate |
|---|---|
| Genome size (proportion of smaller genomes) | YES* ($0.14 < |R| < 0.72$) |
| Community size (# of populations) | YES ($0.6 < -R < 0.7$) (except for large genomes) |
| Diversity (# of phyla) | NO* ($+R = 0.071$) |

Overall, the consideration of our simulations in the context of healthy vs IBD metagenomes suggest that slight biases in our estimates as a function of unequal diversity with sample groups should have driven PPCN calculations toward a conclusion that is opposite of our observations under neutral conditions. Thus, clear differences between healthy vs IBD metagenomes that overcome these biases suggest that biology, and not potential bioinformatics artifacts, is the primary driver of our observations.

## Supplementary methods

Below are the methods used for validation of our approach on simulated metagenomic data. All custom scripts and auxiliary data files are available on Figshare at DOI:10.6084/m9.figshare.26038018. A reproducible workflow for this section is also available at https://merenlab.org/data/ibd-gut-metabolism/.

### Generation of synthetic communities

We generated synthetic metagenomic communities using species cluster representative genomes from the GTDB v95 (*Parks et al., 2022*), the same database utilized in our main analysis. We used genome length to group each genome into different size categories: 7238 'small' genomes of < 2 Mbp; 18,416 'medium'-sized genomes of 2 Mbp to < 5 Mbp; and 6254 'large' genomes of 5 Mbp to 20 Mbp. Two especially large cyanobacterial genomes with over 20 Mbp were excluded from further analysis.

We then used custom scripts to randomly combine these genomes into synthetic communities for each simulated test case. To test the robustness of our approach to genome size, we combined 20 genomes of two size categories (small and medium; small and large; medium and large) in different proportions in each community, varying the number of genomes from the first size group from 0 to 20 and varying the number from the second size group from 20 to 0. We randomly selected genomes from each size group without replacement, and generated 3 samples per proportion value for each size category pair. This produced a total of 189 synthetic communities representing

a gradient of mixed genome sizes for the genome size test case. To test the robustness of our approach to community size, we combined 5, 10, 15, or 20 randomly selected genomes (without replacement) of a single size category (small, medium, or large) in each community. We generated 40 synthetic communities (10 for each community size value) for each genome size category to obtain a total of 120 samples in the population size test case. Finally, to test the robustness of our approach to different diversity levels, we generated communities containing a total of 20 random bacterial genomes, varying the number of phyla represented in the community from 1 to 20 and ensuring that there was an approximately equal number of genomes from each phylum in the sample. Each phylum was selected randomly without replacement from the set of unique bacterial phyla represented in the GTDB. Phyla containing less than 20 genomes were excluded. We randomly selected genomes from each phylum without replacement, and we generated 5 synthetic communities for each diversity level to obtain 100 samples representing a gradient of diversity for the diversity test case.

## Generation of 'ideal' synthetic metagenomes

The 'ideal' synthetic metagenomes represent the scenario in which all microbial populations in a metagenome are fully represented in the assembly (at least to the level of completion for each genome in the GTDB). For each of the aforementioned synthetic communities (in each test case), we concatenated the individual genome FASTA files into one FASTA containing the contig sequences of all genomes in the community. We then applied the approach (see main Methods) used in our main analysis to estimate population sizes, calculate metagenomic copy number, and compute PPCN for metabolic pathways in each synthetic metagenome. We used the same version of the KEGG database (from December 2020) as in our main analysis for consistency with those results.

## Generation of 'realistic' synthetic metagenomes

We generated 'realistic' synthetic metagenomes for the 189 samples in the genome size test case by simulating paired-end short reads from genomic data. To ensure that the populations in these samples had realistic relative abundances, we first computed a relative abundance curve that is typical of healthy human gut metagenomes. For this task, we used the relative abundance data computed by MetaPhlAn 3 for publicly available gut metagenomes (*Beghini et al., 2021*). We filtered this dataset to keep only species-level relative abundance values in 662 healthy control metagenomes, sorted the values in descending order, and kept only the top 20 relative abundances in each sample. We then scaled each sample's data to have the same maximum relative abundance value, averaged the values at each rank, scaled the resulting averages to obtain a minimum average relative abundance value of 1, rounded the averages into integer coverage values, and multiplied the coverages by 20 to obtain high enough sequencing depth for proper assembly of the synthetic metagenomes. This process yielded coverage values of 20–420× for individual populations within a 'typical' healthy human gut metagenome (*Appendix 2—figure 7*).

We used a custom script to randomly select one of these typical coverage values for each genome in a given synthetic community. With the program 'gen-paired-end-reads' (https://github.com/merenlab/reads-for-assembly copy archived at *Eren, 2025*) we generated synthetic short reads from each input genome with the specified target coverage value, only simulating from contigs with length $\geq$ 1000 bp. The simulation parameters for the paired-end reads were as follows: read length of 150, inner distances with a mean of 100 bp and standard deviation of 10 bp, and a 0.05% sequencing error rate. After simulating the short reads, we assembled each individual synthetic metagenome with MEGAHIT (*Li et al., 2015*), specifying '--min-contig-len 1000'. From there we followed the same approach used in our main analysis and for the 'ideal' synthetic metagenomes to obtain PPCNs for metabolic pathways.

## Metabolism estimation for individual genomes

To obtain genomic pathway prediction metrics for comparison to the metagenomic values, we ran 'anvi-estimate-metabolism' on each genome used to generate the synthetic communities. We used the following parameters: '--include-zeros', '--matrix-format', and '--add-copy-number' (which were the same parameters used for estimating metabolism in the synthetic metagenomes).

## Analysis of estimation accuracy and robustness to sample parameters

We wrote custom scripts to compare the PPCN values from metagenomes to the metabolism estimation data from individual genomes, and to compare the estimated number of populations to the true community size. We then used a custom R script for analysis and visualization of these comparisons, which include statistical analysis, Spearman's correlation with the sample parameters, and figure generation.

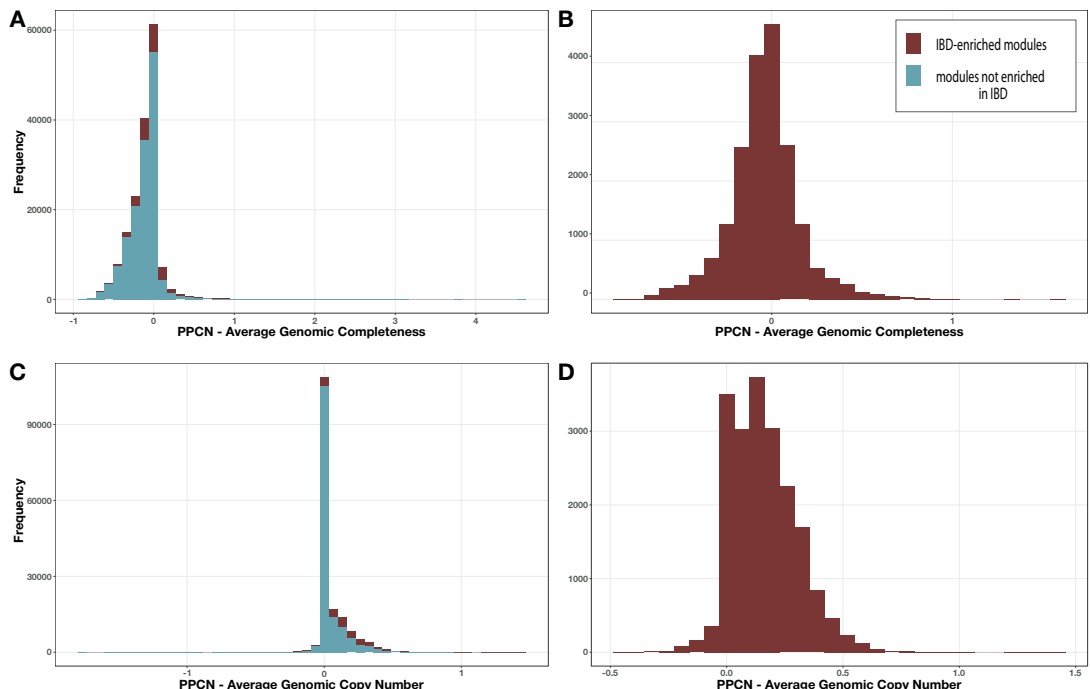

**Appendix 2—figure 1.** Distribution of per-population copy number (PPCN) error relative to average genomic completeness. (**A**, **B**) or average genomic copy number (**C**, **D**) for all modules (**A, C**) or just the inflammatory bowel disease (IBD)-enriched modules (**B, D**).

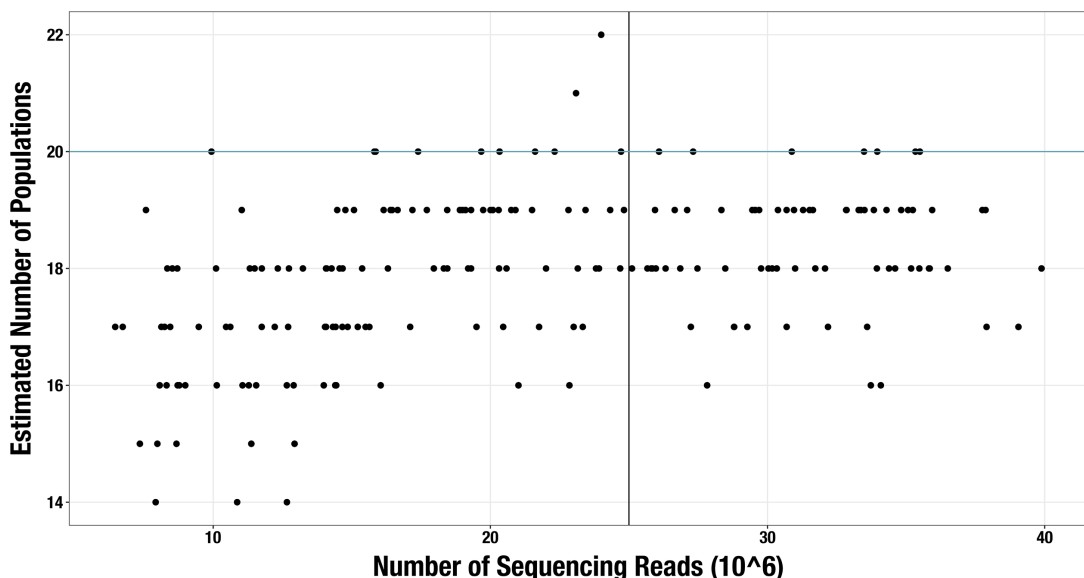

**Appendix 2—figure 2.** Scatterplot of sequencing depth vs estimated number of microbial populations in each of 189 'realistic' synthetic metagenome assemblies. The blue line shows the actual number of genomes in each synthetic community (n = 20) and the black line shows the sequencing depth threshold used in our main analysis.

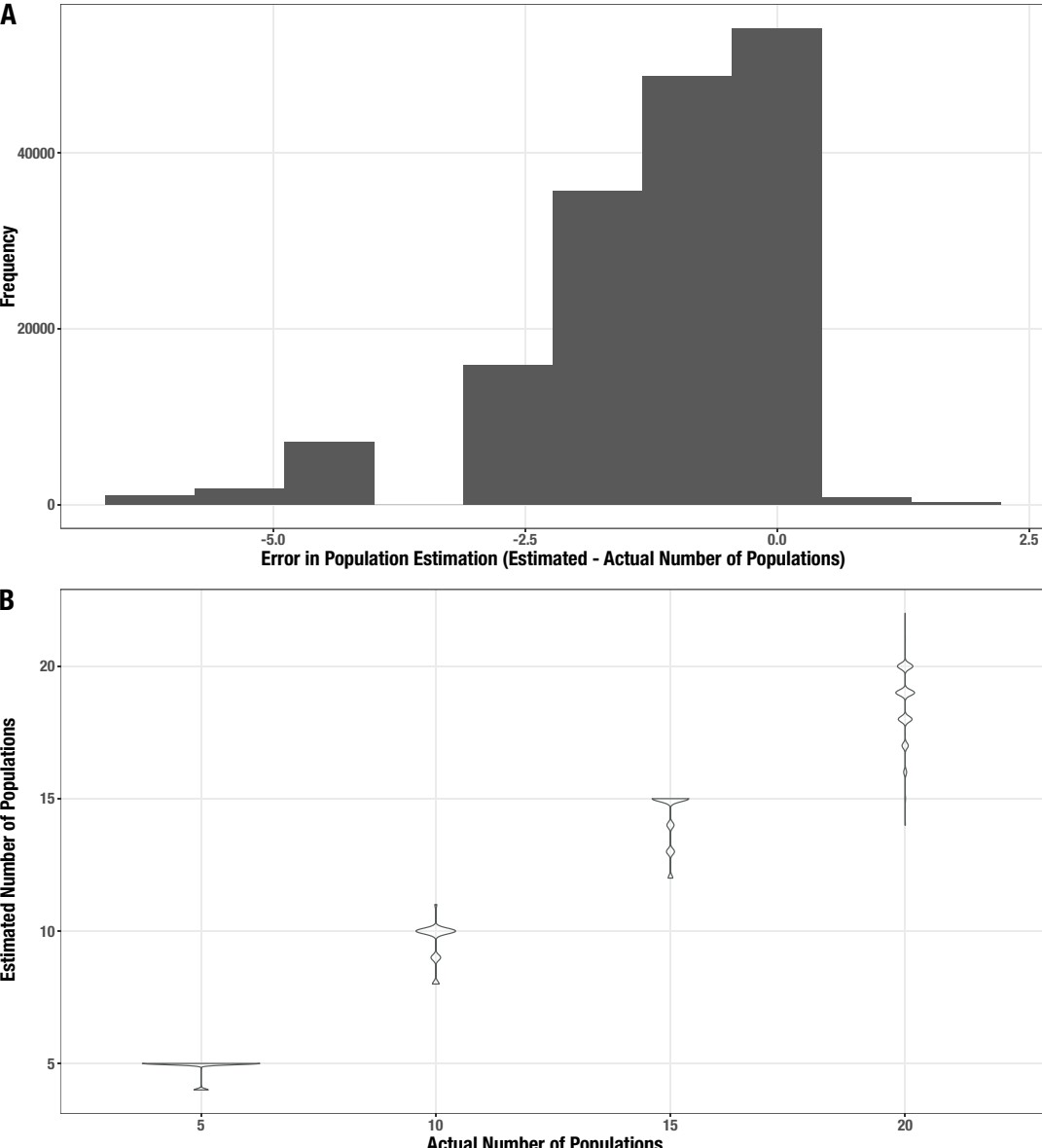

**Appendix 2—figure 3.** Distribution of error for the estimated number of populations in the synthetic metagenomes. (**A**) Histogram of the difference between estimated and actual community size. (**B**) Distribution of estimates (y-axis) for each actual community size (x-axis).

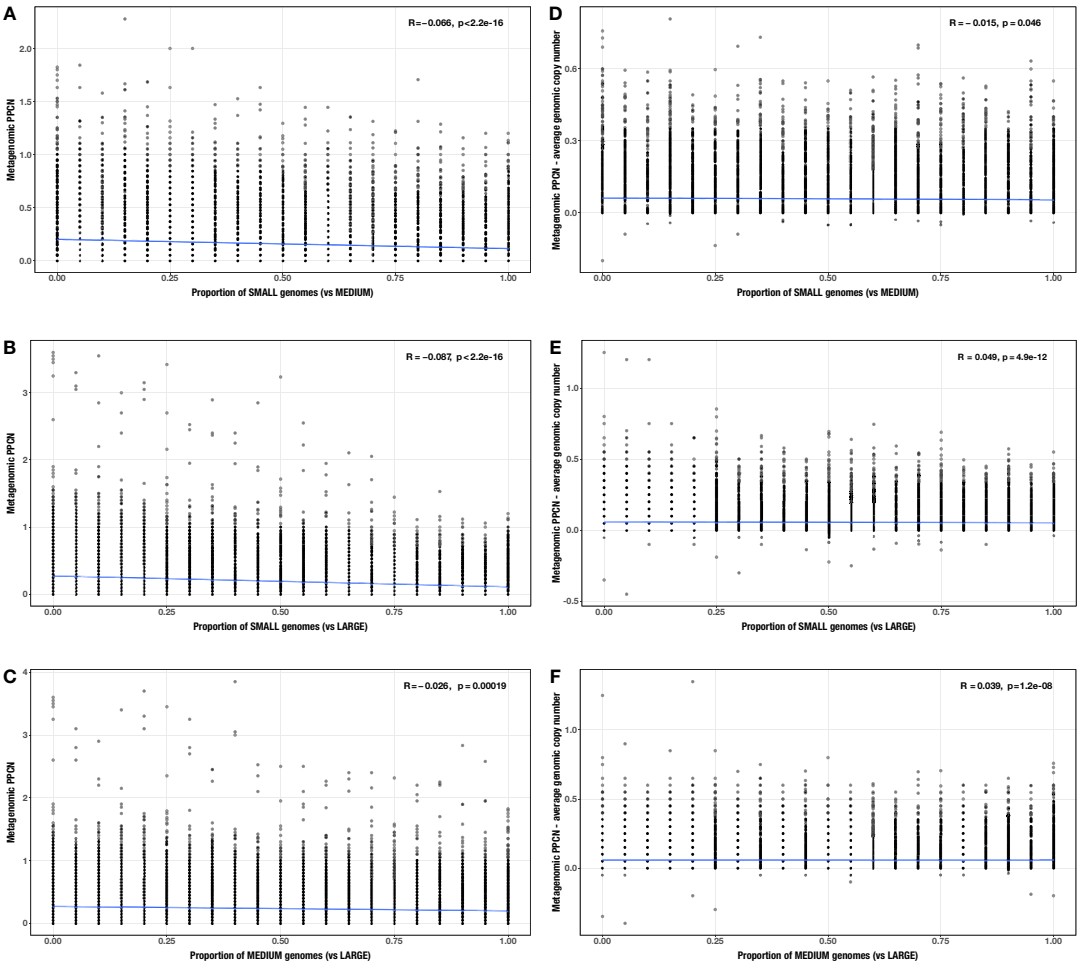

**Appendix 2—figure 4.** Correlations between proportion of genomes in smaller size category and (**A–C**) per-population copy number (PPCN) or (**D–F**) PPCN error relative to average genomic copy number for each size category pair (**A/D**) small vs medium genomes; (**B/E**) small vs large genomes; (**C/F**) medium vs large genomes across all modules. The Spearman's correlation coefficients and p-values are shown in the top-right corner of each plot, and regression lines are plotted in blue.

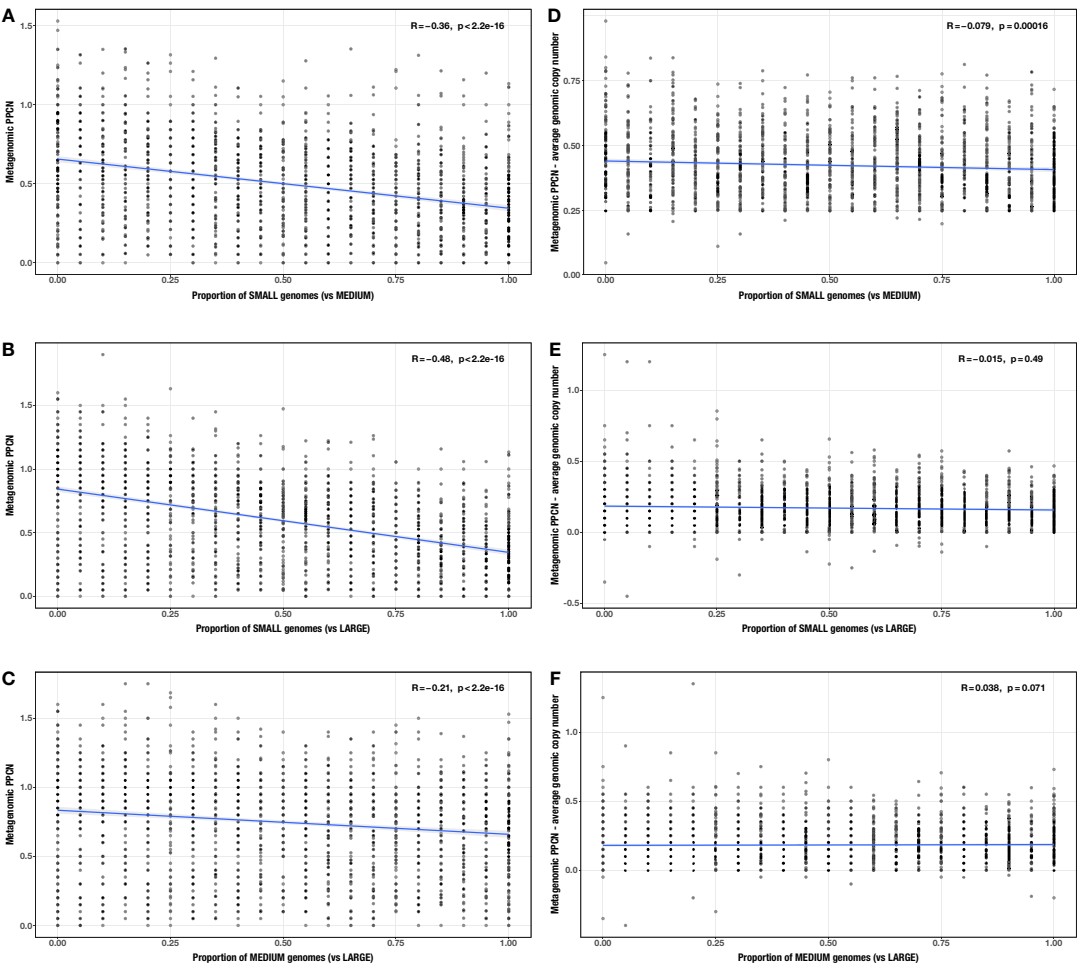

**Appendix 2—figure 5.** Correlations between proportion of genomes in smaller size category and (**A–C**) per-population copy number (PPCN) or (**D–F**) PPCN error relative to average genomic copy number for each size category pair (**A/D**: small vs medium genomes; **B/E**: small vs large genomes; **C/F**: medium vs large genomes) across inflammatory bowel disease (IBD)-enriched modules (n = 33). The Spearman's correlation coefficients and p-values are shown in the top-right corner of each plot, and regression lines are plotted in blue.

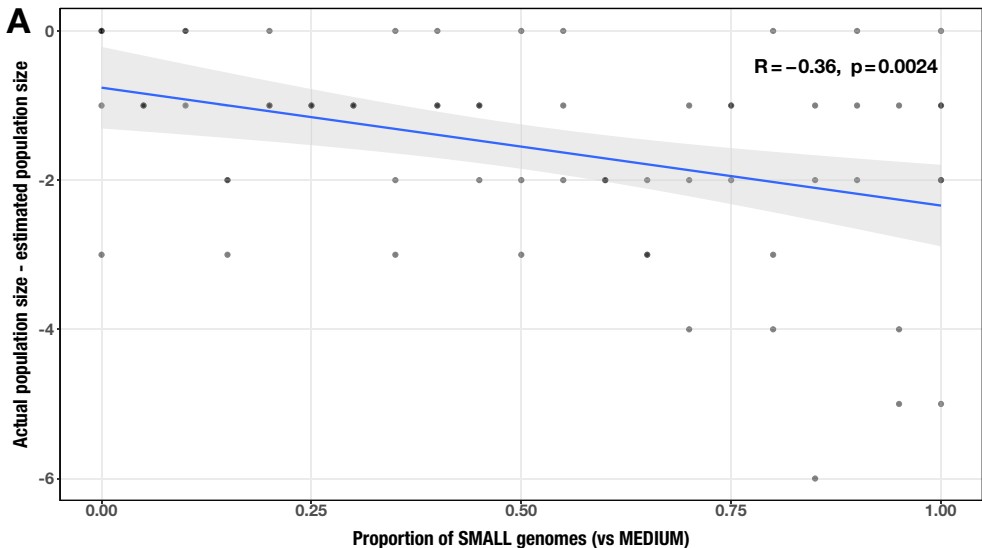

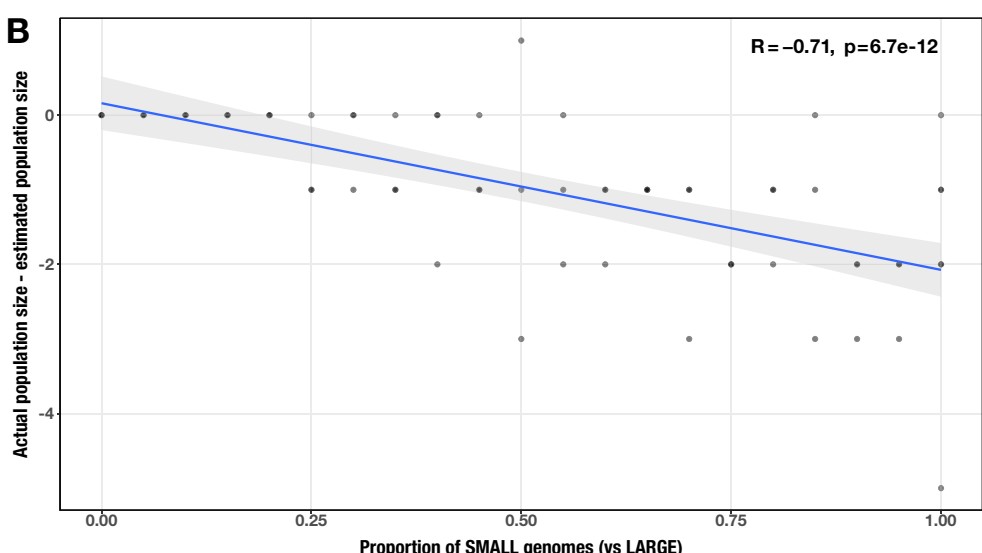

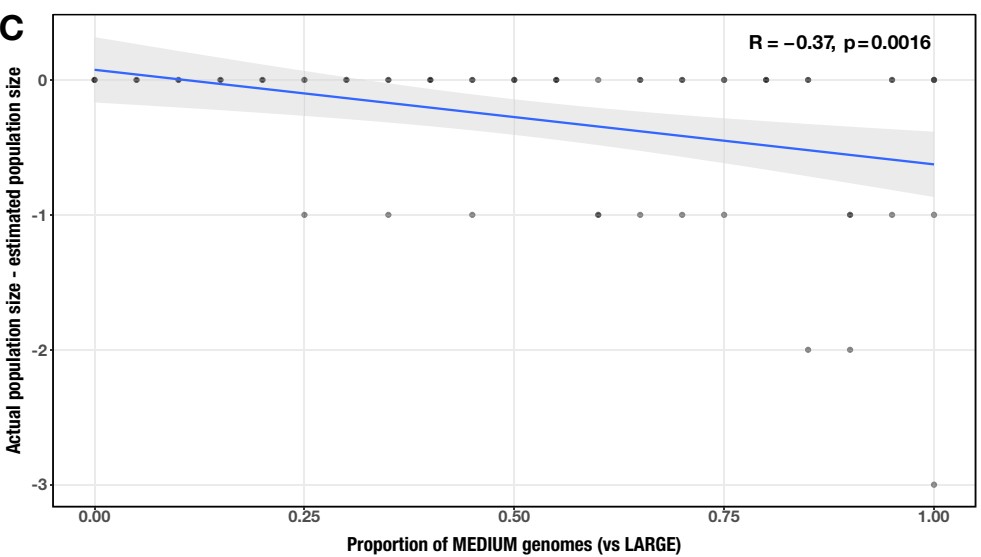

**Appendix 2—figure 6.** Correlation between proportion of genomes in smaller size category and error in community size estimate (relative to actual community size) for each size category pair (**A**: small vs medium genomes; **B**: small vs large genomes; **C**: medium vs large genomes). The Spearman's correlation coefficients and p-values are shown in the top-right corner of each plot, and regression lines are plotted in blue.

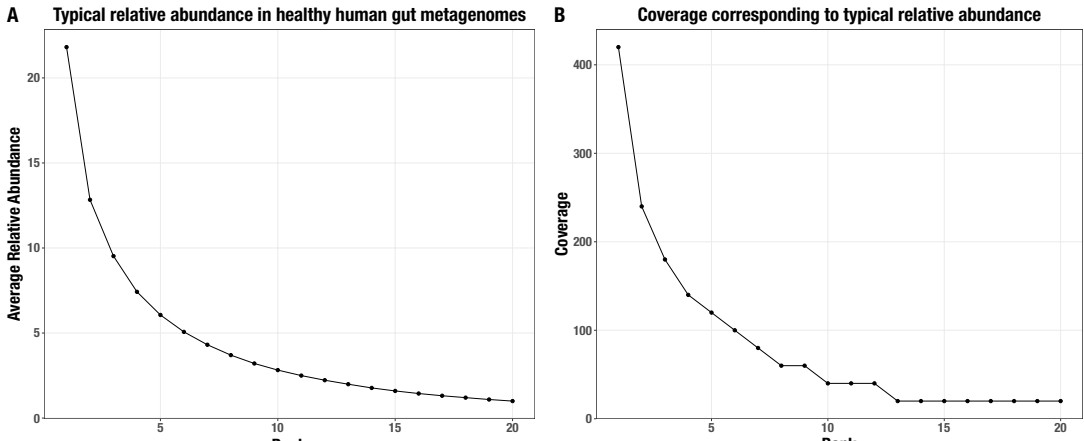

**Appendix 2—figure 7.** Normalized average relative abundance curve. (**A**) for the top 20 most abundant populations in a typical healthy human gut metagenome and (**B**) their corresponding coverage values in our synthetic metagenomes.

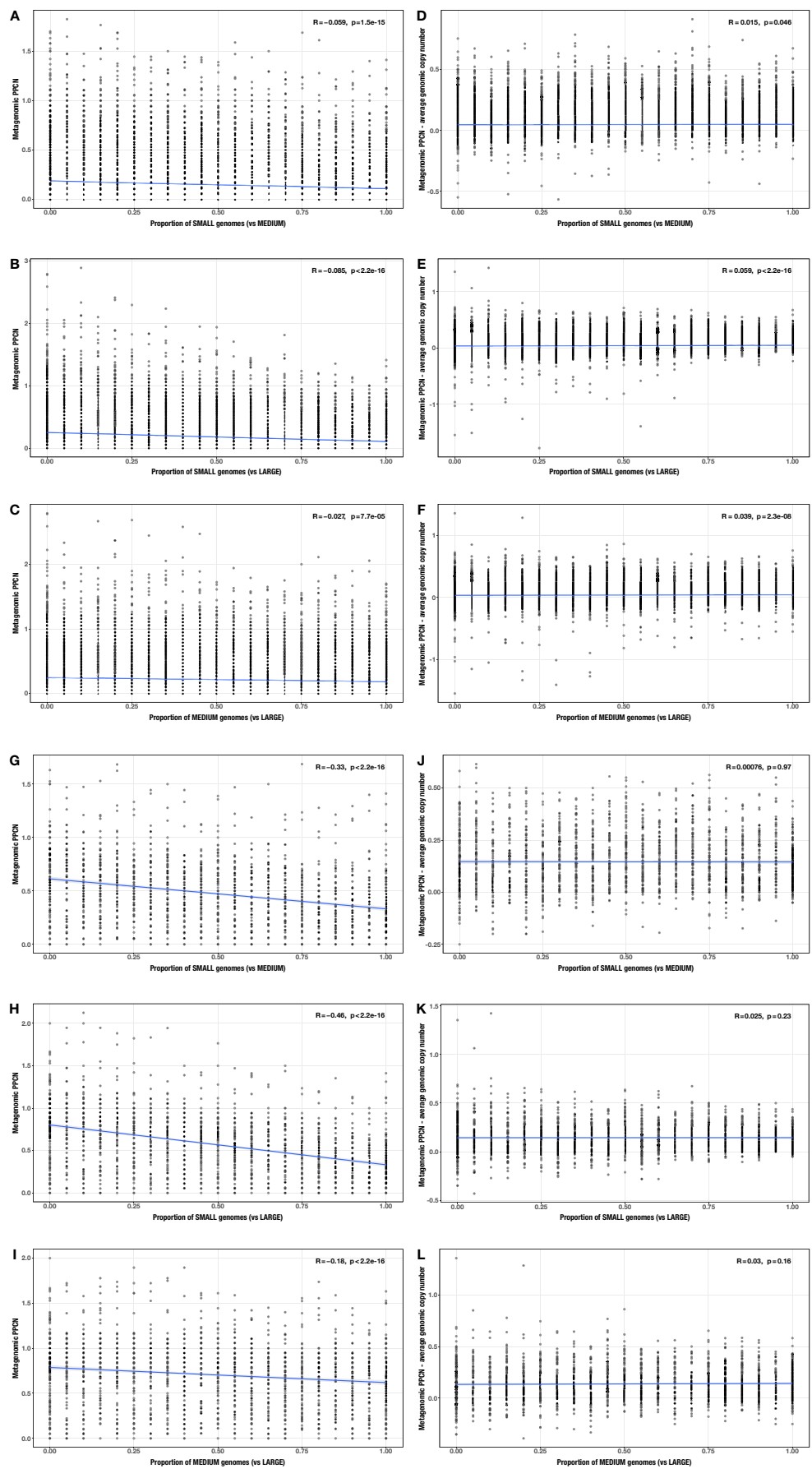

**Appendix 2—figure 8.** Correlations between proportion of genomes in smaller size category and (**A–C, G–I**) per-population copy number (PPCN) or (**D–F, J–L**) PPCN error relative to average genomic copy number for each size category pair across all modules (**A–F**) or the subset of inflammatory bowel disease (IBD)-enriched modules (**G–L**) in the realistic genome size test case. The Spearman's correlation coefficients and p-values are shown in the top-right corner of each plot, and regression lines are plotted in blue.

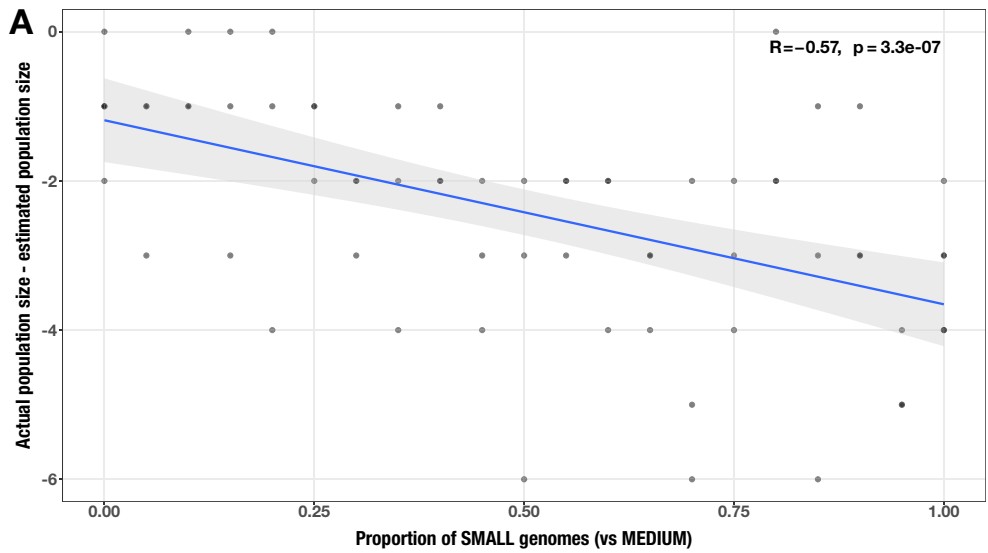

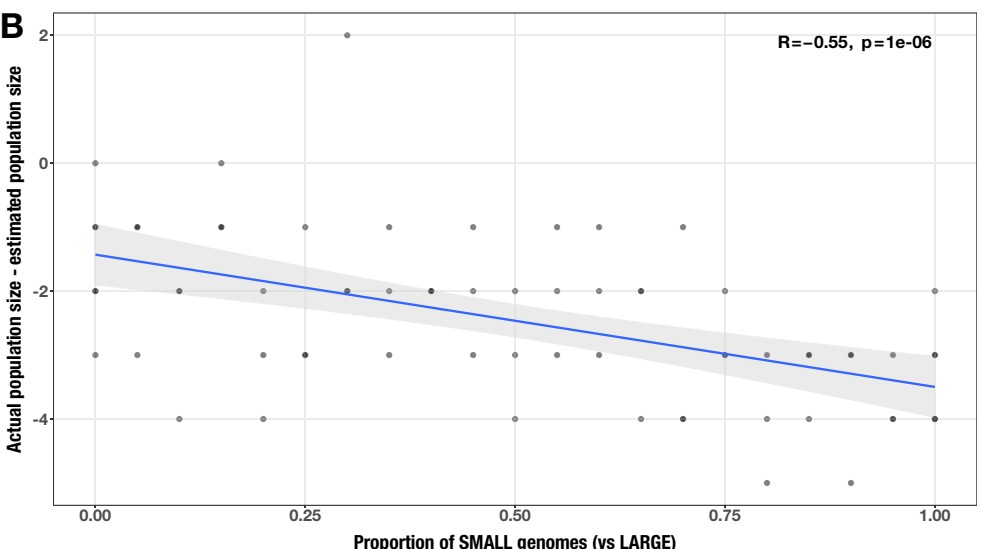

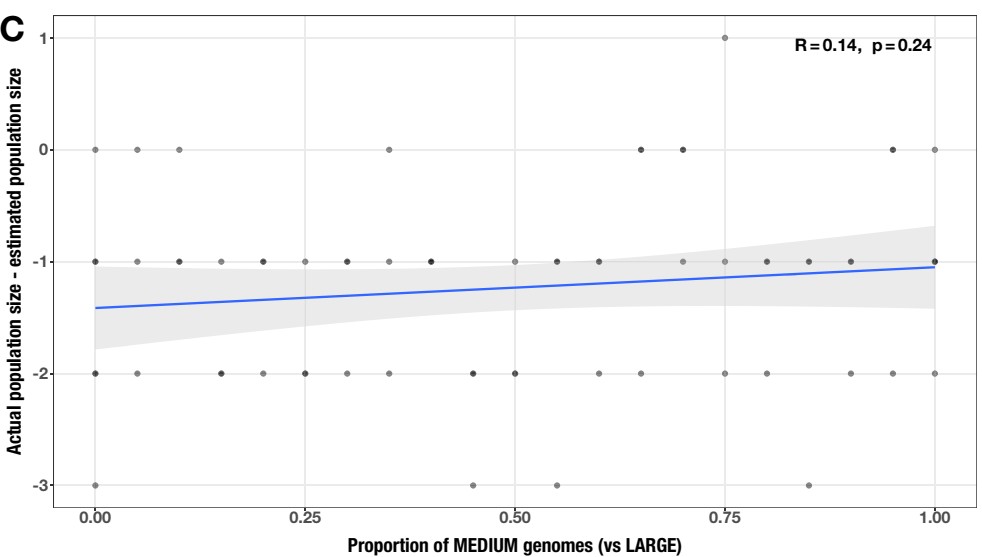

**Appendix 2—figure 9.** Correlations between proportion of genomes in smaller size category and error in community size estimate (relative to actual community size) for each size category pair. (**A**) small vs medium genomes; (**B**) small vs large genomes; (**C**) medium vs large genomes in the realistic genome size test case. The Spearman's correlation coefficients and p-values are shown in the top-right corner of each plot, and regression lines are plotted in blue.

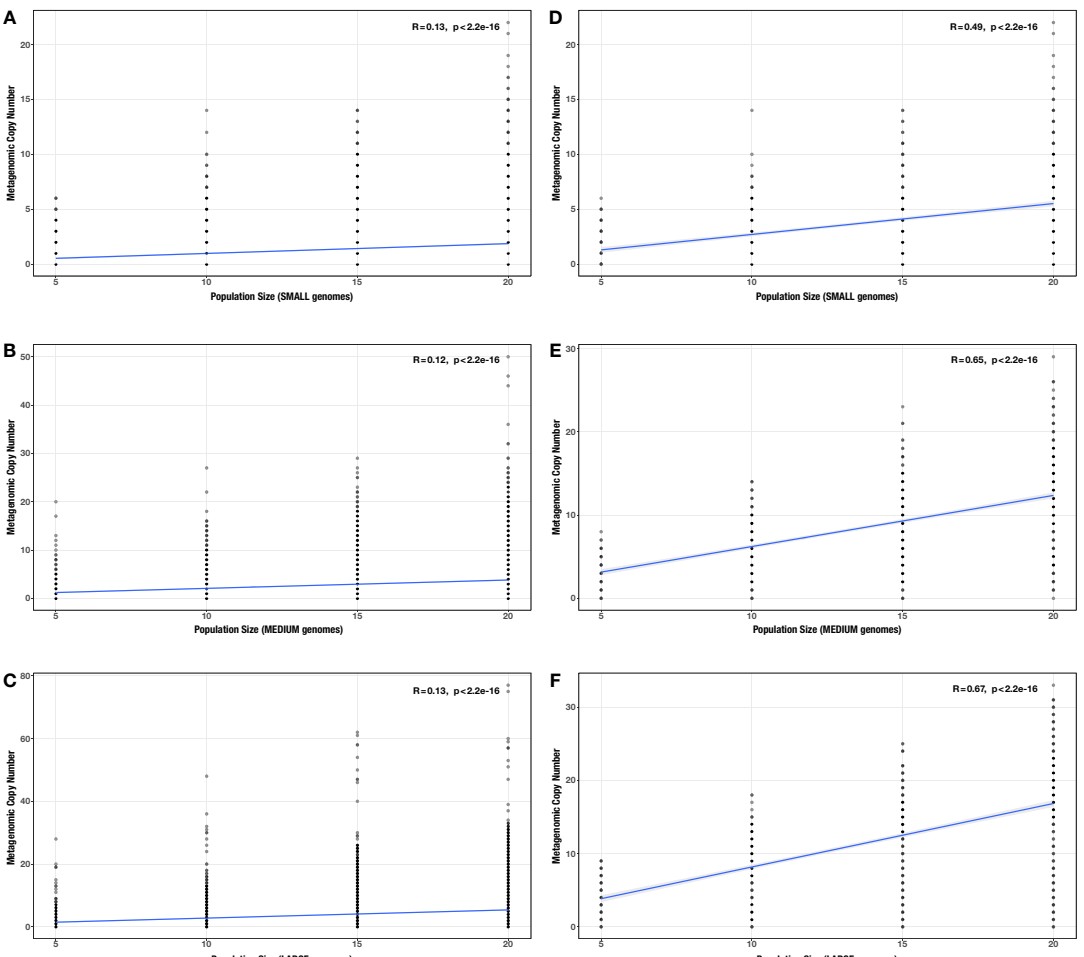

**Appendix 2—figure 10.** Correlation between community size and metagenomic copy number across all modules. (**A–C**) and across the subset of enriched modules (**D–F**) for each genome size category (**A/D**: small genomes; **B/E**: medium genomes; **C/F**: large genomes) in the community size test case. The Spearman's correlation coefficients and p-values are shown in the top-right corner of each plot, and regression lines are plotted in blue.

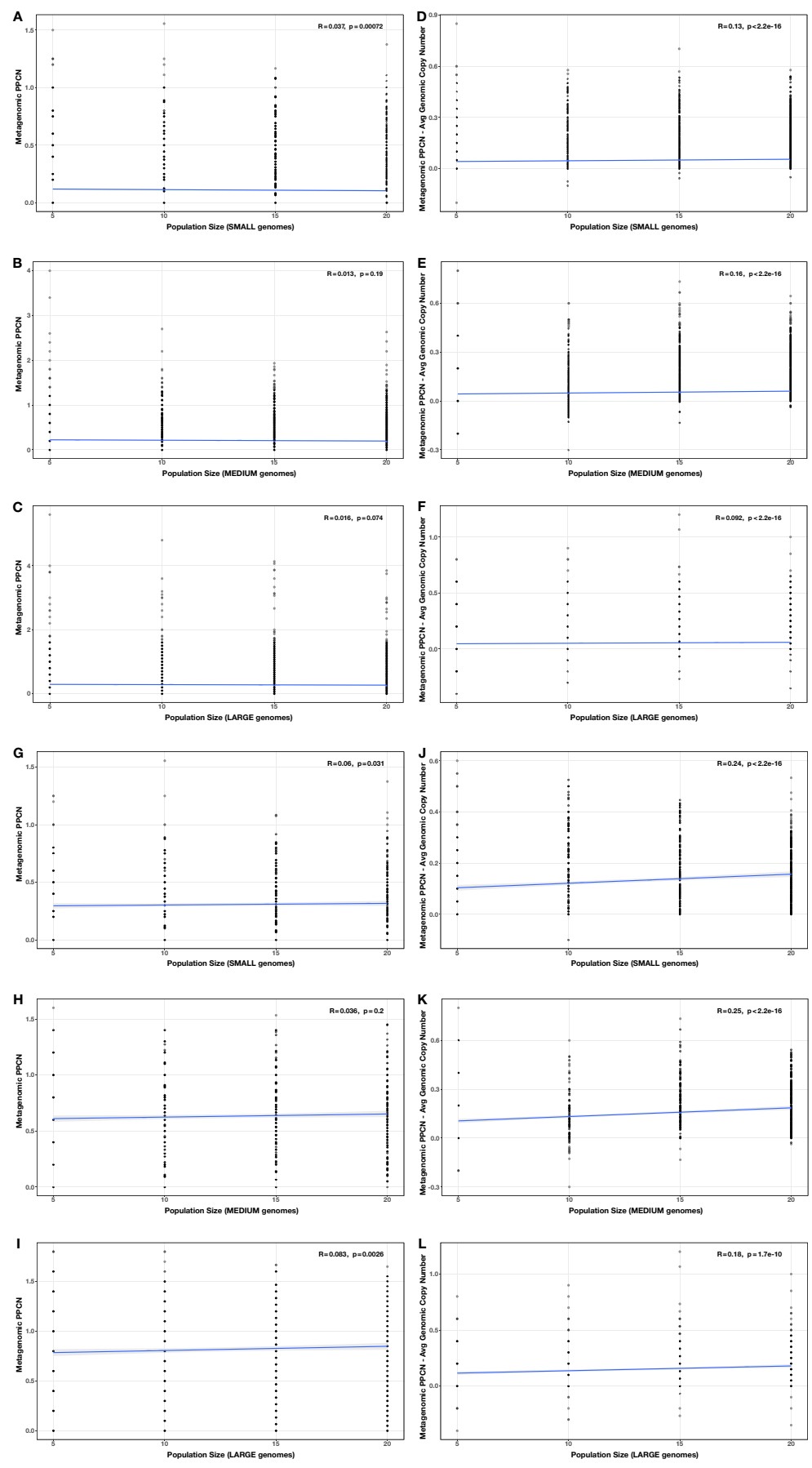

**Appendix 2—figure 11.** Correlations between community size and (**A–C, G–I**) per-population copy number (PPCN) or (**D–F, J–L**) PPCN error relative to average genomic copy number for each genome size category (**A/D/G/J**: small genomes; **B/E/H/K**: medium genomes; **C/F/I/L**: largegenomes), across all modules (**A–F**) or the subset of inflammatory bowel disease (IBD)-enriched modules (**G–L**) in the community size test case. The Spearman's correlation coefficients and p-values are shown in the top-right corner of each plot, and regression lines are plotted inblue.

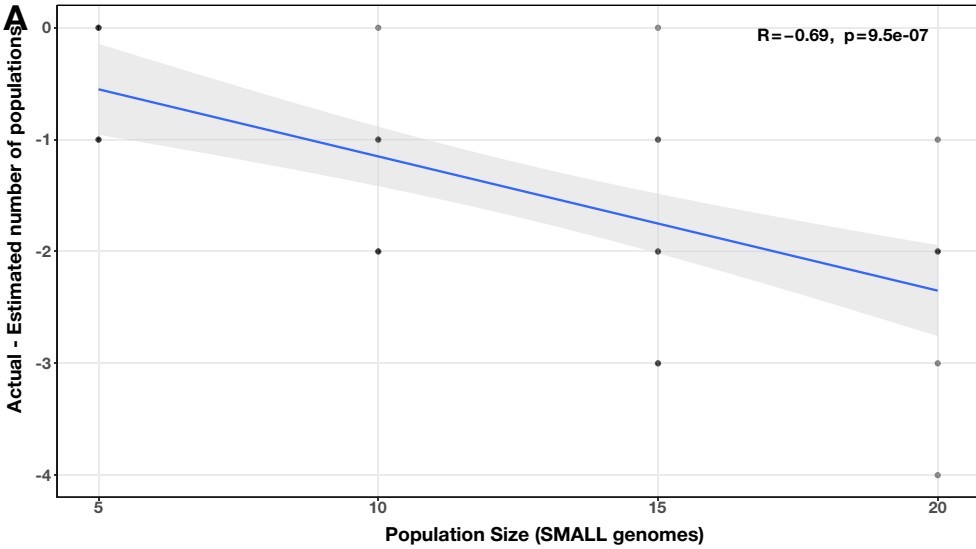

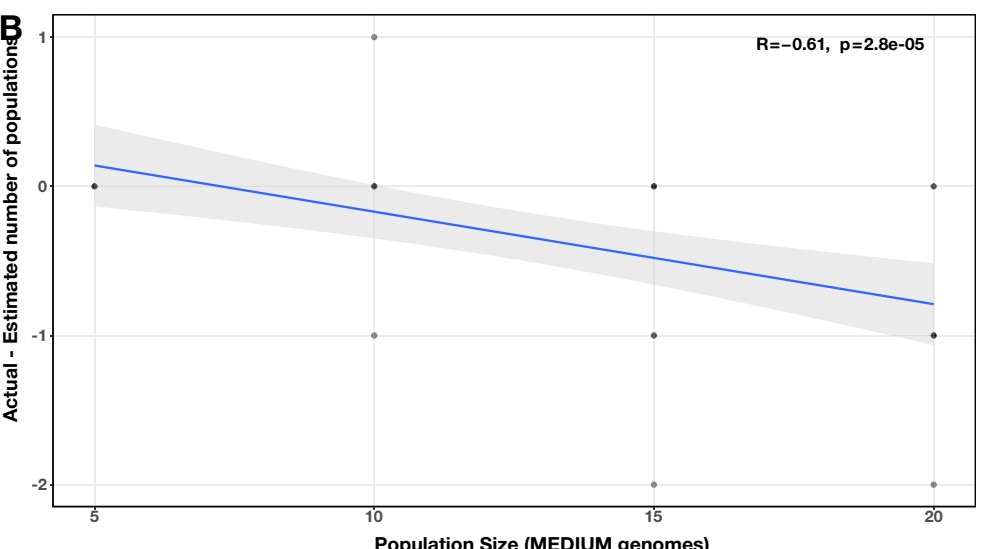

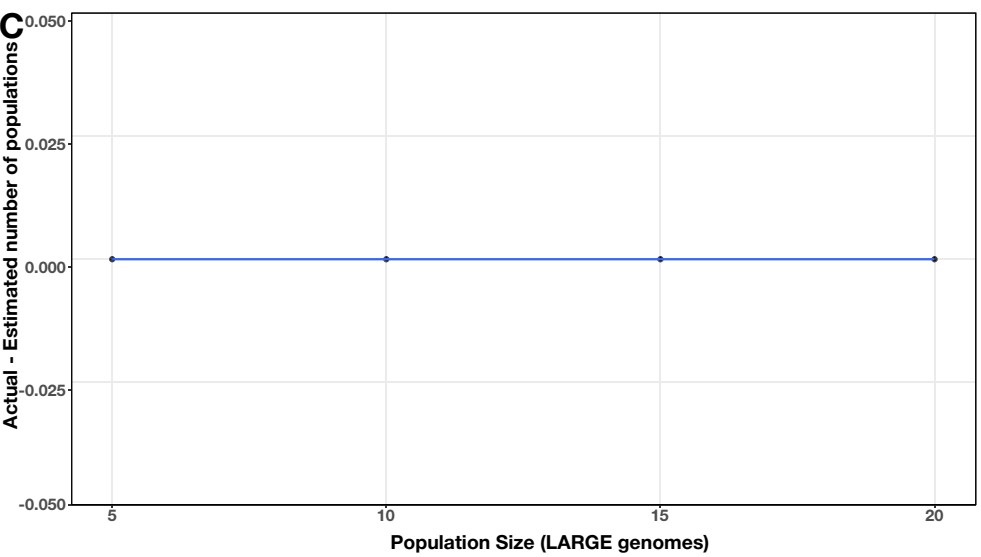

**Appendix 2—figure 12.** Correlations between community size and error in community size estimate (relative to actual community size) for each genome size category. (**A**) small; (**B**) medium; (**C**) large in the community size test case. The Spearman's correlation coefficients and p-values are shown in the top-right corner of each plot, and regression lines are plotted in blue.

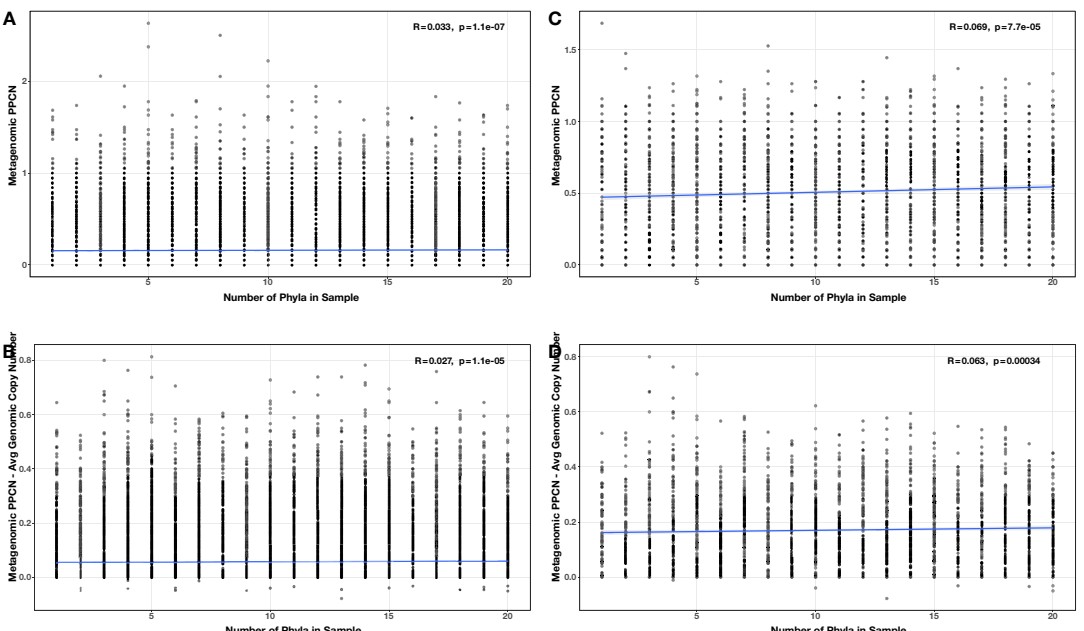

**Appendix 2—figure 13.** Correlations between number of phyla and (**A/C**) per-population copy number (PPCN) or PPCN accuracy relative to average genomic copy number (**B/D**), for all modules (**A/B**) or the subset of inflammatory bowel disease (IBD)-enriched modules (**C/D**) in the diversity test case. The Spearman's correlation coefficients and p-values are shown in the top-right corner of each plot, and regression lines are plotted in blue.

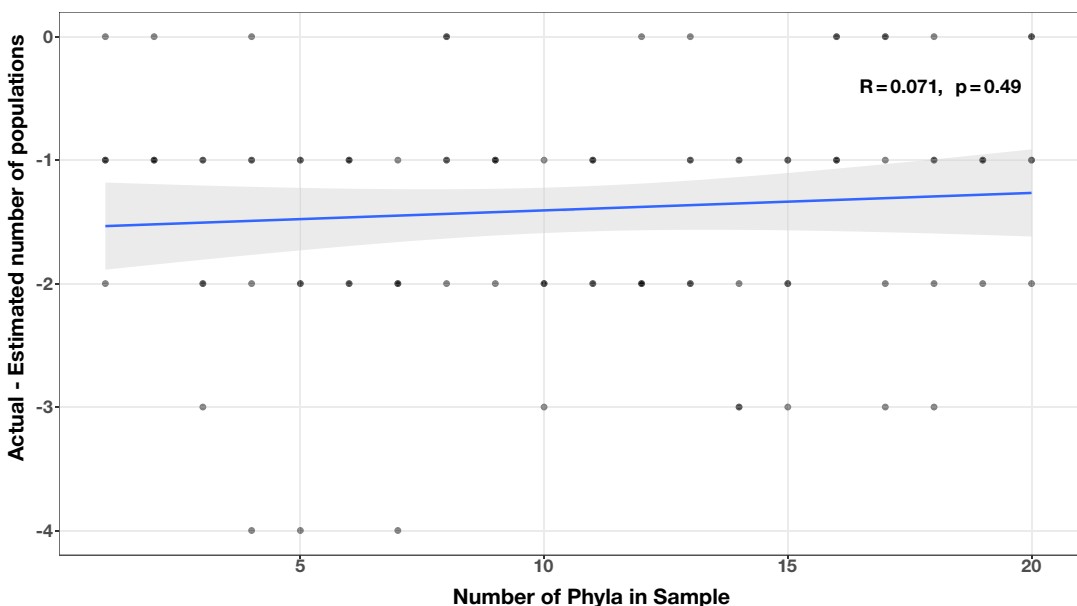

**Appendix 2—figure 14.** Correlation between number of phyla and accuracy of community size estimates (relative to actual community size) in the diversity test case. The Spearman's correlation coefficient and p-value are shown in the top-right corner of each plot, and the regression line is plotted in blue.

