## [Editor Report · eLife Assessment]

This study presents an **important** new bioinformatics tool for normalizing gene copy number from metagenomic assemblies and applies it to gain functional insights into the loss of microbial diversity during conditions of stress. The inclusion of extensive computational validation makes this a **compelling** study that raises intriguing new hypotheses regarding the impact of disease states on the gut microbiome. This paper will likely be of broad interest to researchers studying the role of complex microbial communities in host health and disease.

---

## [Referee Report · Reviewer #1 (Public review)]

In this work, Veseli et al. present a computational framework to infer the functional diversity of microbiomes in relation to microbial diversity directly from metagenomic data. The framework reconstructs metabolic modules form metagenomes and calculates the per-population copy number of each module, resulting in the proportion of microbes in the sample carrying certain genes. They applied this framework to a dataset of gut microbiomes from 109 inflammatory bowel disease (IBD) patients, 78 patients with other gastrointestinal conditions, and 229 healthy controls. The found that the microbiomes of IBD patients were enriched in a high fraction of metabolic pathways, including biosynthesis pathways such as those for amino acids, vitamins, nucleotides, and lipids. Hence, they had higher metabolic independence compared with healthy controls. To an extent, the authors also found a pathway enrichment suggesting higher metabolic independence in patients with gastrointestinal conditions other than IBD indicating this could be a signal for a general loss in host health. Finally, a machine learning classifier using high metabolic independence in microbiomes could predict IBD with good accuracy. Overall, this is an interesting and well-written article and presents a novel workflow that enables a comprehensive characterization of microbiome cohorts.

Comments on revisions:

I believe that after the second round of revisions, the Reviewers sufficiently addressed the comments and improved the manuscript. Open questions have been answered. I have no further comments.

---

## [Referee Report · Reviewer #2 (Public review)]

This study builds upon the team's recent discovery that antibiotic treatment and other disturbances favours the persistence of bacteria with genomes that encode complete modules for the synthesis of essential metabolites (Watson et al. 2023). Veseli and collaborators now provide an in-depth analysis of metabolic pathway completeness within microbiomes, finding strong evidence for an enrichment of bacteria with high metabolic independence in the microbiomes associated with IBD and other gastrointestinal disorders. Importantly, this study provides a new open-source software to facilitate the reconstruction of metabolic pathways, estimate their completeness and normalize their results according to species diversity. Finally, this study also shows that metabolic independence of microbial communities can be used as a marker of dysbiosis. The function-based health index proposed here is more robust to individual's lifestyles and geographic origin than previously proposed methods based on bacterial taxonomy.

The implications of this study have the potential to spur a paradigm shift in the field. It shows that certain bacterial taxa that have been consistently associated with disease might not be harmful to their host as previously thought. These bacteria seem to be the only species that are able to survive in a stressed gut environment. They might even be important to rebuild a healthy microbiome (although the authors are careful in not making this speculation).

This paper provides an in-depth discussion of the results, and limitations are clearly addressed throughout the manuscript (see also the supplementary files for an in-depth assessment of the robustness of the methods). Some of the potential limitations relate to the use of large publicly available datasets, where sample processing and the definition of healthy status varies between studies. The authors have recognised these issues and their results were robust to analyses performed at a per-cohort basis. The potential limitations therefore are unlikely to have affected the conclusions of this study.

Overall, this is manuscript is a magnificent contribution to the field, likely to inspire many other studies to come.

Comments on revisions:

The authors have performed a detailed assessment of the accuracy and robustness of their new methods, and included an informative session comparing their new approach with existing ones. The new analyses have strengthened the manuscript, and the results support the biological interpretations of the study.

I commend the authors for the effort and the excellent research.

---

## [Referee Report · Reviewer #3 (Public review)]

The major strength of this manuscript is the "anvi-estimate-metabolism' tool, which is already accessible online, extensively documented, and potentially broadly useful to microbial ecologists. Inclusion of extensive benchmarking and validation on simulated metagenomes has further increased confidence in this approach. Further, the conceptual insights raise interesting hypotheses that could be pursued in follow-on experimental work.

Comments on revisions:

Thank you for the very thorough response and congratulations!

---

## [Author Response]

The following is the authors’ response to the original reviews.

**Response to Public Reviewer Comments:**

**Reviewer 1:**
In this work, Veseli et al. present a computational framework to infer the functional diversity of microbiomes in relation to microbial diversity directly from metagenomic data. The framework reconstructs metabolic modules from metagenomes and calculates the per-population copy number of each module, resulting in the proportion of microbes in the sample carrying certain genes. They applied this framework to a dataset of gut microbiomes from 109 inflammatory bowel disease (IBD) patients, 78 patients with other gastrointestinal conditions, and 229 healthy controls. They found that the microbiomes of IBD patients were enriched in a high fraction of metabolic pathways, including biosynthesis pathways such as those for amino acids, vitamins, nucleotides, and lipids. Hence, they had higher metabolic independence compared with healthy controls. To an extent, the authors also found a pathway enrichment suggesting higher metabolic independence in patients with gastrointestinal conditions other than IBD indicating this could be a signal for a general loss in host health. Finally, a machine learning classifier using high metabolic independence in microbiomes could predict IBD with good accuracy. Overall, this is an interesting and well-written article and presents a novel workflow that enables a comprehensive characterization of microbiome cohorts.

We thank the reviewer for their interest in our study, their summary of its findings, and their kind words about the manuscript quality.

**Reviewer 2:**
This study builds upon the team's recent discovery that antibiotic treatment and other disturbances favour the persistence of bacteria with genomes that encode complete modules for the synthesis of essential metabolites (Watson et al. 2023). Veseli and collaborators now provide an in-depth analysis of metabolic pathway completeness within microbiomes, finding strong evidence for an enrichment of bacteria with high metabolic independence in the microbiomes associated with IBD and other gastrointestinal disorders. Importantly, this study provides new open-source software to facilitate the reconstruction of metabolic pathways, estimate their completeness and normalize their results according to species diversity. Finally, this study also shows that the metabolic independence of microbial communities can be used as a marker of dysbiosis. The function-based health index proposed here is more robust to individuals' lifestyles and geographic origin than previously proposed methods based on bacterial taxonomy.The implications of this study have the potential to spur a paradigm shift in the field. It shows that certain bacterial taxa that have been consistently associated with disease might not be harmful to their host as previously thought. These bacteria seem to be the only species that are able to survive in a stressed gut environment. They might even be important to rebuild a healthy microbiome (although the authors are careful not to make this speculation).This paper provides an in-depth discussion of the results, and limitations are clearly addressed throughout the manuscript. Some of the potential limitations relate to the use of large publicly available datasets, where sample processing and the definition of healthy status varies between studies. The authors have recognised these issues and their results were robust to analyses performed on a per-cohort basis. These potential limitations, therefore, are unlikely to have affected the conclusions of this study.Overall, this manuscript is a magnificent contribution to the field, likely to inspire many other studies to come.

We thank the reviewer for their endorsement of our study and their precision regarding the evaluation of its strengths. We also appreciate their high expectations for its impact in the field.

**Reviewer 3:**
The major strength of this manuscript is the "anvi-estimate-metabolism' tool, which is already accessible online, extensively documented, and potentially broadly useful to microbial ecologists.

We thank the reviewer for their recognition of the computational advances in this study. We also thank the reviewer for their suggestions that we have addressed below, which allowed us to strengthen our manuscript.

However, the context for this tool and its validation is lacking in the current version of the manuscript. It is unclear whether similar tools exist; if so, it would help to benchmark this new tool against prior methods.

The reviewer brings up a very good point about the lack of context for the `anvi-estimate-metabolism` program. While our efforts that led to the emergence of this software included detailed benchmarking efforts, a formal assessment of its performance and accuracy was indeed lacking. We are thankful for our reviewer to point this out, which motivated us to perform additional analyses to address such concerns. Our revision contains a new, 34-page long supplementary information file (Supplementary File 2) that includes a section titled “Comparison of anvi-estimate-metabolism to existing tools for metabolism reconstruction”. The text therein describes the landscape of currently available software for metabolism reconstruction and describes the features that make `anvi-estimate-metabolism` unique – namely, (1) its implementation of metrics that make it suitable for metagenome-level analyses (i.e., pathway copy number and stepwise interpretation of pathway definitions) and (2) its ability to process user-defined metabolic pathways rather than exclusively relying on KEGG. As described in that section, there is currently no other tool that can compute copy numbers of metabolic pathways from metagenomic data. Hence, it is not quite possible to benchmark the copy number methodology used in our study against prior methods; however, our benchmarking of this functionality with synthetic genomes and metagenomes (described later in this document) does provide necessary quantitative insights into its accuracy and efficiency.

While comparison of the copy number calculations to other tools was not possible due to the unique nature of this functionality, it was possible to benchmark our gene function annotation methodology against existing tools that also annotate genes with KEGG KOfams, which is a step commonly used by various tools that aim to estimate metabolic potential in genomes and metagenomes. In the anvi’o software ecosystem the annotation of genes for metabolic reconstruction is implemented in `anvi-run-kegg-kofams`, and represents a step that is required by `anvi-estimate-metabolism`. As our comparisons were quite extensive and involved additional researchers, we described them in another study which we titled “Adaptive adjustment of significance thresholds produces large gains in microbial gene annotations and metabolic insights” (doi:10.1101/2024.07.03.601779) that is now cited from within our revision in the appropriate context. Briefly, our comparison of anvi’o, Kofamscan, and MicrobeAnnotator using 396 publicly-available bacterial genomes from 11 families demonstrated that `anvi-run-kegg-kofams` is able to identify an average of 12.8% more KO annotations per genome than the other tools, especially in families commonly found in the gut environment (Figure 1). Furthermore, anvi’o recovered the highest proportion of annotations that were independently validated using eggNOG-mapper. Our comparisons also showed that annotations from anvi’o yield at least 11.6% more complete metabolic modules than Kofamscan or MicrobeAnnotator, including the identification of butyrate biosynthesis in *Lachnospiraceae* genomes at rates similar to manual identification of this pathway in this clade (Figure 2a). Overall, our findings that are now described extensively in DOI:10.1101/2024.07.03.601779 show that our method captures high-quality annotations for accurate downstream metabolism estimates.

We hope these new data help increase the reviewer’s confidence in our results.

Simulated datasets could be used to validate the approach and test its robustness to different levels of bacterial richness, genome sizes, and annotation level.

We thank the reviewer for this suggestion. It was an extremely useful exercise that not only helped us elucidate the nuances of our approach, but also enabled us to further highlight its strengths in our manuscript. We created simulated datasets including a total of 409 synthetic metagenomes that we used to test the robustness of our approach to different genome sizes, community sizes, and levels of diversity. Overall, our tests with these synthetic metagenomes demonstrated that our approach of computing PPCN values to summarize the metabolic capacity within a metagenomic community is accurate and robust to differences in all three critical variables. Most of these variables were weakly correlated between PPCN or PPCN accuracy, and the few correlations that were stronger in fact further supported our original hypothesis that we generated from our comparisons of healthy and IBD gut metagenomes. The methods and results of our validation efforts are explained in detail in our new Supplementary File 2 (see the section titled “Validation of per-population copy number (PPCN) approach on simulated metagenomic data”), but we copy here the subsection that summarizes our findings for the reviewer’s convenience:

Overall impact on the comparison between healthy and IBD gut metagenomes

“In summary, our validation strategy revealed good accuracy at estimating metagenome-level metabolic capacity relative to our genome-level knowledge in the simulated data. While it often underestimated average genomic completeness by ignoring partial copies of metabolic pathways and often overestimated average genomic copy number due to the effect of pathway complementarity between different community members, the magnitude of error was overall limited in range and the error distributions were centered at or near 0. Furthermore, we observed these broad error trends in all cases we tested, and therefore we expect that they would also apply to both sample groups in our comparative analysis. Thus, we next considered how the PPCN approach might have influenced our analyses that considered metagenomes from healthy individuals and from those who have IBD – two groups that differed from one another with respect to some of the variables considered in our tests.

Most of the correlations between PPCN or PPCN accuracy and sample parameters were weak, yet significant (Table 1). They showed that community size and diversity level have limited influence on the PPCN calculation, while genome size does not influence its accuracy. The only exception was the moderate correlation between PPCN and genome size, particularly for the subset of IBD-enriched pathways. It was a negative correlation with the proportion of small genomes in a metagenome, indicating that PPCN values for these pathways are larger when there are more large genomes in the community and suggesting that these pathways tend to occur frequently in larger genomes. This is in line with our observation that IBD communities contain more large genomes and therefore confirms our interpretation that the populations surviving in the IBD gut microbiome are those with the genomic space to encode more metabolic capacities.

If we consider even the weak correlations, two of those relationships indicate that our approach would be more accurate for IBD metagenomes than for healthy metagenomes. For instance, PPCN accuracy was slightly higher for smaller communities (as in IBD samples), with a weakly positive correlation between PPCN error and community size. It was also slightly more accurate for less diverse communities (as in IBD samples), with a weakly positive correlation between PPCN error and number of phyla. The only opposing trend was the weakly positive correlation between PPCN error and proportion of smaller genomes, which favors higher accuracy in communities with smaller genomes (as in healthy samples). Given that our analysis focuses on the pathways enriched in IBD samples, an overall higher accuracy in IBD samples would increase the confidence in our enrichment results.

We also examined the accuracy of our method to predict the number of populations within a metagenome based on the distribution and frequency of single-copy core genes (i.e., the denominator in the calculation of PPCN). Our benchmarks show that the estimates are overall accurate, where most errors reflect a negligible amount of underestimations of the actual number of populations. Errors occurred more frequently for the realistic synthetic assemblies generated from simulated short read data than for the ideal synthetic assemblies generated from the combination of genomic contigs. The correlations between estimation accuracy and sample parameters indicated that the population estimates are more accurate for smaller communities and communities with more large genomes, as in IBD samples (Table 2). Thus, this method is more likely to underestimate the community size in healthy samples, and these errors could lead to overestimation of PPCN in healthy samples relative to IBD samples. Thus, the enrichment of a given pathway in the IBD samples would have to overcome its relative overestimation in the healthy sample group, making it more likely that we identified pathways that were truly enriched in the IBD communities.

Overall, the consideration of our simulations in the context of healthy vs IBD metagenomes suggest that slight biases in our estimates as a function of unequal diversity with sample groups should have driven PPCN calculations towards a conclusion that is opposite of our observations under neutral conditions. Thus, clear differences between healthy vs IBD metagenomes that overcome these biases suggest that biology, and not potential bioinformatics artifacts, is the primary driver of our observations.”

Accordingly, we have added the following sentence summarizing the validation results to our paper:

“Our validation of this method on simulated metagenomic data demonstrated that it is accurate in capturing metagenome-level metabolic capacity relative to genome-level metabolic capacity estimated from the same data (Supplementary File 2, Supplementary Table 6).”

Early in this process of validation, we identified and fixed two minor bugs in our codebase. The bugs did not affect the results of our paper and therefore did not warrant a re-analysis of our data. The first bug, which is detailed in the Github issue https://github.com/merenlab/anvio/issues/2231 and fixed in the pull request https://github.com/merenlab/anvio/pull/2235, led to the overestimation of the number of microbial populations in a metagenome when the metagenome contains both Bacteria and Archaea. None of the gut metagenomes analyzed in our paper contained archaeal populations, so this bug did not affect our community size estimates.

The second bug, which is detailed in the Github issue https://github.com/merenlab/anvio/issues/2217 and fixed in the pull request https://github.com/merenlab/anvio/pull/2218, caused inflation of stepwise copy numbers for a specific type of metabolic pathway in which the definition contained an inner parenthetical clause. This bug affected only 3 pathways in the KEGG MODULE database we used for our analysis, M00083, M00144, and M00149. It is worth noting that one of those pathways, M00083, was identified as an IBD-enriched module in our analysis. However, the copy number inflation resulting from this bug would have occurred equivalently in both the healthy and IBD sample groups and thus should not have impacted our comparative analysis.

Regardless, we are grateful for the suggestion to validate our approach since it enabled us to identify and eliminate these minor issues.

The concept of metabolic independence was intriguing, although it also raises some concerns about the overinterpretation of metagenomic data. As mentioned by the authors, IBD is associated with taxonomic shifts that could confound the copy number estimates that are the primary focus of this analysis. It is unclear if the current results can be explained by IBD-associated shifts in taxonomic composition and/or average genome size. The level of prior knowledge varies a lot between taxa; especially for the IBD-associated gamma-Proteobacteria.

The reviewer brings up an important point, and we are thankful for the opportunity to clarify the impact of taxonomy on our analysis. Though IBD has been associated with taxonomic shifts in the gut microbiome, a major problem with such associations is that the taxonomic signal is extremely variable, leading to inconsistency in the observed shifts across different studies (doi:https://doi.org/10.3390/pathogens8030126). Indeed, one of the most comprehensive prior studies into this topic demonstrated that inter-individual variation is the largest contributor to all multi-omic measurements aiming to differentiate between the gut microbiome of individuals with IBD from that of healthy individuals, including taxonomy (doi:10.1038/s41586-019-1237-9). We therefore took a different approach to study this question that is independent of taxonomy, by focusing on metabolic potential estimated directly from metagenomes to elucidate an ecological explanation behind the reduced diversity of the IBD gut microbiome, which studies of taxonomic composition alone are not able to provide. Furthermore, the variability inherent to taxonomic profiles of the gut microbiome makes it unlikely that taxonomic shifts could confound our analysis, especially given our large sample set encompassing a variety of individuals with different origins, ages, and genders.

We agree with the reviewer that our level of prior knowledge varies substantially across taxa. Regardless, the only prior knowledge with any bearing on our ability to estimate metabolic capacity in a taxonomy-independent manner is the extent of sequence diversity captured by our annotation models for the enzymes used in metabolic pathways. During our analysis, we had observed that metagenomes in the healthy group had fewer gene annotations than those in the IBD group and we therefore shared the reviewer’s concern about potential annotation bias, whereby less-studied genomes are not always incorporated into the Hidden Markov Models for annotating KEGG Orthologs, perhaps making it more likely for us to miss annotations in these genomes (and leading to lower completeness scores for metabolic pathways in the healthy samples). Our annotation method partially addresses this limitation by taking a second look at any unannotated genes and mindfully relaxing the bit score similarity thresholds to capture annotations for any genes that are slightly too different from reference sequences for annotation with default thresholds. As mentioned previously, our recent preprint demonstrates the efficacy of this strategy (doi:10.1101/2024.07.03.601779). To further address this concern, we also investigated the extent of distant homology in these metagenomes using AGNOSTOS (doi:https://doi.org/10.7554/eLife.67667), which showed a higher proportion of unknown genes in the healthy metagenomes and suggested that a substantial portion of the unannotated genes are not distant homologs of known enzymes that we failed to annotate due to lack of prior knowledge about them, but rather are completely novel functions. To describe these results, we added the following paragraph and two accompanying figures (Supplementary Figure 4g-h) to the section “Differential annotation efficiency between IBD and Healthy samples” in Supplementary File 1:

“To understand the potential origins of the reduced annotation rate in healthy metagenomes, we ran AGNOSTOS (Vanni et al. 2022) to classify known and unknown genes within the healthy and IBD sample groups. AGNOSTOS clusters genes to contextualize them within an extensive reference dataset and then categorizes each gene as ‘known’ (has homology to genes annotated with Pfam domains of known function), ‘genomic unknown’ (has homology to genes in genomic reference databases that do not have known functional domains), or ‘environmental unknown’ (has homology to genes from metagenomes or MAGs that do not have known functional domains). The resulting classifications confirm that healthy metagenomes contain fewer ‘known’ genes than metagenomes in the IBD sample group – the proportion of ‘known’ genes classified by AGNOSTOS is about 3.0% less in the healthy metagenomes than in the IBD sample group, which is similar to the ~3.5% decrease in the proportion of ‘unannotated’ genes observed by simply counting the number of genes with at least one functional annotation (Supplementary Figure 4g-h, Supplementary Table 1e). Furthermore, the majority of the unannotated genes in either sample group were categorized by AGNOSTOS as ‘genomic unknown’ (Supplementary Figure 4g), suggesting that the unannotated sequences are genes without biochemically-characterized functions currently associated with them and are thus legitimately lacking a functional annotation in our analysis, rather than representing distant homologs of known protein families that we failed to annotate. Based upon the classifications, a systematic technical bias is unlikely driving the annotation discrepancy between the sample groups.”

Furthermore, we have already discussed this limitation and its implications in our manuscript (see section “Key biosynthetic pathways are enriched in microbial populations from IBD samples”). To further clarify that our approach is independent of taxonomy, we have now also amended the following statement in our introduction:

“Here we implemented a high-throughput, taxonomy-independent strategy to estimate metabolic capabilities of microbial communities directly from metagenomes and investigate whether the enrichment of populations with high metabolic independence predicts IBD in the human gut.”

Finally, the reviewer is also correct that genome size is a part of the equation, as genome size and level of metabolic capacity are inextricable. In fact, we observed this in our analysis, as already stated in our paper:

“HMI genomes were on average substantially larger (3.8 Mbp) than non-HMI genomes (2.9 Mbp) and encoded more genes (3,634 vs. 2,683 genes, respectively)”

Since larger genomes have the space to encode more functional capacity, it follows that having higher metabolic independence would require a microbe to have a larger genome. The validation of our method on simulated metagenomic data supported this idea by demonstrating that the IBD-enriched metabolic pathways are commonly identified in large genomes. The validation also proved that genome size does not influence the accuracy of our approach (Supplementary File 2).

It can be difficult to distinguish genes for biosynthesis and catabolism just from the KEGG module names and the new normalization tool proposed herein markedly affects the results relative to more traditional analyses.

We agree with the reviewer that KEGG module names do not clearly indicate the presence of biosynthetic genes of interest. That said, KEGG is a commonly-used and extensively-curated resource, and many biologists (including ourselves) trust their categorization of genes into pathways. We hope that readers who are interested in specific genes within our results would make use of our publicly-available datasets (which include gene annotations) to conduct a targeted analysis based on their expertise and research question.

However, we would like to respectfully note that the ability to distinguish the genes within each KEGG module may not be very useful to most readers, and is unlikely to have a meaningful impact in our findings. As the reviewer most likely appreciates, the presence of individual genes in isolation can be insufficient to indicate biosynthetic capacity, considering that (1) most biosynthetic pathways involve several biochemical conversions requiring a series of enzymes, (2) enzymes are often multi-functional rather than exclusive to one pathway, and (3) different organisms in a community may utilize enzymes encoded by different genes to perform the same or similar biochemical reaction in a pathway. We therefore made the choice to analyze metabolic capacity at the pathway level, because this would better reflect the biosynthetic abilities encoded by the multiple microbial populations within each metagenome.

The reviewer also suggests that our novel normalization method affects our results, yet we believe that this normalization strategy is one of the strengths of our study in comparison to ‘more traditional analyses’ as it enables an appropriate comparison between metagenomes describing microbial communities of dramatically different degrees of richness. Indeed, we suspect that the lack of normalization in more traditional analyses may be one reason why prior analyses have so far failed to uncover any mechanistic explanation for the loss of diversity in the IBD gut microbiome. We hope that our validation efforts were sufficiently convincing in demonstrating the suitability of our approach, and copy here a particularly illuminating section of the validation results that we have added to Supplementary Information File 2:

“As expected, we observed a significant positive correlation between metagenomic copy number (the numerator of PPCN) and community size in each group, likely driven by the increase in the copy number of core metabolic pathways in larger communities (Supplementary Figure 18). Interestingly, this correlation was much stronger for the subset of IBD-enriched pathways (0.49 <= R <= 0.67) than for all modules (0.12 <= R <= 0.13).

“However, the correlation was much weaker and often nonsignificant for the normalized PPCN data in both groups of modules (all modules: 0.01 < R < 0.04, enriched modules: 0.04 < R < 0.09, Supplementary Table 6b, Supplementary Figure 19), which demonstrates the suitability of our normalization method to remove the effect of community size in comparisons of metagenome-level metabolic capacity.”

As such, it seems safer to view the current analysis as hypothesis-generating, requiring additional data to assess the degree to which metabolic dependencies are linked to IBD.

We certainly agree with the reviewer that our study, similar to the vast majority of studies published every year, is a hypothesis-generating work. Any idea proposed in any scientific study in life sciences will certainly benefit from additional data analyses, and therefore we respectfully do not accept this as a valid criticism of our work. The inception of this study is linked to an earlier work that hypothesized high metabolic independence as a determinant of microbial fitness in stressed gut communities (doi:10.1186/s13059-023-02924-x), which lacked validation on larger sets of data. Our study tests this original hypothesis using a large number of metagenomes, and lends further support for it with approaches that are now better validated. Furthermore, there are other studies that agree with our interpretation of the data (doi:10.1101/2023.02.17.528570, doi:10.1038/s41540-021-00178-6), and we look forward to more computational and/or experimental work in the future to generate more evidence to evaluate these insights further.

**Response to Recommendations for the Authors**

**Reviewer 1:**
My main comments include:- From the results reported in lines 178-185, it seems that metabolic pathways in general were enriched in IBD microbiomes, not specifically biosynthetic pathways. Can we really say then that the signal is specific for biosynthesis capabilities?

We apologize for the confusion here. When we read the text again, we ourselves were confused with our phrasing.

The reviewer is correct that a similar proportion of both biosynthetic and non-biosynthetic pathways had elevated per-population copy number (PPCN) values in the IBD samples. However, the low microbial diversity associated with IBD and the on average larger genome size of individual populations contributes to this relative enrichment of the majority of metabolic modules. To remove this bias and identify specific modules whose enrichment was highly conserved across microbial populations associated with IBD, we implemented two criteria: (1) we selected modules that passed a high statistical significance threshold in our enrichment test (Wilcoxon Rank Sum Test, FDR-adjusted p-value < 2e-10), and (2) we accounted for effect size by ranking these modules according to the difference between their median PPCN in IBD samples and their median PPCN in healthy samples, and keeping only those in the top 50% (which translated to an effect size threshold of > 0.12).

This analysis revealed a set of metabolic modules that were consistently and highly significantly enriched in microbial communities associated with IBD. The majority of these metabolic modules encode biosynthesis pathways. Our use of the terms “elevated”, “enriched”, and “significantly enriched” in the previous version of the text was confusing to the reader. We thank the reviewer for pointing this out, and we hope that our revision of the text clarifies the analysis strategy and observations:

“To gain insight into potential metabolic determinants of microbial survival in the IBD gut environment, we assessed the distribution of metabolic modules within samples from each group (IBD and healthy) with and without using PPCN normalization. Without normalizing, module copy numbers were overall higher in healthy samples (Figure 2a) and modules exhibited weak differential occurrence between cohorts (Figure 2b, 2c, Supplementary Figure 3). The application of PPCN reversed this trend, and most metabolic modules were elevated in IBD (Supplementary Figure 5). This observation is influenced by two independent aspects of the healthy and IBD microbiota. The first one is the increased representation of microbial organisms with smaller genomes in healthy individuals (Watson et al. 2023), which increases the likelihood that the overall copy number of a given metabolic module is below the actual number of populations. In contrast, one of the hallmarks of the IBD microbiota is the generally increased representation of organisms with larger genomes (Watson et al. 2023). The second aspect is that the generally higher diversity of microbes in healthy individuals increases the denominator of the PPCN. This results in a greater reduction in the PPCN of metabolic modules that are not shared across all members of the diverse gut microbial populations in health.

To go beyond this general trend and identify modules that were highly conserved in the IBD group, we first selected those that passed a relatively high statistical significance threshold in our enrichment test (Wilcoxon Rank Sum Test, FDR-adjusted p-value < 2e-10). We then accounted for effect size by ranking these modules according to the difference between their median PPCN in IBD samples and their median PPCN in healthy samples, and keeping only those in the top 50% (which translated to an effect size threshold of > 0.12). This stringent filtering revealed a set of 33 metabolic modules that were significantly enriched in metagenomes obtained from individuals diagnosed with IBD (Figure 2d, 2e), 17 of which matched the modules that were associated with high metabolic independence previously (Watson et al. 2023) (Figure 2f). This result suggests that the PPCN normalization is an important step in comparative analyses of metabolisms between samples with different levels of microbial diversity.”

Lines 178-185 from our original submission have been removed to avoid further confusion. These results can be found in Supplementary File 1 (section “Module enrichment without consideration of effect size leads to nonspecific results”).

It is not entirely clear to me what is meant by PPCN normalization. Normalize the number of copy numbers to the overall number of genes?

The idea behind using per-population copy number (PPCN) is to normalize the prevalence of each metabolic module found in an environment with the number of microbial populations within the same sample. PPCN achieves this by dividing the pathway copy numbers by the number of microbial populations in a given metagenome, which we estimate from the frequency of bacterial single-copy core genes. We have updated the description of the per-population copy number (PPCN) calculation to clarify its use:

“Briefly, the PPCN estimates the proportion of microbes in a community with a particular metabolic capacity (Figure 1, Supplementary Figure 2) by normalizing observed metabolic module copy numbers with the ‘number of microbial populations in a given metagenome’, which we estimate using the single-copy core genes (SCGs) without relying on the reconstruction of individual genomes.”

We also note that the equation for PPCN is shown in Figure 1.

It is also not clear to me how the classifier predicts stress on microbiomes rather than dysbiosis.

The reviewer asks an interesting question since it is true that we could also use the term “dysbiosis” rather than “stress”. Yet we refrained from the use of dysbiosis as it is considered a poorly-defined term to describe an altered microbiome often associated with a specific disease (doi:https://doi.org/10.3390/microorganisms10030578), such as IBD, relative to another poorly-defined state, “healthy microbiome” (doi:https://doi.org/10.1002/phar.2731). We do consider that stress is not necessarily a term that is less vague than dysbiosis, yet it has the advantage of being more common in studies of ecology compared to dysbiosis. Our relatively neutral stance towards which term to use has shifted dramatically due to one critical observation in our study: the identical patterns of enrichment of HMI microbes in individuals diagnosed with IBD as well as in healthy individuals treated with antibiotics. We appreciate that the observed changes in the antibiotics case can also fulfill the definition of “dysbiosis”, but the term “stress response” more accurately describes what the classifier identifies in our opinion.

What is the advantage of using the estimate-metabolism pipeline presented in this article over workflows such as those using genome-scale models, which are repeatedly cited and discussed?

Genome-scale models are often appropriate for a big-picture view of metabolism, and especially when the capability to perform quantitative simulations like flux-balance analysis is needed. For our investigation, we wanted a more specific and descriptive summary of metabolic capacity, so we focused on individual KEGG modules, which qualitatively describe subsets of the vast metabolic network with pathway names that all readers can understand, rather than working with an abstract model of the entire network. Furthermore, genome-scale models would have prevented us from assessing the redundancy (copy number) of metabolic pathways, as these networks usually focus on the presence-absence of gene annotations for enzymes in the network rather than the copy number of these annotations. The copy number metric has been critical for our analyses, considering that we are focusing on metabolic capacity at the community level and require the ability to normalize this metabolic capacity by the size of the community described by each metagenome. Finally, assessing a discrete set of metabolic pathways yielded a corresponding set of features that we used to create the machine learning classifier, whereas data from genome-scale models would not be as easily transferable into classifier features.

Minor comments:Figure 2d and e are mentioned in the text before Figure 2a.

We thank the reviewer for catching this. We have rewritten the section as follows to put the figure references in numerical order:

!To gain insight into potential metabolic determinants of microbial survival in the IBD gut environment, we assessed the distribution of metabolic modules within samples from each group (IBD and healthy) with and without using PPCN normalization. Without normalizing, module copy numbers were overall higher in healthy samples (Figure 2a) and modules exhibited weak differential occurrence between cohorts (Figure 2b, 2c, Supplementary Figure 3). After the application of PPCN, most metabolic modules were elevated in IBD (Supplementary Figure 5). This observation is a product of two independent aspects of the healthy and IBD microbiota. The first one is the increased representation of microbial organisms with smaller genomes in healthy individuals (Watson et al. 2023), which increases the likelihood that the overall copy number of a given metabolic module is below the actual number of populations. In contrast, one of the hallmarks of the IBD microbiota is the generally increased representation of organisms with larger genomes (Watson et al. 2023). The second aspect is that the generally higher diversity of microbes in healthy individuals increases the denominator of the PPCN due to the higher number of populations detected in these samples. This results in a greater reduction in the PPCN of metabolic modules that are not shared across all members of the diverse gut microbial populations in health. To go beyond this general trend and identify modules that were highly conserved in the IBD group, we first selected those that passed a relatively high statistical significance threshold in our enrichment test (Wilcoxon Rank Sum Test, FDR-adjusted p-value < 2e-10). We then accounted for effect size by ranking these modules according to the difference between their median PPCN in IBD samples and their median PPCN in healthy samples, and keeping only those in the top 50% (which translated to an effect size threshold of > 0.12). This stringent filtering revealed a set of 33 metabolic modules that were significantly enriched in metagenomes obtained from individuals diagnosed with IBD (Figure 2d, 2e), 17 of which matched the modules that were associated with high metabolic independence previously (Watson et al. 2023) (Figure 2f). This result suggests that the PPCN normalization is an important step in comparative analyses of metabolisms between samples with different levels of microbial diversity.!

How much preparation is needed for users that want to apply the estimate-metabolism pipeline to their own datasets? From the documentation at anvi'o, it still seems like a significant effort.

We thank the reviewer for this important question. The use of anvi-estimate-metabolism is simple, but the concept it makes available and the means it offers its users to interact with their data are not basic, thus its use requires *some* effort. Anvi’o provides users with the ability to directly interact with their data at each step of the analysis to have full control over the analysis and to make informed decisions on the way. In comparison to pre-defined analysis pipelines that often require no additional input from the user, this approach requires some level of involvement of the user throughout the process – namely, they must run a few programs in series rather than running just one pipeline command that quietly handles everything on their behalf. The most basic workflow for using `anvi-estimate-metabolism` is quite straightforward and requires four simple steps following the installation of anvi’o: 1. Run the program `anvi-setup-kegg-data` to download the KEGG data. 2. Convert the assembly FASTA file into an anvi’o-compatible database format with gene calls by running `anvi-gen-contigs-database`. 3. Annotate genes with KOs with the program `anvi-run-kegg-kofams`. 4. Get module completeness scores and copy numbers by running `anvi-estimate-metabolism`. In addition, we provide simple tutorials (such as the one at https://anvio.org/tutorials/fmt-mag-metabolism/) and reproducible bioinformatics workflows online (including for this study at https://merenlab.org/data/ibd-gut-metabolism/) which helps early career researchers to apply similar strategies to their own datasets. We are happy to report that we have been using this tool in our undergraduate education, and observed that students with no background in computation were able to apply it to their questions without any trouble.

**Reviewer 2:**
Congratulations on this great work, the manuscript is a pleasure to read. Minor questions that the authors might want to clarify:L 275: Why use reference genomes from the GTDB (for only 3 phyla) instead of using MAGs reconstructed from the data? I understand that assemblies based on individual samples would probably not yield enough complete MAGs, but I would expect that co-binning the assemblies for the entire dataset would.

We thank the reviewer for their kind words. We certainly agree that metagenome assembled genomes (MAGs) reconstructed directly from the assemblies would by nature represent the populations in these communities better than reference genomes. However, one of our aims in this study was to avoid the often error-prone and time-consuming step of reconstructing MAGs. Most automatic binning algorithms inevitably make mistakes, and especially for metabolism estimation, low quality MAGs can introduce a bias in the analysis. At the same time the manual curation of each bin to remove any contamination would require a substantial effort and make the workflow less accessible for others to use. As an example, in our previous work (doi:10.1186/s13059-023-02924-x), careful refinement of MAGs from just two co-assemblies took two months. Here, we developed the PPCN workflow as a more scalable, assembly-level analysis to avoid the need for binning in the first place.

To supplement and confirm the metagenome-level results, we decided to run a genome-level analysis. We used the GTDB since it represents the most comprehensive, dereplicated collection of reference genomes across the tree of life. We chose those 3 phyla in particular because of their ecological relevance in the human gut environment. Bacteroidetes and

Firmicutes together represent the majority (up to ~90%) of the populations in healthy individuals (doi:10.1038/nature07540), and Proteobacteria represent the next most abundant phylum on average (2% ± 10%) (doi:10.1371/journal.pone.0206484).

L 403: Should the Franzosa and Papa papers be referenced as numbers?

Thanks for pointing this out. The rogue numerical citation was actually an artifact of the submission and was corrected to a long-format citation in the online version of the manuscript on the eLife website.

**Reviewer 3:**
The lack of any experimental validation contributes to the tentative nature of the conclusions that can be drawn at this time. Numerous studies have looked at the metabolism of gut bacterial species during in vitro growth, which could be mined to test if the in silico predictions of metabolism can be supported. Alternatively, the authors could isolate key strains of interest and study them in culture or in mouse models of IBD.

We appreciate these suggestions and agree with the reviewer that experimental validation is important. However, we do not agree that either the use of mouse models or the isolation of individual microbial strains would be an appropriate experimental test in this case. The use of humanized gnotobiotic mice has critical limitations (see doi:10.1016/j.cell.2019.12.025 and references within the section on “human microbiota-associated murine models”). As it is not possible to establish a mouse model whose gut microbiota fully reflect the human gut microbiome, such an approach would neither be appropriate to validate our findings, nor would it have been possible to produce the insights we have gained based on environmental data. We are not sure how exactly a mouse model, even when ignoring the well established limitations, could improve or validate a comprehensive analysis of a large “environmental” datasets that resulted in highly significant signals.

We are also not sure that we understand how the reviewer believes that the isolation of individual strains would aid in validating our findings. While we appreciate that not all relevant genes are captured by the available annotation routines and that some genes may be misannotated, the large dataset used here renders these concerns negligible. Isolating a small subset of bacterial populations would hardly lead to a representative sample and testing their metabolic capacities in vitro would not improve the reliability of our analysis.

Boilerplate suggestions as vague as “isolate key strains of interest” or “experiment in mouse models of IBD” do not add or retract anything from our findings. Our findings and hypotheses are well supported by our data and extensive analyses.

Line 9 - not sure this approach is hypothesis testing in the traditional sense, you might reword.

Hypothesis testing occurs when one makes an observation, develops a hypothesis that explains the observation, and then gathers and analyzes data to investigate whether additional data support or disprove the hypothesis. We are not convinced a reword is necessary.

Line 40 - the lack of consistent differences in IBD and healthy individuals does not mean that the microbiome doesn't impact disease. It's important to consider all the mechanistic studies in animal models and other systems.

Our study does not claim that microbiome has no impact on the course of disease.

Line 50 - this seemed out of place and undercuts the current findings. Upon checking Ref. 31, the analysis seems distinct enough to not mention in the introduction.

We disagree. Ref 31 uses genome-scale metabolic models to identify the loss of cross-feeding interactions in the gut microbiome of individuals with IBD, which is another way of saying that the microbes in IBD no longer rely on their community for metabolic exchange – in other words, they are metabolically independent. This is an independent observation that is parallel to our results and confirms our analysis; hence, it is important to keep in our introduction.

Line 55 - Ref. 32 looked at FMT, which should be explicitly stated here.

The reviewer’s suggestion is not helpful. Ref 32 has a significant focus on IBD as it compares a total of 300 MAGs generated from individuals with IBD to 264 MAGs from healthy individuals and shows differences in metabolic enrichment between healthy and IBD samples independent of taxonomy, thus setting the stage for our current work. What model has been used to generate the initial insights that led to the IBD-related conclusion in Ref 32 has no significance in this context.

Lines 92-107 - this text is out of place in the Results section and reads more like a review article. Please trim it down and move it to the introduction.

We would like to draw the reviewer’s attention to the fact that this is a “Result and Discussion” section. In this specific case it is important for readers to appreciate the context for our new tool, as the reviewer commented in the public review. We kindly disagree with the reviewer’s suggestion to remove this text as that would diminish the context.

Line 107 - is "selection" the word you meant to use?

If the frequency of a given metabolic module remains the same or increases despite the decreasing diversity of the microbial community, it is conceivable to assume that its enrichment indicates the presence of a selective process to which the module responds. It is indeed the word we meant to use.

Line 110 - this is the first mention of this new method, need to add it to the abstract and introduction.

The reviewer must have overlooked the text passages in which we mention the strategy we developed within the abstract:

“Here, we tested this hypothesis on a large scale, by developing a software framework to quantify the enrichment of microbial metabolisms in complex metagenomes as a function of microbial diversity.”

And in the last paragraph of the introduction:

“Here we implemented a high-throughput, taxonomy-independent strategy to estimate metabolic capabilities of microbial communities directly from metagenomes…”

Figure 1 - a nice summary, but no data is shown to support the validity of this model. Consider shrinking the cartoon and adding validation with simulated datasets.

We hope we have addressed this recommendation with the extensive validation efforts summarized above.

Line 134 - need to state the FDR and effect size cutoffs used.

We have reworded this sentence as follows to clarify which thresholds were used:

“We identified significantly enriched modules using an FDR-adjusted p-value threshold of p < 2e-10 and an effect size threshold of > 0.12 from a Wilcoxon Rank Sum Test comparing IBD and healthy samples.”

I'm also concerned about the simple comparison of IBD to healthy without adjusting for confounders like study, geographical location, age, sex, drug use, diet, etc. More text is needed to explain the nature of these data, how much metadata is available, and which other variables distinguish IBD from healthy.

The reviewer is correct that there is a large amount of interindividual variation between samples due to host and environmental factors. However, the lack of adjusting for confounders was intentional, and in fact one of the critical strengths of our study. We observe a clear signal between healthy individuals and individuals diagnosed with IBD, *despite* the amount of interindividual variation in our diverse set of samples from 13 different studies (details of which are summarized in Supplementary Table 1). The clear increase in predicted metabolic capacity that we consistently observe in IBD patients using both metagenomes and genomes across diverse cohorts points to metabolic independence as a high-level trend that is predictive of microbial prevalence in stressed gut environments irrespective of host factors.

Line 145 - calling PPCN normalization an "essential step" is a huge claim and requires a lot more data to back it up. Might be best to qualify this statement.

We hope we have addressed this recommendation with our validation efforts. Supplementary Figures 18 and 19 in particular show evidence for the necessity of the normalization step. It is indeed an essential step *if* the purpose is to compare metabolic enrichment between cohorts of highly different microbial diversity.

Figure 2a - the use of a 1:1 trend line seems potentially misleading. I would replace it with a best-fit line.

Our purpose here was not to show the best fit. Instead, the 1:1 trend line separates the modules based on their relative abundance distribution between healthy individuals and individuals diagnosed with IBD. If the module is to the left of the line, it has a higher median copy number in healthy individuals and if the module is to the right, it has a higher median copy number in individuals with IBD. The line also helps to demonstrate the shift that occurs between the unnormalized data in Figure 2a. Without the normalization, more modules occur to the left of the

1/1 line as a result of the higher raw copy numbers in healthy metagenomes which simply contain more microbial populations. With the normalization (Figure 2d), more modules fall on the right side of the 1/1 line due to higher PPCN values. A best-fit line would not serve well for these purposes.

The text should be revised to state that this analysis actually did find many significant differences and to discuss whether they were the same modules identified in Figure 2d.

We apologize for the confusion and thank the reviewer for bringing this issue to our attention. As mentioned above, the disparate levels of microbial diversity between healthy individuals and individuals with IBD resulted in much larger copy numbers of metabolic modules in healthy samples reflecting the often much larger communities. Hence, we ran statistical tests only on normalized (PPCN) data. The p-values associated with each module in Figure 2a, as well as the colors of each point, are based on the PPCN data in Figure 2d. We aimed to improve the clarity of the visual comparison between normalized and unnormalized results by identifying the same set of IBD-enriched modules in plots a-c and plots d-f.

That being said, the reviewer’s comment made us realize the potential for confusion when using the normalized data’s statistical results in Figure 2a that otherwise shows results from unnormalized data. We have now run the same statistical test on the unnormalized (raw copy number) data and re-generated Figure 2a with the new FDR-adjusted p-values and points colored based on the statistical tests using unnormalized data. We’ve also removed the arrow connecting to Figure 2b (since we no longer show the same set of IBD-enriched modules in Figures 2a and 2b), and added a dashed line to indicate the effect size threshold (similar to the one in Figure 2d). We have updated the legend for Figure 2a-d to reflect these changes:

When we used the same p-value threshold (p < 2e-10) as before and also filtered for an effect size larger than the mean (the same strategy used to set our effect size threshold for the normalized data), there are 10 modules that are significantly enriched based on the unnormalized data. Of course, it is difficult to gauge the relevance of these 10 modules to microbial fitness in the IBD gut environment since their raw copy numbers do not tell us anything about the relative proportion of community members that harbor these modules. Therefore, we are reluctant to add these modules to the results text. For the record, only 3 of those modules were also significantly enriched based on the normalized PPCN values: M00010 (Citrate cycle, first carbon oxidation), M00053 (Pyrimidine deoxyribonucleotide biosynthesis), and M00121 (Heme biosynthesis).

Figure 2c,f - these panels raise a lot of concerns given that the choice of method inverts the trend. Without additional data/validation, it's hard to know which method is right.

We hope we have addressed this recommendation with the extensive validation efforts summarized above. Inversion of the trend is an expected outcome, because the raw copy numbers of most metabolic modules are much lower in the IBD sample group due to lower community sizes.

Line 167 - Need to take the KEGG names with a grain of salt, just because it says "biosynthesis" doesn't mean that the pathway goes in that direction in your bacterium of interest.

We believe the reviewer is under a misapprehension regarding the general reversibility of KEGG metabolic modules, or indeed of metabolic pathways. Most metabolic pathways have one or several (practically) irreversible reactions. To demonstrate this for the 33 IBD-enriched modules, we evaluated their reversibility based upon their corresponding KEGG Pathway Maps, which indicate reaction reversibility via double-sided arrows. Aside from the signature modules M00705 and M00627, in 26 out of 31 pathway modules one or more irreversible reactions render these pathways one-directional. Indeed, on average the majority (54%) of the reactions in a given module are irreversible. When focusing on the 23 “biosynthesis” modules, 22 out of 23 (96%) modules have at least one irreversible reaction, and on average 64% of a given module’s reactions are irreversible. These data (which can be accessed at doi:10.6084/m9.figshare.27203226 for the reviewer’s convenience) challenge the reviewer’s notion that pathway directionality is free to change arbitrarily, since the presence of even one irreversible reaction effectively blocks the flux in the opposing direction. Thus, “biosynthesis” is indeed a meaningful term in KEGG module names.

That said, KEGG Pathway Maps, though highly curated, are likely not the final word on whether a given reaction in a metabolic pathway can be considered reversible or irreversible in each microbial population and under all conditions. And our analysis, like many others that rely on metagenomic data, does not consider the environmental conditions in the gut such as temperature or metabolite concentrations that might influence the Gibbs free energy and thus the directionality of these reactions in vivo. However, even assuming general reversibility of metabolic pathways, this would not invalidate the fact that these microbes have the metabolic capacity to synthesize the respective molecules. In other words, the potential reversibility of pathways is irrelevant to our analysis since we are describing metabolic *potential*. The *lac* operon in *E. coli* might only be expressed in the absence of glucose, but *E. coli* always has the capability to degrade lactose regardless of whether that pathway is active. Thus, our overall conclusion that gut microbes associated with IBD are metabolically self-sufficient (encoding the enzymatic capability to synthesize certain key metabolites) remains valid irrespective of fixed or flexible pathway directionality.

It's also important to be careful not to conflate KEGG modules (small subsets of a pathway) with the actual metabolic pathway. It's possible to have a module change in abundance while not altering the full pathway. Inspection of the individual genes could help in this respect - are they rate-limiting steps for biosynthesis or catabolism?

The reviewer is absolutely correct that KEGG modules do not necessarily represent full pathways. We have updated the language in our manuscript to explicitly refer to “modules” rather than “pathways” whenever appropriate, to restrict the scope of the analysis to metabolic modules rather than full pathways.

That said, we do not see how “inspection of individual genes” would improve our analysis. The strength of looking at complete modules rather than individual genes is that we can gain conclusive insights into a certain metabolic capacity. Of course, no pathway or module stands alone. However, the enrichment of metabolic modules does conclusively indicate that these modules are beneficial under the given conditions, such as stress caused by inflammation or antibiotic use. Whether a certain step in a module or pathway is rate limiting is completely irrelevant for this analysis.

Line 177 - I'm not a big fan of the HMI acronym. Is there a LMI group? It seems simplistic to lump all of metabolism into dependent or independent, which in reality will differ depending on the specific substrate, the growth condition, and the strain.

While we are sorry that our study failed to provide the reviewer with a term they could be a fan of, their input did not change our view that HMI, an acronym we have adapted from a previously peer-reviewed study (doi:10.1186/s13059-023-02924-x), is a powerfully simplistic means to describe a phenomenon we observe and demonstrate in multiple different ways with our extensive analyses. The argument that HMI or LMI status will differ given the growth condition, substrate availability, or strain differences is not helping this case either: our analyses cut across a large number of humans and naturally occurring microbial systems in their guts that are exposed to largely variable ‘growth conditions’ and ‘substrates’ and composed of many strain variants of similar populations. Yet, we observe a clear role for HMI despite all these differences. Perhaps it is because HMI simply describes a higher metabolic capacity based on a defined subset of largely biosynthetic pathways that we observe to be consistently enriched in a large dataset covering a large variety of host, environmental and diet factors and indicates that a population has a higher metabolic capacity to not rely on ecosystem services. We show in our analysis that in the inflamed gut these capacities are indeed required, which is why HMI populations are enriched in IBD samples. HMI has no relation to any of the constraints mentioned by the reviewer, which is one of the major strengths of this metric.

Line 198 - It seems like a big assumption to state that efflux and drug resistance are unrelated to biosynthesis, as they could be genetically or even phenotypically linked.

We agree with the reviewer and are thankful for their input. We have weakened the assertion in this statement.

“These capacities may provide an advantage since antibiotics are a common treatment for IBDs (Nitzan et al. 2016), but are not necessarily related to the systematic enrichment of biosynthesis modules that likely provide resilience to general environmental stress rather than to a specific stressor such as antibiotics.”

Lines 202-218 - I'd suggest removing this paragraph. The "non-IBD" data introduces even more complications to the meta-analysis and seems irrelevant to the current study.

We thank the reviewer for this suggestion. Non-IBD data is important, but its relevance to the primary aims of the study is indeed negligible. We now have moved this paragraph to Supplementary File 1 (under the section “‘Non-IBD’ samples are intermediate to IBD and healthy samples”).

The health gradient is particularly problematic, putting cancer closer to healthy than IBD.

We took the reviewer’s advice and have swapped the order of the studies in Supplementary Figure 6 to place the cancer samples from Feng et al. closer to the IBD samples, on the other side of the non-IBD samples from the IBD studies.

Lines 235-257 - should trim this down and move to the discussion.

As mentioned above, we have opted for a “Results and Discussion format” for our manuscript, so we believe this discussion is in the correct place. We find it important to clearly highlight the limitations and potential biases of our work and trimming this text would take away from that goal.

Figure 3 - panels are out of order. Need to put the current panel D below current panel C. Also, relabel panel letters to go top to bottom (the bottom panel should be D). Could change current panel 3D to a violin plot to match current 3C.

We have updated Figure 3 by converting panel A into a new supplementary figure (Supplementary Figure 8), moving panels C and D below panel B, and relabeling the panels accordingly.

Figure 3B - this panel was incredibly useful and quite surprising to me in many respects. I would have assumed that the Bacteroides would be in the "HMI" bin. Is this a function of the specific strains included here? Was B. theta or B. fragilis included?

The reviewer makes an excellent observation that has been keeping us awake at night, yet somehow was not appropriately discussed in the text until their input. We are very thankful for their attention to detail here.

It is indeed true that *Bacteroides* genomes are often detected with increased abundance in individuals with IBD and likely have a survival advantage in the IBD gut environment, *Bacteroides fragilis* and *Bacteroides thetaiotaomicron* being some of the most dominant residents of the IBD gut. Their non-HMI status is not a function of which strains were included, since all taxa here are represented by the representative genomes available in the publicly available Genome Taxonomy Database. Their non-HMI status comes from the fact that they have HMI scores of around 24 to 26, which fall slightly below the threshold score of 26.4 that we used to classify genomes as HMI. This threshold is back-calculated from the metabolic completion requirement of at least 80% average completion of all 33 metabolic modules that are significantly enriched in IBD. So these genomes are right there at the edge, but not quite over it.

Thanks to this comment by our reviewer, we started wondering whether we should follow a more ‘literature-driven’ approach to set the threshold for HMI, rather than the 80% cutoff, and in fact attempted to lower the HMI score threshold to see if we could include more of the IBD-associated *Bacteroides* in the HMI bin. Author response table 1 below shows the relevant subset of our new Supplementary Table 3h, which describes the data from our tests on different thresholds.

**Author response table 1. sa4table1:** Number and proportion of *Bacteroides* genomes classified as HMI at each HMI score threshold. There were 20 total *Bacteroides* genomes in the set of 338 gut microbes identified from the GTDB. The HMI score is computed by adding the percent completeness of all 33 IBD-enriched KEGG modules. The full table can be viewed in Supplementary Table 3h.

Average percentcompleteness of	Corresponding HMIscore threshold	Number of	
IBD-enrichedmodules	24.75	Bacteroides genomesclassified as HMI	Hercent of Bacteroidesgenomes classified as
75	25.08	6	30%
76	25.41	4	20%
77	25.74	3	15%
78	26.07	0	0%
79	26.4	0	0%
80			

Lowering the threshold to 24.75, which corresponds to an average of 75% completeness in the 33 IBD-enriched modules, enabled the classification of 6 *Bacteroides* genomes as HMI, including *B. fragilis*, *B. intestinalis*, *B. theta,* and *B. faecis*. However, it also identified several microbes that are not IBD-associated as HMI, including 75 genomes from the Lachnospiraceae family and 18 genomes from the Ruminococcaceae family. In the latter family, several *Faecalibacterium* genomes, including 10 representatives of *Faecalibacterium prausnitzii*, were considered HMI using this threshold. These microbes are empirically known to decrease in abundance during inflammatory gastrointestinal conditions (doi:10.3390/microorganisms8040573, doi:10.1093/femsre/fuad039), and therefore these genomes should not be considered HMI – at least not under the working definition of HMI used in our study. To avoid including such a large number of obvious false positives in the HMI bin, we decided to maintain a higher threshold despite the exclusion of *Bacteroides* genomes.

This outcome demonstrates that our reductionist approach does not successfully capture every microbial population that is associated with IBD. Nevertheless, and in our opinion very surprisingly, the metric does capture a very large proportion of genomes with increased detection and abundance in IBD samples, as demonstrated by the peaks of detection/abundance that match to HMI status Author response image 1.

**Author response image 1. sa4fig1:** Screenshots of Figure 3 that demonstrate the overlapping signal between HMI status and genome detection/abundance in IBD.

Furthermore, the violin plots in Figure 3B (formerly Figure 3C) clearly reflect the increased representation of HMI populations in IBD metagenomes. Although our classification method is imperfect, it still demonstrates the predictive power of metabolic competencies in identifying which microbes will survive in stressful gut environments. To ensure that readers recognize the crude nature of this classification strategy and the possibility that high metabolic independence can be achieved in different ways, we have added the following sentences to the relevant section of our manuscript:

“Given the number of ways a genome can pass or fail this threshold, this arbitrary cut-off has significant shortcomings, which was demonstrated by the fact that several species in the Bacteroides group were not classified as HMI despite their frequent dominance of the gut microbiome of individuals with IBD (Saitoh et al. 2002; Wexler 2007; Vineis et al. 2016) (Supplementary File 1). That said, the genomes that were classified as HMI by this approach were consistently higher in their detection and abundance in IBD samples (Figure 3a). It is likely that there are multiple ways to have high metabolic independence which are not fully captured by the 33 IBD-enriched metabolic modules identified in this study.”

We have also included a discussion of these findings in Supplementary Information File 1 (see section “Examining the impact of different HMI score thresholds on genome-level results”).

This panel also makes it clear that many of these modules are widespread in all genomes and thus unlikely to meaningfully differ in the microbiome. It would be interesting to use this type of analysis to identify a subset of KEGG modules with high variability between strains.

The figure makes it ‘look like’ many of these modules are widespread in all genomes and thus unlikely to meaningfully differ in the microbiome, but our quantitative analyses clearly demonstrate that these modules indeed differ meaningfully between microbiomes of healthy individuals and those diagnosed with IBD. For instance, the classifier that we built relying exclusively upon these modules’ PPCN values was able to reliably distinguish between the healthy and IBD sample groups in our dataset. The fact that the differentiating signal does not rely on rare metabolic or signature modules is what makes the classifier powerful enough to differentiate between “healthy” and “stressed” microbiomes in 86% of cases. Modules that are by nature less common could not serve this purpose. That said, we do agree with the reviewer that it might be interesting to study variability of KEGG modules as a function of variability between strains. This does not fall into the scope of this work, but we hope to assist others with the technical aspects of such work.

Considering the entirety of the exchange in this section, perhaps there is a broader discussion to be had around this topic. In retrospect, not being able to perfectly split microbes into two groups that completely recapitulate their enrichment in healthy or IBD samples by a crude metric and an arbitrary threshold is not surprising at all. What is surprising is that such a crude metric in fact works for the vast majority of microbes and predicts their increased presence in the IBD gut by only considering their genetic make up. In some respects, we believe that the inability of this cutoff to propose a perfect classifier is similar to the limited power of metabolic independence concept and the classes of HMI or LMI to capture and fully explain microbial fitness in health and disease. What is again surprising here is that these almost offensively simple classes do capture more than what one would expect. We can envision a few ways to implement a more sophisticated HMI/LMI classifier, and it is certainly an important task that is achievable. However, we are hopeful that this technical work can also be done better by others in our field, and that step forward, along with further scrutinizing the relevance of HMI/LMI classes to understand metabolic factors that contribute to the biodiversity of stressful environments, will have to remain as future work.

We thank the reviewer again for their comment here and pushing us to think more carefully and address the oddity regarding the poor representation of *Bacteroides* as HMI by our cutoff.

Given that a lot of the gaps are in the Firmicutes, this panel also makes me more concerned about annotation bias. How many of these gaps are real?

Analyses relying on gene annotations all suffer equally from the potential for missannotation or missing annotations, which primarily result from limitations in our reference databases for functional data. For instance, the Hidden Markov models for microbial genes in the KEGG Ortholog database are generated from a curated set of gene sequences primarily originating from cultivable microorganisms and particularly from commonly-used model organisms; hence, they do not capture the full extent of sequence diversity observed in populations that are less well-represented in reference databases – a category which includes several Firmicutes, as the reviewer points out. For KEGG KOfams in particular, the precomputed bit score thresholds for distinguishing between ‘good’ and ‘bad’ matches to a given model are often too stringent to enable annotation of genes that are just slightly too divergent from the set of known sequences, thus resulting in missing annotations. Based on our experience with these sorts of issues, we implemented a heuristic that reduces the number of missing annotations for KOs and captures significantly more homologs than other state-of-the-art approaches, as described in doi:10.1101/2024.07.03.601779. We refer the reviewer to our response to the related public comment about annotation bias above, which includes additional details about our investigations of annotation bias in our data. In comparison to the current standard, the heuristic we implemented improves functional annotation results. However, neither our nor any other bioinformatic study that relies on functional gene annotation can exclude the potential for annotation bias.

Figure 3B plotting issues - need to use the full names of the modules; for example, M00844 is "arginine biosynthesis, ornithine => arginine", which changes the interpretation. Need a key for the heatmap on the figure. The tree is difficult to see, needs a darker font.

We have darkened the lines of the tree and dendrogram, and added a legend for the heatmap gradient (see new version of Figure 3 above). Unfortunately, we could not fit the full names of the modules into the figure due to space constraints. However, the full module name and other relevant information can be found in Supplementary Table 2a, and the matrix of pathway completeness scores in these genomes (e.g., the values plotted in the heatmap) can be found in Supplementary Table 3b. We are not sure what the reviewer refers to when stating that “for example, M00844 is "arginine biosynthesis, ornithine => arginine", which changes the interpretation”. There is no ambiguity regarding the identity of KEGG module M00844, which is arginine biosynthesis from ornithine.

Line 321 - more justification for the 80% cutoff is needed along with a sensitivity analysis to see if this choice matters for the key results.

Inspired by this comment, and the one above regarding the classification of *Bacteroides* genomes, we tested several HMI score thresholds ranging from 75% to 85% average completeness of the 33 IBD-enriched modules. For each threshold, we computed all the key statistics reported in this section of our paper, including the statistical tests. We found that the choice of HMI score threshold does not influence the overall conclusions drawn in this section of our manuscript. Author response table 2 below shows the relevant subset of our new Supplementary Table 3h, which describes the results for each threshold:

**Author response table 2. sa4table2:** Key genome-level results at each HMI score threshold. The HMI score is computed by adding the percent completeness of all 33 IBD-enriched KEGG modules. WRS – Wilcoxon Rank Sum test; KW – Kruskal-Wallis test. The full table can be viewed in Supplementary Table 3h

Average percentcompleteness ofIBD-enriched modules	HMIscorethreshold	Number ofHMIgenomes	WRS p-value for HMI vs non-HMI detection in IBD	WRS p-value forHMI vs non-HMI detection in healthy	KW p-value for fraction of HMI in IBD vs nonIBD samples
75	24.75	129	2.18E-08	0.001776512	1.24E-16
76	25.08	115	2.70E-08	0.002832713	2.54E-19
77	25.41	98	2.10E-06	0.012257189	1.60E-18
78	25.74	78	4.16E-06	0.087182579	4.36E-21
79	26.07	69	5.46E-06	0.078132573	1.75E-21
80	26.4	59	4.82E-06	0.267265165	8.98E-25
81	26.73	48	0.000168971	0.700660484	5.03E-25
82	27.06	39	0.000217924	0.836279996	2.89E-29
83	27.39	35	0.001750966	0.962660229	3.77E-30
84	27.72	28	0.024916547	0.997986302	5.66E-30
85	28.05	24	0.043823626	0.999220691	5.21E-30

We’ve summarized these findings in a new section of Supplementary File 1 entitled “Examining the impact of different HMI score thresholds on genome-level results”. We copy below the relevant text for the reviewer’s convenience:

“Determining the HMI status of a given genome required us to set a threshold for the HMI score above which a genome would be considered to have high metabolic independence. We tested several different thresholds by varying the average percent completeness of the 33 IBD-enriched metabolic modules that we expected from the

‘HMI’ genomes from ≥ 75% (corresponding to an HMI score of ≥ 24.75) to ≥ 85% (corresponding to an HMI score of ≥ 28.05). For each threshold, we computed the same statistics and ran the same statistical tests as those reported in our main manuscript to assess the impact of these thresholds on the results (Supplementary Table 3h). At the highest threshold we tested (HMI score ≥ 28.05), a small proportion of the reference genomes (7%, or n = 24) were classified as HMI, so we did not test higher thresholds.

We found that the results from comparing HMI genomes to non-HMI genomes are similar regardless of which HMI score threshold is used to classify genomes into either group. No matter which HMI score threshold was used, the mean genome size and mean number of genes were higher for HMI genomes than for non-HMI genomes. On average, the HMI genomes were about 1 Mb larger and had 1,032 more gene calls than non-HMI genomes. We ran two Wilcoxon Rank Sum statistical tests to assess the following null hypotheses: (1) HMI genomes do not have higher detection in IBD samples than non-HMI genomes, and (2) HMI genomes do not have higher detection in healthy samples than non-HMI genomes. For both tests, the p-values decreased (grew more significant) as the HMI score threshold decreased due to the inclusion of more genomes in the HMI bin. The first test for higher detection of HMI genomes than non-HMI genomes in IBD samples yielded p-values less than α = 0.05 at all HMI score thresholds. The second test for higher detection of HMI genomes than non-HMI genomes in healthy samples yielded p-values less than α = 0.05 for the three lowest HMI score thresholds (HMI score ≥ 24.75, ≥ 25.08, or ≥ 25.41). However, irrespective of significance threshold and HMI score threshold, there was always far stronger evidence to reject the first null hypothesis than the second, given that the p-value for the first test in IBD samples was 1 to 5 orders of magnitude lower (more significant) than the p-value for the second test in healthy samples.

IBD samples harbored a significantly higher fraction of genomes classified as HMI than healthy or non-IBD samples, regardless of HMI score threshold (p < 1e-15, Kruskal-Wallis Rank Sum test). The p-values for this test increased (grew less significant) as the HMI score threshold decreased. This suggests that, at higher thresholds, relatively more genomes drop out of the HMI fraction in healthy/non-IBD samples than in IBD samples, thereby leading to larger differences and more significant p-values. Consequently, the HMI scores of genomes detected in IBD samples must be higher than the HMI scores of genomes detected in the other sample groups – indeed, the average HMI score of genomes detected within at least one IBD sample is 24.75, while the average score of genomes detected within at least one healthy sample is 22.78. Within a given sample, the mean HMI score of genomes detected within that sample is higher for the IBD group than in the healthy group: the average per-sample mean HMI score is 25.14 across IBD samples compared to the average of 23.00 across healthy samples.”

Lines 357 and 454 - I would remove the discussion of the "gut environment" which isn't really addressed here. The observed trends could just as easily relate to microbial interactions or the effects of diet and pharmaceuticals. Perhaps the issue is the vague nature of this term, which I read to imply changes in the mammalian host. Given the level of evidence, I'd opt to keep the options open and discuss what additional data would help resolve these questions.

We are in complete agreement with the reviewer that microbial interactions are likely an important driver of our observations. In healthy communities, microbial cross-feeding enables microbes with lower metabolic independence to establish and increase microbial diversity. Which is exactly why we are stating that “Community-level signal translates to individual microbial populations and provides insights into the microbial ecology of stressed gut environments”.

Diet or usage of prescription drugs on the other hand, as discussed previously, likely varies substantially over the various cohorts investigated, and is thus not a driver of the observed trends. Instead, HMI works as a high level indicator that is not influenced by these variable host habits.

Lines 354-394 - Could remove or dramatically trim down this text. Too much discussion for a results section.

We kindly remind the reviewer that our manuscript is written following a “Results and Discussion” format. This section provides necessary context and justification for our classifier implementation, so we have left it as-is.

Lines 395-441 - This section raised a lot of issues and could be qualified or even removed. The model was trained on modules that were IBD-associated in the same dataset, so it's not surprising that it worked. An independent test set would be required to see if this model has any broader utility.

The point that we selected the IBD-enriched modules as features should not raise any concerns, as these modules would have emerged as the most important (ie, most highly weighted) features in our model even if we had included all modules in our training data. This is because machine learning classifiers by design pick out the features that best distinguish between classes, and the 33 IBD-associated modules are a selective subset of these (if they were not, they would not have been significantly enriched in the IBD sample group). That said, a carefully conducted feature selection process prior to model training is a standard best-practice in machine learning; thus, if anything, this should be interpreted as a point of confidence rather than a concern. Furthermore, we evaluated our model using cross-validation, a standard practice in the machine learning field that assesses the stability of model performance by training and testing the model on different subsets of the data. This effort established that the model is robust across different inputs as demonstrated by the per-fold confusion matrix and the ROC curve. These are all standard approaches in machine learning to quantify the model tradeoff between bias and variance. As for the independent test set, we went far and beyond, and applied our model to the antibiotic time-series dataset described later in this section, which, in our opinion, and likely also in the opinion of many experts, serves as one of the most convincing ways to test the utility of any model. Classification results here show that our hypothesis concerning the relevance of metabolic independence to microbial survival in stressed gut environments applies beyond the IBD case and includes antibiotic use, which is indeed a stronger validation for this hypothesis than any test we could have done on other IBD-related datasets. Regardless, we agree that any ‘broader’ utility of our model, such as its applications in clinical settings for diagnostic purposes, is something we certainly can not make strong claims about without more data. We have therefore qualified this section by adding the following sentence:

“Determining whether such a model has broader utility as a diagnostic tool requires further research and validation; however, these results demonstrate the potential of HMI as an accessible diagnostic marker of IBD.”

The application to the antibiotic intervention data raises additional concerns, as the model will predict IBD (labeled "stress" in Figure 5) where none exists.

We apologize for this misunderstanding. The label “stress” actually means stress, not IBD. The figure the reviewer is referring to demonstrates that metabolic modules enriched in the gut microbiome of IBD patients are also temporarily enriched in the gut microbiome of healthy individuals treated with antibiotics for the duration of the treatment. While the classifier uses PPCN values for 33 metabolic modules enriched in microbiomes of IBD patients, it does not mean that this enrichment is exclusive to IBD. The classifier will distinguish between metagenomes in which the PPCN values for those 33 metabolic modules is higher and metagenomes in which the PPCN values are lower. Hence, our analysis demonstrates that during antibiotic usage in healthy individuals, the PPCN values of these 33 metabolic modules spike *in a similar fashion* to how they would in the gut community of a person with IBD. This points to a more general trend of high metabolic independence as a factor supporting microbial survival in conditions of stress; that is, the increase in metabolic independence is not specific to the IBD condition but rather a more generic ecological response to perturbations in the gut microbial community. We have clarified this point with the following addition to the paragraph summarizing these results:

“All pre-treatment samples were classified as ‘healthy’ followed by a decline in the proportion of ‘healthy’ samples to a minimum 8 days post-treatment, and a gradual increase until 180 days post treatment, when over 90% of samples were classified as ‘healthy’ (Figure 5, Supplementary Table 4b). In other words, the increase in the HMI metric serves as an indicator of stress in the gut microbiome, regardless of whether that stress arises from the IBD condition or the application of antibiotics. These observations support the role of HMI as an ecological driver of microbial resilience during gut stress caused by a variety of environmental perturbations and demonstrate its diagnostic power in reflecting gut microbiome state.”

We’ve also added the following sentence to the end of the legend for Figure 5:

“Samples classified as ‘healthy’ by the model were considered to have ‘no stress’ (blue), while samples classified as ‘IBD’ were considered to be under ‘stress’ (red).”

Figure S5A - should probably split this into 2 graphs since different data is analyzed.

It is true that different sets of modules are used in either half of the figure; however, there is a significant amount of overlap between the sets (17 modules), which is why there are lines connecting the points for the same module as described in the figure legend. We are using this figure to make the point that the median PPCN value of each module increases, in both sets of modules, from the healthy sample group to the IBD sample group. Therefore, we believe the current presentation is appropriate.

Figure S6A – this shows a substantial study effect and raises concerns about reproducibility.

We examined potential batch effects in Supplementary Information File 1 (see section “Considerations of Batch Effect”), and found that any study effect was minor and overcome by the signal between groups:

“The similar distribution of the median normalized copy number for each of the 33 IBD-enriched metabolic modules (summarized across all samples within a given study), across all studies within a given sample group (Supplementary Figure 6b), confirms that the sample group explains more of the trend than the study of origin.”

Furthermore, within Supplementary Figure 6a, there is a clear increase between the non-IBD controls from Franzosa et al. 2018 and the IBD samples from the same study, as well as between the non-IBD controls from Schirmir et al. 2018 and the IBD samples from that study. As there is no study effect influencing those two comparisons, this reinforces the evidence that there is a true increase in the normalized copy numbers of these modules when comparing samples from more healthy individuals to those from less healthy individuals.

Figure S7B - check numbers, which I think should sum to 33.

The numbers should not sum to 33. In this test to determine whether the two largest studies had excessive influence on the identity of the IBD-enriched modules, we repeated our strategy to obtain 33 IBD-enriched modules (those with the 33 smallest p-values from the statistical test) from each set of samples – either (1) samples from Le Chatelier et al. 2013 and Vineis et al. 2016, or (2) samples that are not from those two studies. The 2 sets, containing 33 modules each, gives us a total of 66 IBD-enriched modules. By comparing those two sets, we found that 20 modules were present in both sets – hence the value of 20 in the center of the Venn Diagram. In each set, 13 modules were unique – hence the value of 13 on either side. 13 + 13 + 2*20 = 66 total modules.

We again thank our reviewers for their time and interest, and invaluable input.